# Neuro-computational mechanisms and individual biases in action-outcome learning under moral conflict

Laura Fornari[1,5], Kalliopi Ioumpa[1,5], Alessandra D. Nostro[1], Nathan J. Evans[2], Lorenzo De Angelis[1], Sebastian P. H. Speer[1], Riccardo Paracampo[1], Selene Gallo[1], Michael Spezio [3], Christian Keysers [1,4,6] & Valeria Gazzola [1,4,6] ✉

Learning to predict action outcomes in morally conflicting situations is essential for social decision-making but poorly understood. Here we tested which forms of Reinforcement Learning Theory capture how participants learn to choose between self-money and other-shocks, and how they adapt to changes in contingencies. We find choices were better described by a reinforcement learning model based on the current value of separately expected outcomes than by one based on the combined historical values of past outcomes. Participants track expected values of self-money and other-shocks separately, with the substantial individual difference in preference reflected in a valuation parameter balancing their relative weight. This valuation parameter also predicted choices in an independent costly helping task. The expectations of self-money and other-shocks were biased toward the favored outcome but fMRI revealed this bias to be reflected in the ventromedial prefrontal cortex while the pain-observation network represented pain prediction errors independently of individual preferences.

We often have to learn that certain actions lead to favorable outcomes for us, but harm others, while alternative actions are less favorable for us but avoid or mitigate harms to others[1]. Much is already known about the brain structures involved in making moral choices when the relevant action-outcome contingencies are known[2–9], but how we learn these contingencies remains poorly understood, especially in situations pitting gains to self against losses for others.

Reinforcement learning theory (RLT) has successfully described how individuals learn to benefit themselves[10,11] and most recently, how they learn to benefit others[12–15]. At the core of reinforcement learning is the notion that we update expected values (EV) of actions via prediction errors (PE)−the differences between actual outcomes and expected values. Ambiguity in morally relevant action-outcome associations raises specific questions with regard to RLT, especially if outcomes for

self and others conflict. If actions benefit the self and harm others, are these conflicting outcomes combined into a common valuational representation; or do we track separate expectations for benefits to the self and harm to others[16]? Also, people differ in how they represent benefits and harms to self[17], and in whether they prefer to maximize benefits for the self vs. minimize harms to others[3,4,6]. How can such differences be computationally represented using RLT? Would people maximizing benefits for the self show reduced prediction errors and expected value signals for other harm, as motivated accounts of empathy may suggest[18,19], or are expectations tracked independently of one's preferences, such that preferences only play out when decisions are made? These important questions can only be addressed by studying the dynamics and neural underpinnings of action-outcome associations while outcomes for self and others conflict.

[1]Netherlands Institute for Neuroscience, KNAW, Meibergdreef 47, 1105BA Amsterdam, The Netherlands. [2]School of Psychology, University of Queensland, Brisbane, QLD, Australia. [3]Psychology, Neuroscience, & Data Science, Scripps College, 1030 Columbia Ave, CA 91711 Claremont, CA, USA. [4]Department of Psychology, University of Amsterdam, Nieuwe Achtergracht 129-B, 1018 WT Amsterdam, The Netherlands. [5]These authors contributed equally: Laura Fornari, Kalliopi Ioumpa. [6]These authors jointly supervised this work: Christian Keysers, Valeria Gazzola. ✉e-mail: v.gazzola@nin.knaw.nl

To address these questions, we combined online behavioral and fMRI data from two independent studies. In the core task (Conflict condition, Fig. 1a, b; Table 1), common to both experiments, participants had to learn that one of two symbols led to high monetary gains

for the self 80% of the time, and to a painful but tolerable shock to the hand of a confederate with the same probability. We refer to this symbol as 'lucrative', since it was associated with higher monetary outcomes. The other symbol led to low monetary gains for the self 80%

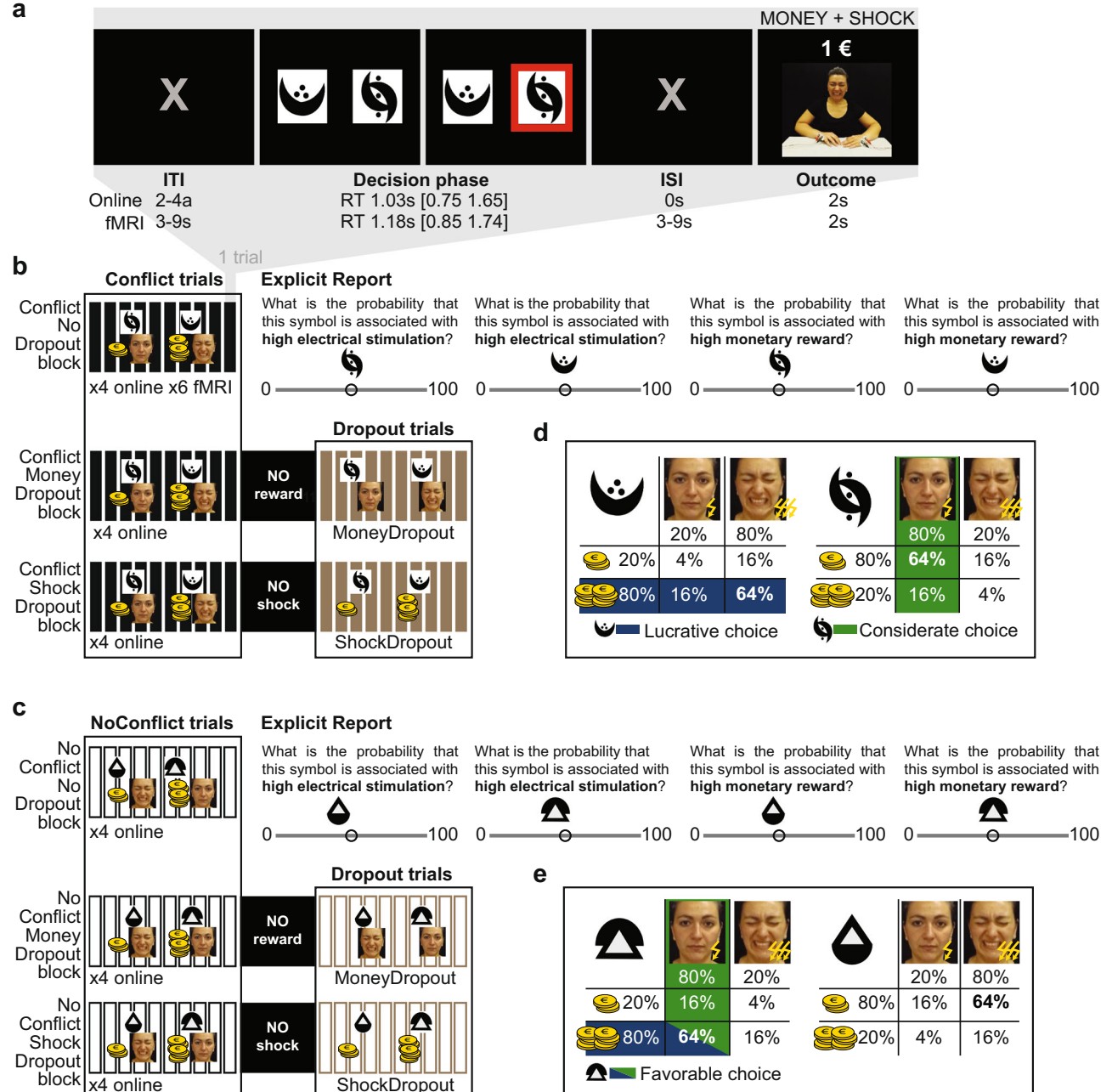

**Fig. 1 | Learning task. a** Trial structure. Red outline indicates an exemplary choice for a particular trial. Pictures showing the confederate's response are still frames captured from one of the videos used in the Online experiment in which the confederate received a painful shock (Supplementary Note §1). Inter-trial (ITI) and inter-stimulus (ISI) intervals were adapted to the fMRI and Online situation, and are indicated separately below each relevant instance of the trial. For illustrative purposes, median reaction times (RT), and 25% and 75% quartiles, are estimated from the fMRI and Online ConflictNoDropout blocks, and the first 10 T of the Online ConflictDropout blocks, as they have the same structure as ConflictNoDropout blocks. When considering all Online trials, the median is 0.91 [0.67 1.38]. **b, c** Different types of block used for the Conflict (**b**; filled in rectangles) and NoConflict (**c**; empty rectangles) conditions (see also Table 1). Each rectangle represents a single trial within a block. The fMRI experiment only included the ConflictNoDropout blocks. The Online experiment included six different types of

blocks: ConflictNoDropout, ConflictMoneyDropout, ConflictShockDropout, NoConflictNoDropout, NoConflictMoneyDropout, and NoConflictShockDropout. The Dropout blocks always included 10 trials of NoDropout, in which both money and shock were presented in the outcome phase, followed either by 10 trials of MoneyDropout, in which money is removed from the outcome, or by 10 trials of ShockDropout, in which shock is removed from the outcome. An informative screen indicated participants which outcome will be removed. The money and face on top of each block indicates which outcome was most likely for that particular pair of symbols to occur. ×4 or ×6 number of block repetitions for each experiment. The Explicit report task was presented after the NoDropout blocks only in the Online experiment. See Supplementary Note §2 for an overview of the experimental procedure across studies. **d, e** Probability table associated with each symbol of a pair for the Conflict (**b**) and NoConflict (**c**) condition. The most likely outcome is specified in bold font.

**Table 1 | Participants and task-conditions overview**

|  |  | Online Exp. | fMRI Exp. |
|---|---|---|---|
| Participants | Total participants | 79 | 27 |
|  | Average Age ± SD | 25 y ± 7 | 37 y ± 17 |
|  | Females, males | 39,40 | 27,0 |
|  | Subj. with Considerate preference | 29 (37%) | 13 (48%) |
|  | Subj. with Lucrative preference | 24 (30%) | 3 (11%) |
|  | Subj. with Ambiguous preference | 26 (33%) | 11 (41%) |
| Conditions in the learning task | ConflictNoDropout | $4Bx10_{Conf}T/B$ | $6Bx10_{Conf}T/B$ |
|  | NoConflictNoDroput | $4Bx10_{NoConf}T/B$ | n.a. |
|  | Explicit learning task | 8Tx4 Questions | n.a. |
|  | ConflictMoneyDropout | $4Bx(10_{Conf}T/B+10_{Drop}T/B)$ | n.a. |
|  | ConflictShockDropout | $4Bx(10_{Conf}T/B+10_{Drop}T/B)$ | n.a. |
|  | NoConflictMoneyDropout | $4Bx(10_{NoConf}T/B+10_{Drop}T/B)$ | n.a. |
|  | NoConflictShockDropout | $4Bx(10_{NoConf}T/B+10_{Drop}T/B)$ | n.a. |
| Helping Task | Helping Blocks | n.a. | 30 T |

For each experiment, the table reports the total number of participants included in the behavioral analysis, and how many of those fell within the Considerate, Lucrative, and Ambiguous preference pattern. Additionally, the table reports the number of repetitions of each condition included in that particular experiment. *T* trial, *B* block, *Conf* Conflict, *NoConf* NoConflict, *Drop* Dropout. fMRI data only included the right handed participants, 25 (35 y ± 15 SD; 25 f), while behavioral data were collected for the 27 indicated in the table. Note: although age differed between the fMRI and Online experiment (two-tailed Mann–Whitney test, $W = 559.5$, $BF_{10} = 8.022$, $p < 0.001$), which neurocognitive model best described choices was unaffected by gender and age (Supplementary Note §11). For details on the Helping Task, see Supplementary Note §3. Source data are provided as a Source Data file.

of the time, and to lower intensity, non-painful shocks to the confederate with the same probability. We refer to this symbol as 'considerate'. At the beginning of each block, participants did not know the associations between symbols and outcomes. Choosing which symbol best satisfies the moral values that participants act upon in the task thus requires learning to predict the outcomes associated with each symbol. When a child's selfish actions cause pain to a sibling, parents intuitively resort to drawing the child's attention to the distressed facial expressions of the victim. To optimize our study to capture how such sights become a learning signal we (i) made shocks visible to the participants through pre-recorded videos showing facial expressions from the confederate, instead of the symbolic feedback more often used in neuroeconomic paradigms (Supplementary Note §1); and (ii) collected and analyzed neuroimaging data with a focus on how the brain updates values when learning from the facial reaction of others. Given the extensive literature on empathy for pain, we expect networks involved in processing the painful facial expressions of others[20–23], as captured by the affective vicarious brain signature (AVPS[24]), to have BOLD signals that covary with learning-relevant signals such as the prediction errors for shocks ($PE_S$). Given that the ventromedial prefrontal cortex (vmPFC) is known to have BOLD signals that covary positively with the current value of multiple outcomes, in particular for chosen options[25,26], and that outcomes for others appear to also be encoded in this region[27,28], we also expect the vmPFC to have such learning-relevant signals regarding shocks to others. Finally, given the involvement of financial rewards in our task, we expect the reward circuitry, as captured by a neural reward signature[29], to have signals that covary with prediction errors at least for money ($PE_M$). Whether signals in either network or region would be stronger in participants with a stronger preference for reducing shocks to others remains unexplored, and will be a key question.

In the Online experiment, participants performed a number of additional tasks to explore whether they learned the symbol-outcome association probabilities for both the self-money and other shocks, even if their preferences may prioritize one. First, we added blocks in which we removed the conflict. In this NoConflict condition (Fig. 1c), the symbol that led to high money in 80% of cases also led to low shocks in 80% of the cases. Second, after 1/3 of the Conflict and NoConflict blocks, we explicitly asked participants to report the learned associations. Only if participants learn the symbol-outcome association for the less preferred outcome, should they report different probabilities under the Conflict and NoConflict for this outcome.

Third, we also leveraged the devaluation approach pioneered in animals[16] by adding blocks (Dropout blocks, Fig. 1b, c) in which after 10 trials to learn the symbol-outcome associations, we informed participants that the self-money (MoneyDropout) or the other-shocks (ShockDropout) would not be delivered on the following 10 trials. We then examined the choice on the 11th trial, before participants witnessed the modified outcome. If participants track separable representations for self-money and other-shocks, we expect them to show different choices depending on which outcome is removed in ConflictDropout blocks, with choices expected to change substantially, if the outcome they weigh more is removed. Such changes should not occur in the NoConflictDropout blocks. If participants track a single, combined value for each option, based on the history of past experienced values, we expect them to continue choosing their previously favored option in either Conflict or NoConflict blocks. We then used Hierarchical Bayesian Model comparisons to test which computational formulations of the RLT models better describe participant's choices, and gain insights into how people combine the two outcomes in their morally relevant learning experience.

Using these approaches, here we show that participants vary substantially in whether they choose to maximize self-money or minimize other-shocks. Their choices are best described by a reinforcement learning model separately tracking the values of this self-money and other-shocks. Importantly, we find that individual differences are best captured by including an individual valuation parameter that biases expected values towards the outcome that bears more weight in the decision-making. Signals in the ventromedial prefrontal cortex reflect this bias, while the pain-observation network represents pain prediction errors independently of these individual preferences.

## Results
### Participants' choice preferences
Figure 2 shows participants' choices in the Conflict condition. As averaging the learning curves of participants showing opposite preferences would occlude learning, we split participant's choices in 'Considerate', 'Ambiguous', or 'Lucrative' preferences using a binomial distribution (see methods). While in the fMRI experiment the majority of participants had Considerate preferences, participants in the Online experiment were evenly subdivided across these three preference subgroups (Table 1). The groups with Lucrative and Considerate choice preferences show typical learning curves, starting at chance (50% preference for the Considerate option) on the first trial of each

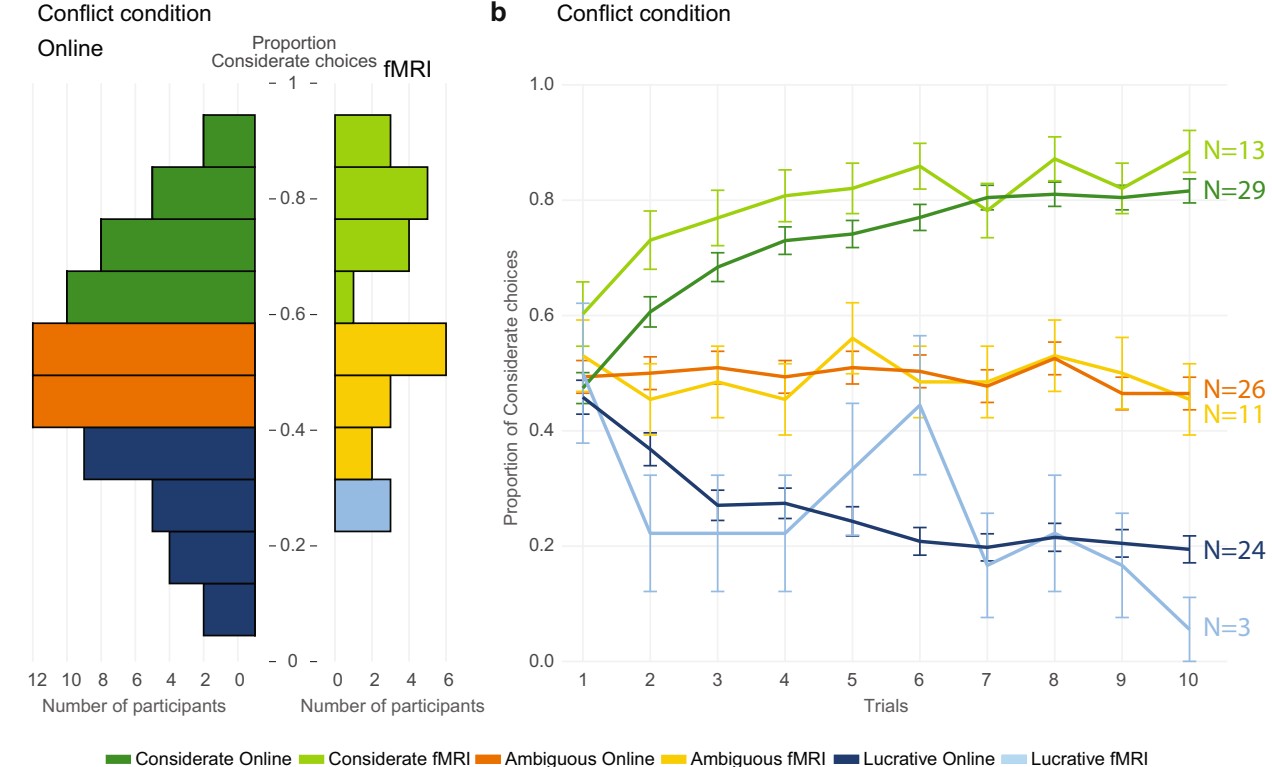

**Fig. 2 | Choice allocation in the Conflict condition. a** Histogram showing the proportion of considerate choices per participant over the first 10 trials of all available blocks of the Online (12 blocks of 10 trials: 4 blocks × 10 trials from the ConflictNoDropout; 4 blocks × 10 trials from the ConflictMoneyDropout; and 4 blocks × 10 trials from the ConflictShockDropout conditions) and fMRI (6 blocks × 10 trials of the ConflictNoDropout condition) experiment. Participants with above chance proportion of choices (Considerate preference) are shown in dark (Online) or light (fMRI) green; participant with below chance (Lucrative preference) are shown in dark (Online) or light (fMRI) blue, and participants within

chance level (Ambiguous preference) are shown in orange (Online) or yellow (fMRI). **b** Choices as a function of trial number of a block, averaged over blocks, separately for the three preference subgroups, with error bars representing the s.e.m. over participants. Color-code as in **a**. As in **a**, all first 10 trials of the Conflict conditions are included, independently of whether they belong to a Dropout or NoDropout block. For the stability of preferences across the different Conflict conditions see Supplementary Note §7. Source data are provided as a Source Data file.

block and then gravitating towards the Lucrative or Considerate option, respectively, with the last four trials showing relatively stable preferences ~80%, as would be expected for an RLT with 80/20 probability. The curve of the group with Ambiguous preference, consistently remains ~50% showing no clear learning curve (Supplementary Note §12). Finally, participant's average choices were not associated with how much participants believed the confederate to receive shocks (Supplementary Note §5), and matched participant's reported motivations (Supplementary Note §6).

**Symbol-outcome association reports biased by preferences**

To probe participants' explicit learning, in the Online experiment, we asked them to report how likely each symbol was associated with high-shock and high-money. Overall, participants tended to report higher probabilities for symbols with higher probability, also capturing the difference between the Conflict and NoConflict condition (comparing the *N* and U shapes in Fig. 3a, b). Because choices are thought to be driven by the difference in expected value across options, we summarized explicit reports as the difference in reported probability between the low-shock and high-shock symbol, with larger reported differences in the correct direction providing more evidence for (explicit) learning (Fig. 3c, d). We observed that the reported differences in probability across the symbols were different from zero, and in the correct direction for all subgroups and blocks (Supplementary Table 4). In particular, this shows that participants with Considerate and Lucrative preferences also learned about the association of the symbols with outcomes that drive their decisions less, and that the lack

of clear preferences amongst participants with Ambiguous preferences was not due to an utter lack of explicit symbol-outcome learning (but see Supplementary Note §12). We provide Bayes factors (BF, see Supplementary Tables 4 and 5 for *p* values), to quantify how much more likely the data is if an effect were present than if it were absent to infer, using traditional bounds, whether the data provides evidence for an effect ($BF_{10} > 3$) or for the absence of an effect ($BF_{10} < 1/3$), or remains inconclusive ($1/3 < BF_{10} < 3$, see[30] for a tutorial on how to interpret these tests).

Interestingly, the magnitude of explicitly reported probability difference was biased towards the outcome participants seem to weigh more strongly. Participants favoring the Considerate option show a stronger difference, that was also closer to the actual (80%–20% = 60%) difference, for shocks than money outcome probabilities, and those favoring the Lucrative option, for money than the shocks (difference between the filled green violins in Fig. 3c for the Considerate preference = 18.2 ± 4.45 s.e.m., *W* = 377.5, $BF_{10}$ = 577.22, *p* < 0.001 and between the filled blue violins for the lucrative = 9.2 ± 3.25 s.e.m., *W* = 67, $BF_{10}$ = 7.67, *p* < 0.018; Supplementary Note §9). Because the thresholds for grouping participants in preference groups is somewhat arbitrary, we also show that over all participants, the preference (i.e., proportion of considerate choices) was correlated with their bias (Fig. 3e, Pearson's *r* = 0.51; $t_{(77)}$ = 5.2, *p* = 1.6 × 10⁻⁶). Although Ambiguous participants explicitly learn the symbol-outcome associations, their accuracy tends to be lower than the accuracy of the Considerate and Lucrative groups for their outcome of value (Supplementary Note §12).

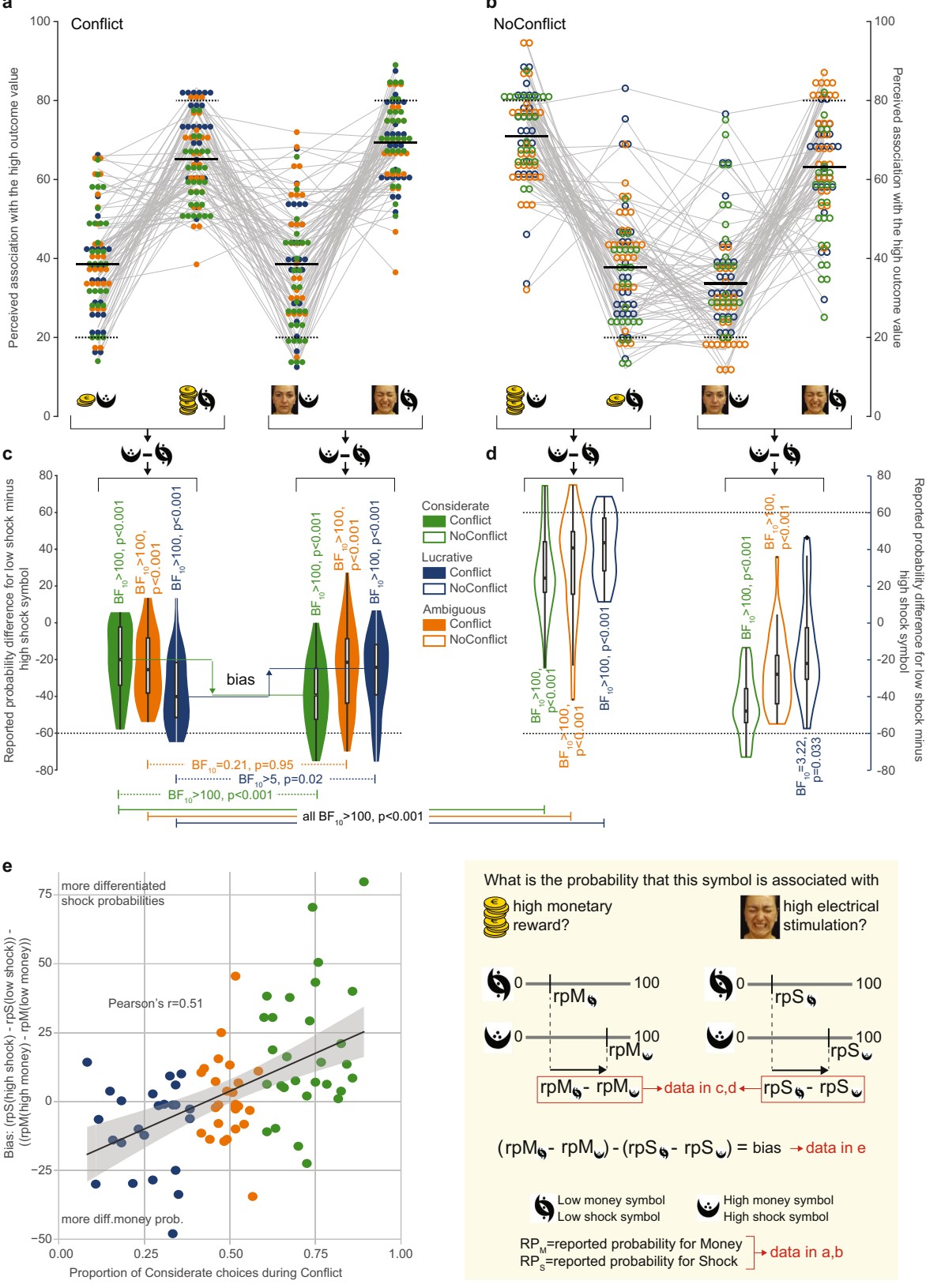

## Participants represent separable but biased outcome

Examining the choices on the critical 11th Dropout trial, suggests participants have separable representations of self-money and other-shock, and have also learned more about the outcome that drives their decisions than about the other outcome. We find that for the Considerate and Lucrative groups, which showed strong preferences on

the 10th trial, removing the quantity that guided their choice most, led to a highly significant change in their choice allocation (Fig. 4), while removing the quantity that guided their choices less, left their choices unchanged.

Dropout trials in NoConflict blocks did not lead to comparable changes in choices (Supplementary Fig. 6), confirming that the change

**Fig. 3 | Participants' report bias. a** Participant reports of perceived high-outcome probability for each symbol in the Conflict condition. **b** Same for the NoConflict conditions. The *x* axis specifies which symbol and the probed question: short/tall pile of money = reported high-money probability for the symbol actually associated with low/high-money; painless/painful face reported high-shock probability for the symbol associated with low/high-shock. As a group, most participants correctly assigned higher probabilities to symbols that had higher probability, and reported a different pattern of probabilities after Conflict and NoConflict blocks. Thick black lines = average of reported probabilities; dotted black lines = programmed/expected probability. Square brackets below the graphs in **a** and **b** indicate the direction of the difference computed in **c**, **d**. **c** Difference between reported probabilities for the low-shock minus high-shock symbol, separately for the Considerate (green), Lucrative (blue) and Ambiguous (orange) preference group, and for the Conflict condition. The yellow inlet in the bottom right corner illustrates how the differences in **c**–**e** are calculated. Violin plots represent the value distribution, the boxplot within, the median and quartiles, the whiskers, the range of datapoints between Q1–1.5IQR and Q3 + 1.5IQR. **d** Same for the NoConflict condition. The BF10

and *p* values above a violin represent the result of a two-tailed Wilcoxon signed-rank test of the differences vs. zero. The BF and p values on the dotted green and blue lines indicate the result of a two-tailed Wilcoxon signed rank test comparing the difference for money and shock. The BF and p values below a pair of violins represent the result of a two-tailed Wilcoxon signed rank test comparing the Conflict and NoConflict conditions. All statistical values are presented in Supplementary Tables 4, 5. Dotted black lines = actual probability difference (80%−20% = 60%) as programmed in the task. **e** Correlation between the participant's report bias and the proportion of considerate choices during the Conflict task, with Bias = (rpS(high-shock)−rpS(low-shock))−((rpM(high-money)−rpM(low-money)), where rpS/rpM stand for reported probability of high-Shock/high-Money outcome, and high-shock, low-shock, high-money, low-money refer to the symbols based on their actually most likely outcome. Black line and gray shading represent the regression line and the 95% confidence interval. *N* = 79 (29 Considerate, 24 Lucrative, 26 Ambiguous) for all panels. Source data are provided as a Source Data file.

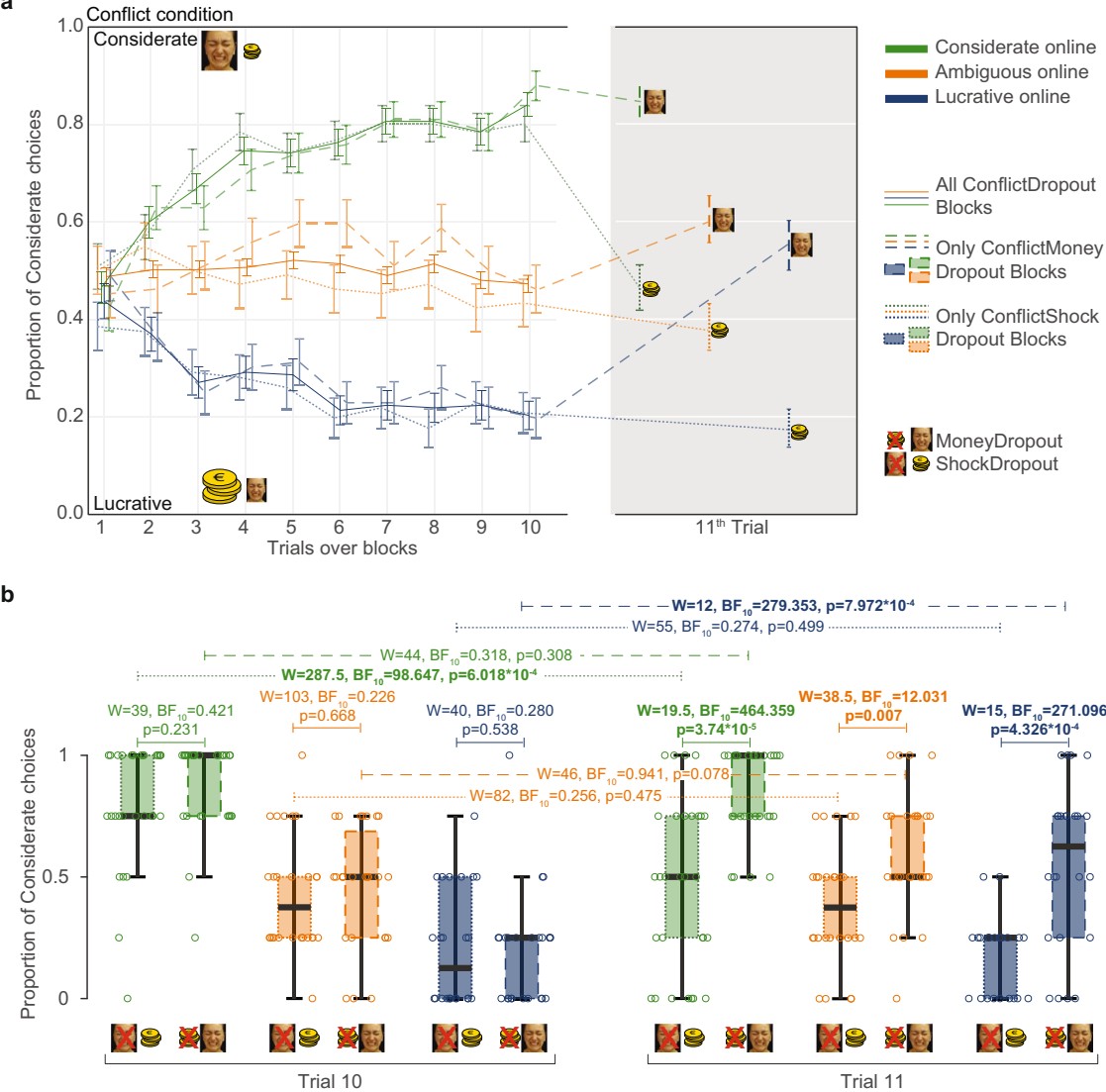

**Fig. 4 | Participant's choices at the 11th trial. a** Choices in the ConflictDropout blocks averaged separately for the preference subgroups, as classified using all Conflict trials. For each subgroup, the solid line represents the average of all trials, the dashed line, blocks in which money will be withheld, and the dotted lines, blocks in which shocks will be withheld. Standard errors of the mean across participants are indicated for each trial. **b** Boxplot of choices across participants at the 10th and 11th trial of ConflictDropout blocks as a function of group and outcome

withheld. Dots are participants' proportions of considerate choices, thick center black lines represent the median, box the upper and lower quartiles; whiskers, the range of datapoints between Q1−1.5IQR and Q3 + 1.5IQR. Wilcoxon signed-rank two-tail test were used to test the differences and statistical values are indicated. Bold text highlights significant differences. *N* = 79 (29 Considerate, 24 Lucrative, 26 Ambiguous) for all panels. Source data are provided as a Source Data file.

in choices in the Conflict condition are not the result of a simple rule 'reset choice when my preferred outcome is removed', but considers the probability of outcomes associated with the remaining outcome, which are the only difference between Conflict and NoConflict blocks.

Interestingly, removing the guiding outcome in Conflict trials did not lead to a mirror-symmetrical (i.e., ~80%) preference for the other symbol: the average choice allocation switched to just below, or above, ~50%, on the 11th trial (Fig. 4a). That the first choice after Dropout is not as polarized as the 10th trial echoes the bias we observed in the explicit reports (Fig. 3b): with the now less polarized choices mirroring the lesser differences in outcome probabilities for the remaining outcome. For the Ambiguous group, there was no robust evidence that removing either quantities lead to a change of preference towards the better option for the remaining outcome. It is important to emphasize that the choice on trial 11 occurs before the participant is presented with the single remaining outcome, and thus probes choices exclusively based on expected values that were learned during conflict trials. Supplementary Fig. 6 also shows choices on trials 12–20, but as these trials contain one rather than two potentially conflicting outcomes, we do not expect our models to adequately represent participants' reasoning and learning over those subsequence trials.

In summary, choices reveal substantial individual variability in preference, but most participants show evidence of some form of learning (See Supplementary Note §12 for learning in participants with Ambiguous preferences). Together this supports the notion that choices in our conflictual conditions are dominated by a form of learning in which participants represent separable outcomes for self-money and other-shocks, with the individual preferences influencing the magnitude of their respective symbol-outcome association.

## M2Out predicts participant's choices more accurately

To gain further insights into how people combine shock and money outcomes in their morally relevant learning experience, we defined different formulations of RLT models (Fig. 5a, b) and used the hierarchical Bayesian framework to test which alternative better describes participant's choices. Except in our random-choice model (M0), in all the learning models we compare, shock and money are additively combined using an individual weighting factor $wf$ ranging from 0 to 1. This $wf$ captures the value of the monetary outcome for self relative to the value of the shock to the other, and is not unlike the salience $\alpha$ in the Rescorla-Wagner Learning Rule[31,32] or the harm aversion parameter $\kappa$ in the decision model of Crockett[15,33]. A $wf$ closer to zero would characterize a participant preferring to minimize the harm to the other person, a $wf$ closer to one corresponds to more lucrative choices. In particular, we investigate whether choices are better described (i) by an RLT model that combines money and shock as soon as outcomes are revealed (M1) or (ii) models that keep separate representations for the two types of outcomes (M2). M1 instantiates a learning in which participant's decisions are based on the history of past reward values, without representation of the nature of outcomes, while M2 instantiates learning with separable representations of the expected outcomes for self and others. For M2, we further compared a variant that scales outcomes based on personal preferences for money or shock (M2Out) vs. a variant that tracks expectations independently of personal preferences, but introduces weights at the decision-phase (M2Dec).

We fitted our competing models on the first 10 trials of the ConflictDropout blocks and then examined (i) how well they predicted those first 10 choices using an approximation of the leave-one-out information criterium (LOOIC[34]) and (ii) how well they predicted choices on the 11th trial, when one of the outcomes is removed. Because the 11th trial was not included in the fit, we can directly quantify the likelihood of the competing models given the observed choices on the 11th trial to assess which best predicts behavior under devaluation. A priori, we expect all but M0 to perform comparably well during the conflict trials. This is because due to the distributivity of

multiplication over addition, the accumulated utility over trials that is fed to the softmax is the same for M2Out ($\Sigma[wf \cdot LR_M PE_M] + \Sigma[(1-wf) \cdot LR_S PE_S]$) or M2Dec ($wf \cdot \Sigma[LR_M PE_M] + (1-wf) \cdot \Sigma[LR_S PE_S]$), and virtually identical to that for M1 ($\Sigma[LR \cdot (wf \cdot PE_M + (1-wf) \cdot PE_S)]$), except for the use of a single $LR$. However, we expect them to make different predictions on the 11th trial. M0, by design, continues to predict random choices. Because M1 does not have separable representations of money and shock, its decisions for the 11th trial can only be based on the composite expected value and it will predict that participants continue to choose their previously preferred option. Finally, because the M2 models have separate EV for money and shock, we programed the model to transform the information that participants receive before they perform the 11th choice (i.e., one outcome will no longer be delivered), into a revised decision criterion based on the remaining expected value only, without using a $wf$ in their choice, as there is no longer a conflict to resolve (Fig. 5b). We thus expect choices not to change much if the less preferred outcome is removed, and to change significantly, if the preferred outcome is removed.

Figure 5c compares the predictions of the three learning models (M1, M2Out, M2Dec) with the actual choices for the critical Dropout blocks. During the initial 10 trials in which the conflict is present, all three learning models capture the general shape of the learning curve, and can accommodate individual differences in preference through $wf$, generating increasingly more Considerate choices as the trial number increases for participants with Considerate preferences, and fewer Considerate choices for participants with Lucrative preference. Including all participants, over the first 10 trials, the LOOIC confirms that M1, M2Out, and M2Dec perform similarly well, i.e., remain within a standard error of one-another, and perform better than the random-choice model M0 (Fig. 5d; and Supplementary Fig. 7a, b for results that exclude the Ambiguous and non-believers groups). A similar pattern is true for the choices in the fMRI experiment (Supplementary Fig. 7c, d). Next, we used the 11th trial to arbitrate across M1, M2Out, and M2Dec. When money is removed and only shocks remain (Fig. 5e, gray background), all learning models correctly predict that Considerate preference participants do not change their choices, but only M2Out (purple and black arrow-head) accurately predicts that participants with Lucrative preference change their preference to just above 50%. In contrast, M1 (gray) fails to predict any change for those lucrative participants, and M2Dec (light blue) overestimates their change. When shocks are removed and only money remains (Fig. 5e, yellow background), only M2Out correctly predicts the magnitude of change in the Considerate preference participants. When considering all participants, M2Out consistently predicts the choices on the 11th trial better than the other models at the group level (Fig. 5d, f) and in the majority of participants (Supplementary Fig. 7e). Importantly, that the distribution of likelihood of the observed data given M2Out (indigo, Fig. 5f) does not overlap with that given M2Dec (turquoise) or M1 (gray) shows that our approach focusing on the 11th trial in four blocks is sufficiently powered to adjudicate between our candidate models: a given choice pattern is uniquely more likely under one model than the others.

That M2Out outperforms M2Dec in particular, is because M2Out scales expected values based on $wf$ so that for a considerate participant with $wf \approx 0$, $Out_M$ are multiplied with $wf \approx 0$ to calculate $PE_M$ (Fig. 5a, b), and $EV_M$ remains close to zero for both symbols, so that when shocks are removed, choices are between two symbols with $EV_M \approx 0$, predicting the ~50% preferences observed. In contrast, for M2Dec, such scaling of EV by $wf$ does not occur, and on the 11th trial, participants have EV values for their less-favored outcome that are as differentiated as those of the favored outcome. Note that a version of M2Dec that would maintain $wf$ in the decision-phase of the 11th trial, would make predictions extremely similar to M2Out. Finally, we also considered a model in which the $wf$ is influencing both the decision and outcome phase (M2DO; Supplementary Fig. 8). This model

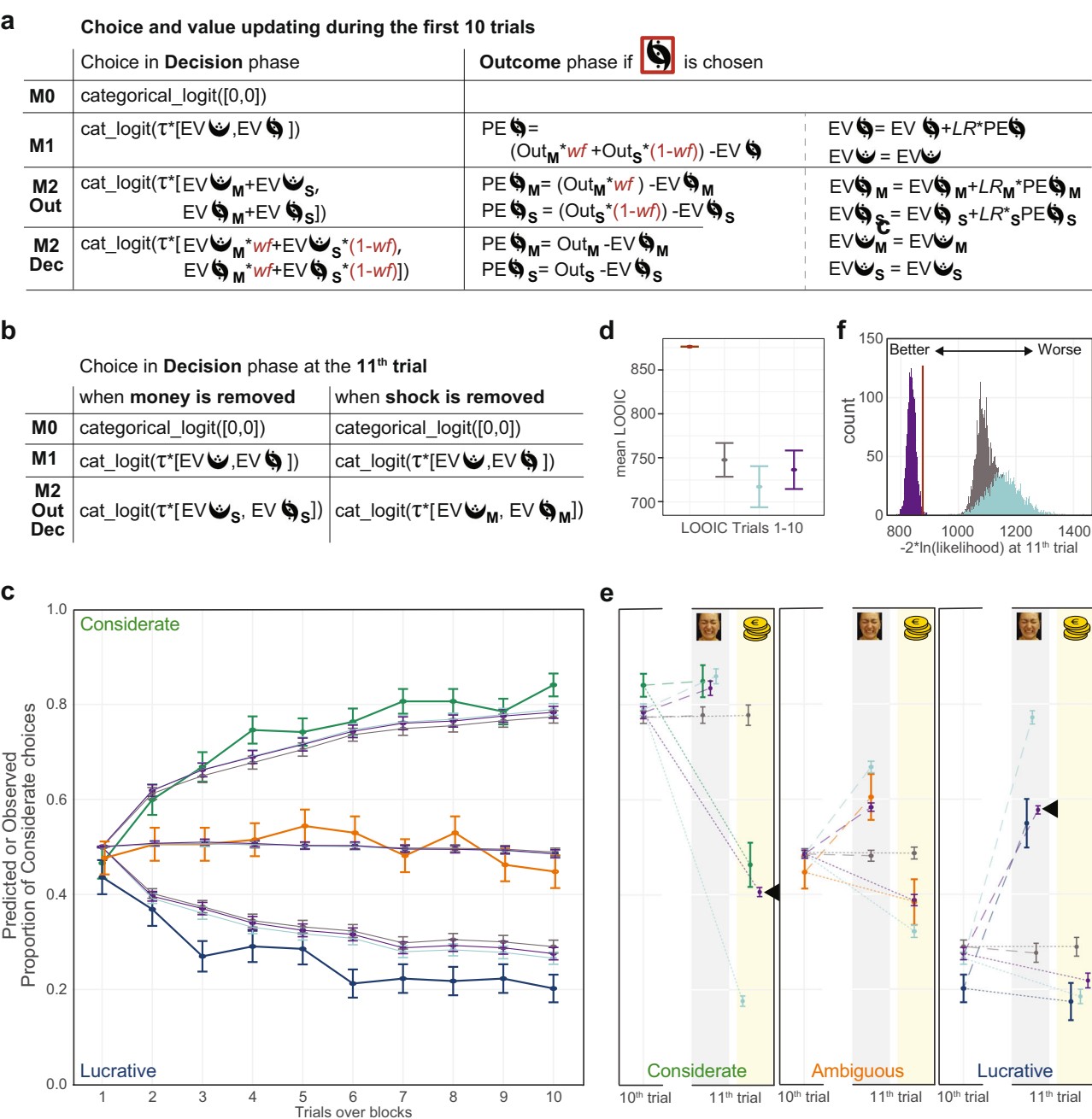

**Fig. 5 | Model comparison. a** Model formalization for the chosen (red contour) option. In all models, decisions are based on a categorical logit function (aka softmax), contrasting the expected value across the alternatives with inverse temperature $\tau \in [0,5]$ and learning rate $LR \in [0,1]$. EV = Expected Value, PE = Prediction Error, Out = Outcome. Subscript M = money and S = shocks. Outcomes are coded by value: high-shock $Out_S = -1$, low-shock $Out_S = +1$, high-Money $Out_M = +1$, low-money $Out_M = -1$. If outcome is withheld during Dropout, $Out_S$ or $Out_M = 0$. Outcomes are set to the same values (+1 and −1) for money and shock because of the optimization task (Supplementary Note §4). **b** Decision Model at the 11th trial for the Dropout condition. **c** Choices in the first 10 trials of the Con-flictDropout blocks (Online experiment) as a function of preference together with the predictions by M1, M2Dec, and M2Out. M0 always predicts 0.5 and is not shown. Choices and predictions averaged over all blocks, error bars: s.e.m. across participants ($N = 29$ Considerate, $N = 24$ Lucrative, $N = 26$ Ambiguous). **d** Leave-one-out information criterion (LOOIC34) of the models over the first 10 trials. Error bars: standard error of the estimation LOOICerror. M0 performs poorly, the three learning models perform similarly. The information criterion (IC) scale captures how much information is lost when comparing model predictions with actual choices, and smaller IC thus characterize models that better describe behavior. $N = 79$. **e** Change of participant choices ($N = 29$ Considerate, $N = 24$ Lucrative, $N = 26$ Ambiguous) and model predictions for the 10th to the 11th trial. Dashed lines connect 10th to 11th trials when money is removed (gray background), dotted lines, when shocks are removed (yellow background). 11th trial not included in model fitting. M2Out (black arrowheads) makes the best predictions for the 11th trial. Error bars: standard error of the mean across participants. **f** Distribution over 4000 posterior draws of the summed log-likelihood of the 11th trial over all participants multiplied by −2 to place values on the information criterion scale as for LOOIC. M2Out outperformed all other models. We use LOOIC in **d**, because these first 10 trials were included in the fitting of the model, but log-likelihood in **f** because it is not included in the model fit. Source data are provided as a Source Data file.

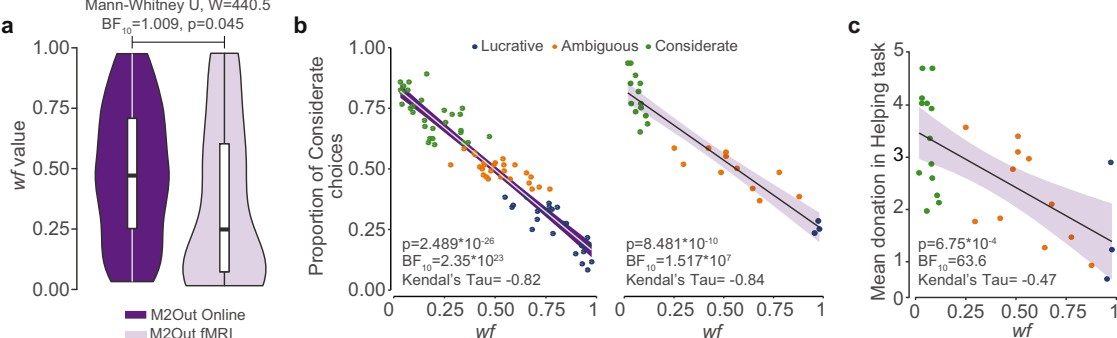

**Fig. 6 | *Wf* Predictive power. a** Distribution of *wf* parameter estimates for M2Out in the Online (purple, only including the first 10 trials of the ConflictDropout blocks) or fMRI (lavender, including all trials) across participants. The whisker plots represent the quartiles, dots outliers. The distribution of the other parameters of M2Out can be found in Supplementary Note §13. Distributions were compared with a two-tail Mann–Whitney *U* test. **b** Relation between *wf* and proportion of Considerate choices in the online (left) and fMRI (right) experiment. For the online experiment only the 10 first trials of Dropout blocks are considered in the model fit. The line and shading represent the regression line and its standard error, *p* and BF values are for a two-tailed test on Kendall's Tau value. **c** Relation between *wf* in fMRI experiment and donation in the helping experiment. *p* and BF values are for a two-tailed test on Kendall's Tau value.

outperformed M2Dec, but not M2Out, with M2Out still predicting choices in critical conditions (MoneyDropout for Lucrative and ShockDropout for Considerate participants) closest to the actual choices of our participants. For parsimony, we, therefore, use M2Out for further analyses.

## *Wf* is accurately recovered and has external validity

Fitting our winning model M2Out on the first 10 trials of the Dropout blocks leads to a wide range of *wf* values (Fig. 6a) across participants that correlates highly with the proportion of considerate choices (Fig. 6b). Parameter recovery further shows that if one simulates participants with different *wf*, M2Out accurately recovered much of that variance ($r(wf_{simulated}, wf_{estimated}) = 0.69$, $p < 10^{-6}$, $BF_{10} > 10^6$ Supplementary Note §13). For our fMRI participants, we also performed a Helping task not involving learning that we had previously developed[4] (Supplementary Note §3). In each Helping trial, participants viewed a victim receive a painful stimulation, and could decide to donate money. If no money was donated, the victim received a second stimulation of equal intensity. If some money was donated, each € donated reduced the second stimulation by one point on the 10 point pain scale. We found that the *wf* value estimated in our learning task in the fMRI experiment was significantly correlated with how much money participants on average donated to reduce the victim's pain in this Helping task (Kendall $Tau_{(wf,choice)} = -0.47$, $BF_{10} = 76$, $p < 0.001$, Fig. 6c), providing evidence that *wf* captures a property of the participant that generalizes to other situations in which participants need to resolve a conflict between self-money and other-shocks. A Bayesian multiple linear regression including *wf* together with more traditional trait questionnaires (four subscales of the IRI[26] as a traditional measure of empathy and MA[27] as a traditional measure of the attitude towards money), found strong evidence that including *wf* improves the predictive power ($BF_{incl} = 11.46$, $p = 0.009$, Supplementary Table 6), while none of these questionnaires explained variance in the Helping Task (all $BF_{incl} < 0.7$, all $p > 0.09$ Supplementary Table 6). This establishes the external validity of our learning task as a predictor of costly helping in a different task.

## Neural correlates of value updating

Participants showed a wide network of brain activity when outcomes were revealed (Fig. 7a).

To examine the contribution of pain-witnessing and reward processing networks in prediction error coding during the outcome phase, we extracted the *wf*-normalized $PE_S$ and $PE_M$ parameter estimate image ($\beta_{PES}$ and $\beta_{PEM}$) from each participant (Supplementary Note §15), and dot-multiplied them with the affective vicarious pain

signature (AVPS[24]) and a newly developed reward signature (RS[29]). We used neural signatures because they provide a principled method to reduce the involvement of a distributed neurocognitive systems to a univariate measure that can then be analyzed using Bayes factor hypothesis testing to provide evidence both for or against the involvement of this system[30,35–37]. We found $PE_S$ but not $PE_M$ to load significantly on the AVPS (Fig. 8a, e; Supplementary Table 9). The $\beta_{PES}$ loading on AVPS is negative, because the AVPS was developed to provide larger values when viewing painful compared to non-painful facial expressions[24], while we coded shock outcomes in terms of their value, i.e., a non-painful shock had value +1 and a painful shock of −1. The strength of the loading did not depend on *wf* (evidence of absence, Fig. 8b; Supplementary Table 9). In our paradigm, this pain-witnessing network thus carries signals that negatively covary with prediction errors for shocks that do not depend on personal preferences (*wf*). With regard to the reward signature, we found $PE_S$ and $PE_M$ to load positively on RS (Fig. 8c, f; Supplementary Table 9) showing that receiving less intense than expected shocks or higher than expected monetary rewards both triggered a pattern of activity typical of receiving a reward. Examining whether the loading on RS depended on *wf* led to inconclusive evidence leaning in favor of absence (Fig. 8d; Supplementary Table 9).

Next, we performed a more explorative voxelwise linear regression that predicts the parameter estimate of the $PE_S$ modulator using a constant and *wf*. As expected for a region involved in valuation, we found the vmPFC to have signals covarying positively with $PE_S$ (i.e., higher signals when shocks were lower than expected) with a more ventral cluster associated with $PE_S$ in a way that depended on *wf* (Fig. 7b, red), while a more dorsal vmPFC cluster showed an association with $PE_S$ after removing variance explained by *wf* (Fig. 7b, yellow). In addition, we find that the left somato-motor cortex, including BA4 and 3, also harbored signals covarying positively with $PE_S$ in ways that depended on *wf*. The same voxelwise analysis applied to $PE_M$ only generates significant results when the cluster-cutting threshold was reduced to $p_{unc} < 0.01$ (Fig. 7c), and reveals striatal and ventral prefrontal clusters in line with those described in the literature for $PE_M > 0$ after removing the variance explained by *wf*[26,38–40], and clusters in the right cerebellum, ventral temporal lobe and hippocampus with $PE_M$ signals that depend on *wf* (Fig. 7d). Comparing the networks associated with $PE_S$ and $PE_M$ reveals that these networks are largely non-overlapping (Fig. 7c). However, inclusively masking the contrast $PE_M > 0$ with the contrast $PE_S > 0$ reveals five clusters of overlap, the largest of which was in the vmPFC (Supplementary Fig. 17), but these clusters remained too small to survive a whole brain FWE cluster-size correction.

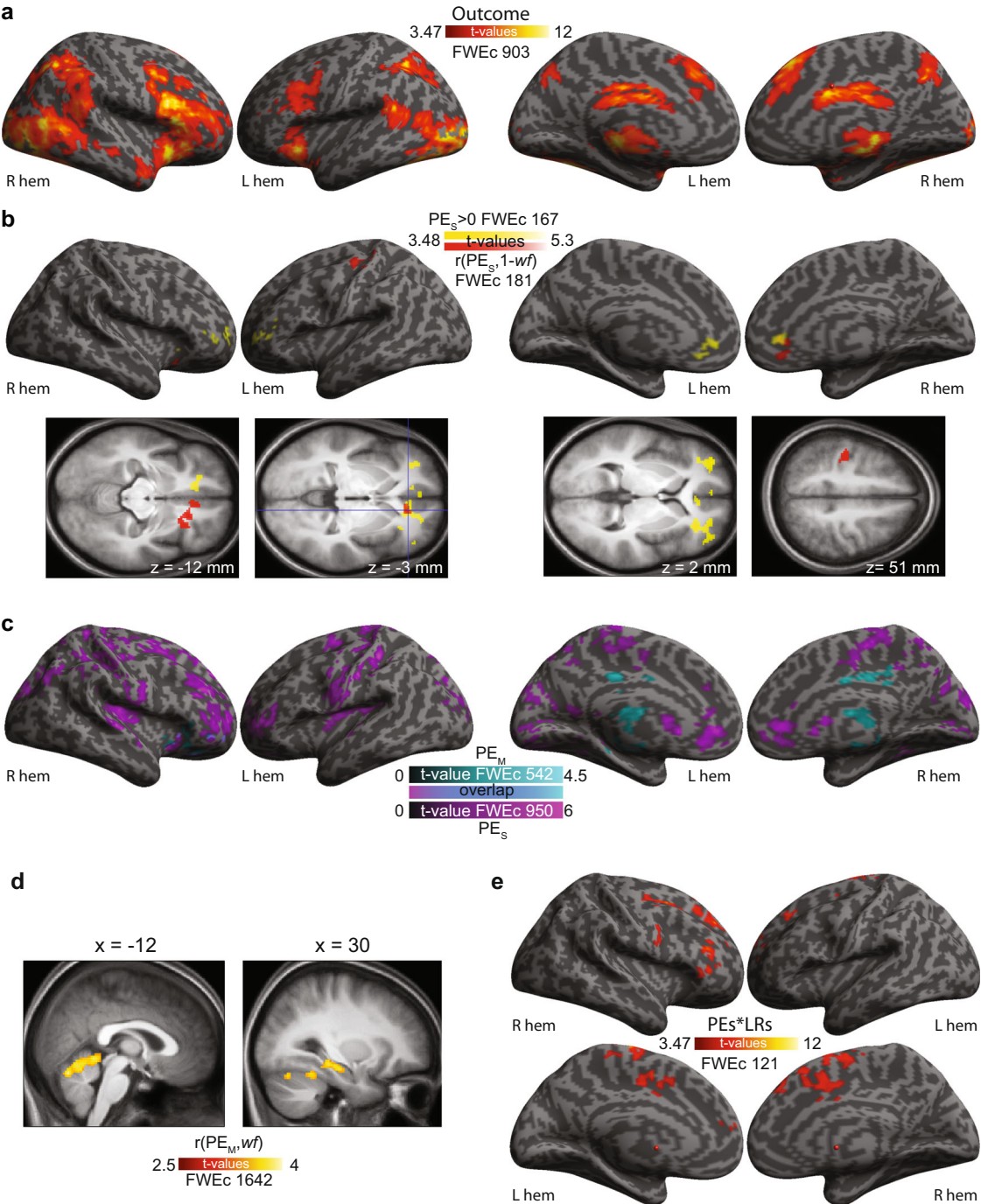

**Fig. 7 | Key fMRI results. a** Voxels where BOLD signals are increased during the outcome phase, independently of prediction errors, (Outcomes>0), $p_{unc} < 0.001$, $k$ = FWEc = 903 voxels (Supplementary Table 8). **b** Red: regression between $PE_S$ and 1-*wf* ($p_{unc} < 0.001$, $k$ = FWE $c$ = 181 voxels, Supplementary Table 10). This identifies voxels with signals that increase more for shock outcomes that are less intense than expected (i.e., positive $PE_S$) in participants that place more weights on shocks (high 1-*wf* values). Yellow: results of the contrast constant > 0 in the same linear regression with (1-*wf*) ($p_{unc} < 0.001$, $k$ = FWEc = 167 voxels). This identifies voxels with signals that increase with increasing $PE_S$ after the variance explained by (1-*wf*) is removed (Supplementary Table 11). **c** Cyan: $PE_M > 0$ (after removing variance explained by *wf*; Supplementary Table 12) at $p_{unc} < 0.01$, $k$ = FWEc = 542 voxels. Purple: $PE_S > 0$ presented the same cluster-cutting threshold of $p_{unc} < 0.01$, $k$ =

FWEc = 950 voxels (Supplementary Table 11) in order to visualize that the two networks are largely distinct (but see Supplementary Fig. 17 for overlap). **d** Correlation between $PE_M$ and *wf*, $p_{unc} < 0.01$, $k$ = FWEc = 1642 voxels (Supplementary Table 13). **e** Results of the contrast $PE_S*LR > 0$, $p_{unc} < 0.001$, $k$ = FWEc=121 voxels (Supplementary Table 14), which represent the BOLD signal associated with shock value updating. Note that across all panels, results are FWE cluster corrected at α < 0.05 using (i) cluster-cutting at $p_{unc} < 0.001$ (**a, b, e**) or (ii) cluster-cutting at $p_{unc} < 0.01$ (**c, d**) then excluding clusters with an extent below the critical FWE cluster-size FWEc. Cluster-extent threshold size always indicated in figure panels. Renders were created in SPM12, based on the cortex_20484 surface from the SPM12 templates; slices are taken from the average T1 anatomical scan from our participants and visualized with SPM12.

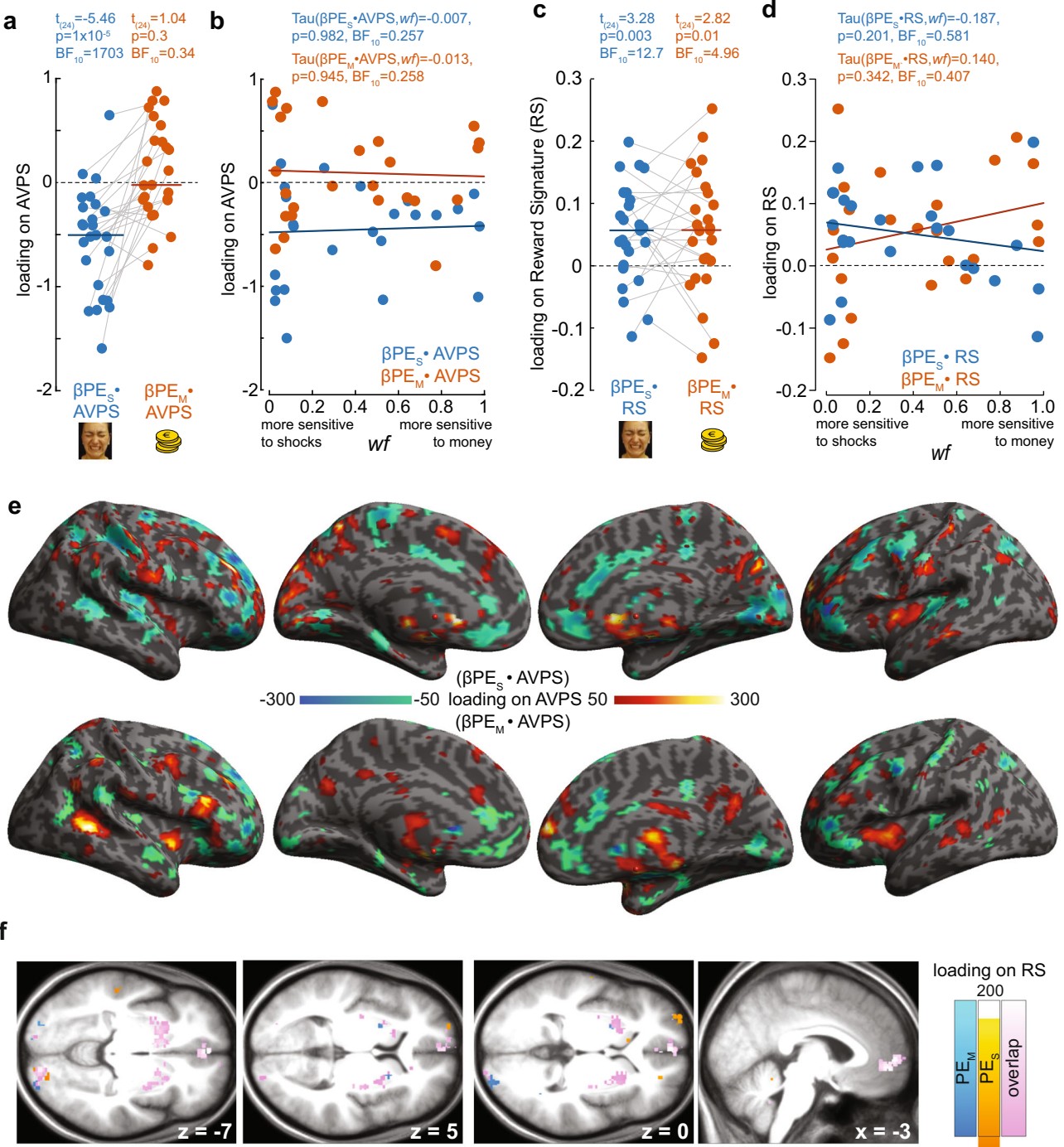

**Fig. 8 | Loading on AVPS and RS. a** Loading of the normalized parameter estimate images ($\beta_{PE}$) for $PE_S$ (blue) and $PE_M$ (orange) on the AVPS[24]. Each dot is one participant, the horizontal line is the mean. $p$ and BF values reflect a one-sample parametric (because normality was not violated) $t$ test against zero in JASP. **b** Loading of the parameter estimate for $PE_S$ ($\beta_{PES}$) (blue) and $PE_M$ ($\beta_{PEM}$) (orange) onto the AVPS as a function of $wf$. $p$ and BF values represent the outcomes of a parametric correlation analysis using Pearson's $r$. **c**, **d** Same as in **a**, **b**, but for the RS. It should be noted, that with our sample size ($n = 25$), we are adequately powered to detect strong associations with $wf$ ($\rho = 0.5$, $\alpha = 0.05$, power = 87%) but cannot exclude weaker associations for which we were underpowered ($\rho = 0.3$, $\alpha = 0.05$, power = 45%), as 80% power for such associations would require over 60 participants.

**e** Voxels contributing substantially to the loading of $\beta_{PES}$ (top) and $\beta_{PEM}$ (bottom) onto AVPS. Because the overall loading is calculated by first multiplying each voxel's $\beta_{PES}$ value with the value in this voxel of AVPS, and then summing this multiplication over all voxels, here we simply calculated the first step (the voxelwise multiplication of $\beta_{PES}$ and AVPS, and $\beta_{PEM}$ and AVPS), and averaged the result across our participants. To not overcrowd the image, we only show voxels with values above 50 or below −50 after multiplying the images by $10^6$. **f** Same as in **e**, but for the RS. Renders were created in SPM12, based on the cortex_20484 surface from the SPM12 templates; slices are taken from the average T1 anatomical scan from our participants and visualized with SPM12. Source data for **a**–**d** are provided as a Source Data file.

Using the method developed by Zhang and colleagues[41] suggests that the AVPS, the RS, and the ventral prefrontal network correlating with $PE_S$ during the outcome phase do indeed represent $PE_S$ rather than only encoding the raw witnessed outcomes (Supplementary Note §18).

In reinforcement learning theory, value updating is based on prediction errors and learning rates, with a participant updating expected values by $PE_S \times LR_S$, and $PE_M \times LR_M$, respectively. Accordingly, we also examined where in the brain signals covary with $PE_S$ in ways that depend on $LR_S$ and with $PE_M$ in ways that depend on $LR_M$, to identify signals related to value updating. The former revealed a robust network of prefrontal and in particular, medial prefrontal regions (Fig. 7e), while the latter failed to reveal any signals surviving corrections at either cluster-cutting thresholds ($p < 0.001$ or $p < 0.01$, $k$ = FWEc).

Our learning paradigm is optimized to capture the neural processes involved in learning symbol-outcome associations, i.e., processing the outcomes and computing prediction errors, and these processes can be assumed to occur when the outcomes are presented. In contrast, as the options amongst which to choose remain constant across the trials of a block, participants need not wait for the decision-screen to choose which symbol to choose on the next trial (Fig. 1a). Attempts to isolate brain activity related to this uncertainly timed decision-process are therefore inefficient in our paradigm and are only included in Supplementary Fig. 18. Choice paradigms in which options vary from trial to trial and are only revealed once a decision can be made are a more powerful means to isolate choice-related brain activity see for example, [3,4,33].

## Discussion

Our results indicate that participants' choices appear to be dominated by separable but biased representations of self-money and other-shocks. The availability of separable representations was apparent (i) from the ability to provide above-chance explicit reports for both outcomes that also captured the difference between Conflict and NoConflict trials and (ii) from the reaction to devaluation by not altering choices if the preferred outcome was preserved, but by switching preference when the preferred outcome was removed in the ConflictDropout trials. That representations were biased was borne out from the fact that (i) individuals had more differentiated reports for their favored outcome-type (ii) that preferences after devaluation were asymmetric, with preference levels depending on the weight of the remaining quantity.

Bayesian model-comparison formalizes this conclusion (see Supplementary Note §19 for a detailed discussion of the predictions and rationale of each model). When one outcome was devalued, M2Out outperformed M2Dec and M1 and was the only model that captured the reduced preference, when the most valued outcome was removed. We also briefly explored a more general model (M2DO), that uses a parameter α to distribute the effect of preference across the outcome and decision phase. Fixing α to 1 or 0 transforms M2DO into M2Out or M2Dec, respectively, and fixing it to 0.5e are a more powerful means to half-way between M2Out or M2Dece are a more powerful means to outperform M2Dec but not M2Out, and we, therefore, opted to use the simpler M2Out for the rest of our analyses. In the future, fitting α to each participant's specific devaluation behavior may enable the use α to quantify how much a given participant's preference pervades their expected value representations under conflict.

Some have found that learners that respond flexibly to devaluation, as most of ours did, which indicates goal-directed behavior instead of habit, are more likely to have used a model-based form of learning[16,42] Others found participants' choices to be dominated by model-free learning when preventing shocks to others in two-choice Markov decision tasks[15]. It may thus be that participants represent the nature of outcomes (as revealed by our devaluation study) while avoiding the cognitively expensive decision-tree necessary to ascribe rewards following a rare transition in a two-choice Markov decision task to an alternative (non-selected) action.

Participants varied substantially in their preferences under conflict. In our models, we find that the weighting factor *wf* is an effective way to capture these individual differences. Importantly, *wf* had external validity: it predicted how much money the participant gave to reduce shocks to the same confederate in a different task not requiring learning[4] and did no better than trait measures of empathy and money attitude. If state empathy is regulated by motives and context (see[18,19] for reviews), such as our financial incentives to downregulate empathy, the IRI, which attempts to find a stable, context-independent measure of empathy, may have less explanatory power than a *wf* estimated during a similar moral conflict.

In our Online study, equal-sized groups showed significantly considerate, lucrative, and ambiguous preferences. In our fMRI study, most participants (13/27) showed a significant preference for the considerate option, with only 3/27 showing a significant preference for the lucrative option. Why this distribution differed from the Online experiment is unclear, but two factors may have played a role. First, in the fMRI experiment, conducted on site, participants were under the impression that all shocks were delivered in real-time to the other person; while in the Online experiment, only a subset would later be delivered, reducing the value of considerate choices. Second, the vicinity of the experimenter in the fMRI experiment may have increased the motivation to act in a socially desirable way. Finally, age and sex distributions differed across the two studies (Table 1) but are unlikely to explain this difference because model selection did not depend on gender, and no significant correlation was observed between age and preference (Supplementary Note §11).

Our fMRI data further characterizes how preferences bias learning, by showing that earlier stages of processing such as the network associated with pain-observation[20,22–24], as quantified using the AVPS[24] have signals associated with $PE_S$ that do not depend on *wf*, while later stages of processing associated with valuation, particularly in the vmPFC[14,25–28] have $PE_S$ signals that depend on *wf*. That the ventral but not the dorsal vmPFC had valuation signals that depend on *wf* dovetails with a finding of Nicolle et al.[43] that if participants have to switch from trial to trial between focusing on making a monetary decision according to their own values or those of someone else, more ventral mPFC voxels had valuation signals corresponding to the values of the current focus, while more dorsal clusters had valuation signals corresponding to those of the alternative focus.

The AVPS has been traditionally characterized in tasks in which participants are unable to form predictions, as high- and low-pain trials are randomized[20,22–24], making it impossible to distinguish whether it purely represents outcomes or prediction errors. Our learning paradigm refines our understanding of this network by showing that it receives sufficient information about expectations to transform visual inputs into the prediction error signals thought critical for learning (Supplementary Note §18). Theories that emphasize the motivational modulation of empathy predict that to maximize financial gains, less considerate participants suppress empathy-related processes often ascribed to this network[18]. That the AVPS network was not measurably modulated by *wf, therefore,* invites us to reconsider where in the brain such empathy modulation occurs, with ventral prefrontal valuation regions revealed as a potential location in our data.

Our study has several additional limitations. First, we limited our model comparison to a number of hypotheses driven by RLT models. We did not test ratio or logarithmic ratio models of valuational representation in this study. These valuational structures are known to occur but are less often indicated in modeling gains and losses to the self[17]. Future experiments could be optimized to explore whether such alternative ways to combine these values may be more appropriate under certain moral conflicts. Second, our participants, or at least

some of them, may not use a RLT model at all. Instead, they may use rules such as 'choose a symbol randomly, and only switch if you encounter × unfavorable outcomes in a row'. Future studies may benefit from studying how well such heuristics may perform. Third, we used *wf* to address quantitative individual differences within our specific Conflict task. Future experiments may wish to address whether (i) *wf* reflects stable moral preferences by acquiring evidence of long-term moral commitments and values, or is specific to a smaller class of similar situations such as our helping task, and (ii) whether different individuals may be best captured using qualitatively different models. This might be particularly relevant when including participants with independently demonstrated morally considerate commitments or psychiatric disorders affecting social functioning. Fourth, our evidence that choices are dominated by separable but biased representations hinges on dropout trials in our Online experiment, and our request for probability reports. We introduced these trials to rigorously reveal the nature of the learning that participants deploy in conflict situations - and they indeed provided the data necessary to adjudicate across our models. However, we must consider the possibility that these very trials also influenced participants to separate their representations for self-money and other shocks to enable more optimal decisions during drop-out trials, and more accurate probability reports. Performing a conflict task with only a single devaluation trial on a large number of participants may be a way to exclude this possibility. However, if participants were to have adapted their strategy to optimally fit the requirements of the task, M2Dec would have been even more adaptive.

To summarize, our data sheds light onto the processes at play when adults have to learn that certain actions lead to favorable outcomes for us but harm others, while alternative actions are less favorable for us but avoid harm to others. We show that in our task, the choices of a majority of participants are best captured by a process based on separable but biased representations, in which some brain signals (e.g., in the pain-observation network and our reward signature) covary with prediction errors for the harm to others in ways that do not depend on whether participants prefer to maximize gains for self or minimize harm to others, while others (e.g., in the vmPFC) do depend on this individual preference. We foresee that the mathematical formulation of learning under conflict we introduce and this task will be particularly useful to understand how neurocomputational processes may differ in atypical populations in the spirit of computational psychiatry[44], particularly in antisocial populations.

## Methods

Experiments complied with all relevant ethical regulations, and were approved by the Ethics Committee of the University of Amsterdam, The Netherlands (2017-EXT-8201, 2018-EXT-8864, 2020-EXT-12450). Two independent experiments were performed: an Online behavioral study and an fMRI study. Behavior was tested in a larger sample of participants, but this had to be done online, using the Online platform Gorilla (https://gorilla.sc/), due to COVID-19 restrictions in place at the time. Brain activity was measured in a smaller number of participants in our fMRI scanner. Table 1 gives an overview of the number of participants and experimental conditions included in each study.

### Participants

**fMRI sample.** A total of 27 (37 y ± 17 SD; 27 f; Table 1) volunteers with normal or corrected-to-normal vision, and no history of neurological, psychiatric, or other medical problems, or any contraindication to fMRI (for the fMRI experiment only) were recruited for the fMRI experiment through advertisements on social media. Two of the 27 participants were left-handed. Because the stimuli showed movements of the actor's right hand, to reduce variability induced by lateralization of brain responses, these two participants only performed the tasks off-line (i.e., no fMRI data acquired). Sample size was based on a power calculation to have 80% power to detect a medium effect size for *t* tests

at α = 0.05, and was similar to that in related fMRI studies[3,12] Participants were paid 10€/h for participating and received a bonus payment in accordance to their performance in the learning task. The bonus payment was calculated by dividing by 10 the total amount of earned money during the learning task.

**Online sample.** 79 (25 y ± 7 SD, 39 f) volunteers were recruited through the online platform Prolific (https://prolific.ac/). Sample size was based on a power calculation to have >80% power to detect medium associations (|ρ| = 0.3) between variables across participants at α = 0.05.

Participants taking part in the Online experiment were significantly younger than the ones who took part in the fMRI study (Mann–Whitney test, BF10 = 8.022, $p < 0.001$). Supplementary Note §11 however shows age and gender do not influence model selection. Participants of the online experiment were paid through Prolific the amount of 2.80€ every 30 min, plus the bonus. The Bonus was calculated summing the amount of money corresponding to 15 trials randomly extracted from the learning task plus two trials from the optimization task.

The studies were approved by the Ethics Committee of the University of Amsterdam, The Netherlands (2017-EXT-8201, 2018-EXT-8864, 2020-EXT-12450). All participants provided informed consent and authorization for the publication of images have been obtained.

### Online experiment general procedure

The study started with a general information page, including the informed consent and a description of all steps and tasks (Supplementary Note §2). Participants were also informed that while they perform the experimental tasks online, a second participant is present in the lab. It was explained that the separation of the two participants allowed the experiment to be performed under COVID19 restrictions. In reality, no second participant was invited to our lab, and this cover story served to create a situation in which participants believed their decisions had real implications for self and others, while at the same time minimizing discomfort to others (no shocks actually delivered to the other person). The text also informed them that 15 trials from the Learning task and 2 from the Optimization task (Supplementary Note §4) would be randomly selected at the end of the experiment, and that the money from these trials would be added to their payment, while the electrical stimulation associated with these trials would be delivered on the hand of the other participant in the lab. Participants were informed they would have the choice to witness, via Zoom the other person receiving the stimulation. The experimental session then started with a Stress Tolerance questionnaire assessing participants' susceptibility to stress, which generated the advice to continue or not with the experiment (Supplementary Note §20). No participant received the advice to withdraw, and the session continued with the Optimization Task, aimed at determining the amount of money necessary to create a true conflict for a given person between self-money and other-shocks (this value is referred to as indifference point). This was done by asking participants to choose between different predetermined combinations of money and stimulation intensity (Supplementary Note §4). The experiment then continued with the core probabilistic Reinforcement Learning Task (see below) in which participants had to learn several symbol-outcome associations within the first 10 trials of a block. The experiment ended with a debriefing about the deception, asking how much they believed the cover story (Supplementary Note §5) and with a series of questions to investigate participants' motivations and their feelings toward the experiment (Supplementary Note §6 and 21). Participants were aware that they could withdraw from the experiment at any time.

### Reinforcement learning task
**Conflict condition (Fig. 1b).** Participants had to learn symbol-outcome associations under a moral conflict—i.e., in a context in which choices

that are better for the self (high monetary gain) are usually worse for the other (painful shock), while those that are worse for the self (low monetary gain) are usually better for the other (non-painful shock). Two types of blocks included the conflict: ConflictNoDropout and ConflictDropout blocks. The first 10 trials were common to both block types, and served for participants to learn the symbol-outcome associations. The same symbol pair was presented in all trials of a block, but changed across blocks. In each block one of the symbols would most likely lead to higher monetary reward for the participant, and a noxious electrical stimulation to the dorsum of the actor's hands (lucrative symbol or choice), and the other symbol to lower monetary reward and a non-noxious stimulation to the actor (considerate symbol or choice). Importantly, to partially de-correlate representations of shock and money, for each individual trial, the outcomes of high and low monetary reward and painful and no-painful shock to others were drawn independently, resulting in the four possible outcome combinations shown in Fig. 1d, e.

While the monetary reward was presented in the outcome screen as a numerical amount of euros, the outcome for the other was presented as a video recording of the author A.N. while receiving the stimulation (Supplementary Note §1). Participants could therefore infer whether the stimulation was or not noxious only based on the facial reaction of the actor in the videos. Participants were aware that the person in the video and the other participant in the lab were not the same individual, and that the videos were pre-recorded to illustrate what responses to the electrical stimulations would look like. The ConflictNoDropout blocks only contained these first 10 trials, after which participants were asked to report the probability of each symbol to be associated with high money and high shock (Fig. 1b, c). The ConflictDropout blocks, after these initial 10 T, continued with 10 additional trials. Before the 11th trial is presented, a text informs participants that either the outcome money is removed (ConflictMoneyDropout) or the shock (ConflictShockDropout). Over the last 10 trials of the ConflictMoneyDropout, participants would therefore only see the videos of the actor receiving the stimulation, and no money would be rewarded, while over the last 10 trials of the ConflictShockDropout participants would only see the money outcome. Participants were informed that the probability of each symbol of being associated with high or low stimulation would remain the same over the 20 trials.

**NoConflict condition (Fig. 1c).** This condition was identical to the Conflict condition with the exception that the symbol-outcome associations did not likely introduce a moral conflict because the symbol that was usually best for the self was also usually best for the other: one symbol was most likely associated with high monetary reward for the participants and non-noxious stimulation to the other participant, while the other symbol was usually associated with low monetary reward and noxious stimulation (Fig. 1e). There was, therefore, a clear incentive to choose for the symbol leading to the higher reward, as it was also the symbol that would be most beneficial for the other. As for the Conflict condition, two types of block were present: the NoConflictNoDropout in which only the first 10 trials were presented, followed by the probability report, and the NoConflictDropout which included 20 trials and after the 10th trial, either money or shock outcome was removed.

The position of each symbol on the screen within a pair was randomized across trials. Symbols composing a pair were kept the same across individuals (i.e., the same two symbols formed the same pair throughout the experiment and across participants), but pairs were randomly distributed across conditions. Symbol-outcome associations were determined based on the matrix in Fig. 1d, c and were kept constant across participants. ConflictDropout, ConflictNoDropout, NoConflictDropout, and NoConflictNoDropout blocks were presented in a randomized order within and across participants.

**Explicit recall questions.** At the end of each of the four ConflictNoDropout and four NoCoflictNoDropout blocks only, participants were asked, for each symbol separately, to recall the probability of that symbol to be associated with higher monetary reward and with the noxious stimulation using a scale from 0 to 100% (Fig. 1b, c). The order in which the two symbols were presented and the type of question (association with high shock or with high reward) was randomized between participants. The slider starting position was always at 50%.

**fMRI experiment general procedure**
The fMRI experiment included four sessions of the Helping task, following procedures similar to those used in combination with EEG in Gallo et al.[4], and one session of the ConflictNoDropout condition of the reinforcement learning task used in the Online experiment (Supplementary Fig. 2), which was always collected after the first two sessions of Helping. fMRI results of the Helping task will be the object of a separate publication. In the current work we only report the fMRI data associated with the ConflictNoDropout condition, and the amount of money participants were willing to give up to reduce the intensity of the confederate's stimulation in the Helping task in order to test the predictive power of the weighting factor.

*Helping task* is a decision task, in which each trial starts by showing a video of another participant (author S.G.) receiving noxious stimulations of different intensities (Supplementary Note §3). Participants were then endowed with 6 euros, which they could decide to keep for themselves or to give all or in part up to proportionally reduce the intensity of the next stimulation. A second video showing a second stimulation of intensity equal to the "intensity of the first video - donation", was then presented. Videos were pre-recorded and have been rated on perceived pain intensity by an independent group of participants. We used these independent ratings to select which video to present. The intensity of the first video was selected randomly at the beginning of the experiment. Videos either showed the facial reaction of the confederate or the hand reaction without facial expressions. Because the Learning experiment only showed facial reactions, only donations associated with the trials showing the facial reactions in the Helping task were included in the current manuscript (results would though not change when including the full donation dataset (Kendall's $Tau_{(wf,donation)} = -0.54$, $BF_{10} = 13.19$, $p = 0.004$). Participants were led to believe that the videos they saw were a life-video-feed of another participant receiving these shocks in a closeby room, although in reality they were pre-recorded movies. All participants were presented the same videos, albeit in randomized order, so that we can compare the average donation of the participants as a willingness to give up money to reduce the pain of others.

As the participants of the fMRI experiment came in person to the lab, the cover story was slightly different from that adopted in the online version. For the fMRI we used the same cover story used and validated in Gallo et al.[4]. Each participant was paired with what they believed to be another participant like them, although in reality it was a confederate, author S.G. They drew lots to decide who plays the role of the learner (or the donor in the Helping task) and of the pain-taker. The lots were rigged so that the confederate would always be the pain-taker. The participant was then taken to the scanning room while the confederate was brought to an adjacent room, connected through a video camera. Participants were misled to think that electrical stimulations were delivered to the confederate in real-time, and that what the participants saw on the monitor was a live feed from the pain-taker's room. In reality, we presented pre-recorded videos of the confederate's reactions.

In contrast to the Online experiment, in which the high-money amount was selected for each participant using the Optimization task, in the fMRI Learning task, the high-money amount offered was fixed at 1.5€ for all participants, and corresponded to the average amount

associated with the indifferent point in the Optimization task (1.53 ± 0.37 SD; Supplementary Note 2). The low-money amount was the same as in the Online experiment, 0.5€.

At the end of the fMRI tasks, participants were debriefed and asked to fill out the interpersonal reactivity index (IRI) empathy questionnaire[45], and the money attitude scale (MAS) [46]. To assess whether participants believed that the other participant really was receiving electrical shocks, at the end of the experiment, participants were asked 'Do you think the experimental setup was realistic enough to believe it' on a scale from 1 (strongly disagree) to 7 (strongly agree). All participants reported that they at least somewhat agreed with the statement (i.e., 5 or higher).

The fMRI task was programmed in Presentation (Version 22.1) www.neurobs.com), and presented under Windows 10 on a 32inch BOLD screen from Cambridge Research Systems visible to participants through a mirror (distance eye to mirror: ~10 cm; from mirror to the screen: ~148 cm).

## Statistical approach to behavioral data

Quantification of choices was performed in RStudio (Version 1.1.453) and Matlab (https://nl.mathworks.com/). Statistical analyses were then performed using JASP (https://jasp-stats.org, version 0.11.1), to provide both Bayes factors and $p$ values. Bayes factors were important because in many cases we are as interested in evidence of absence than in evidence of the presence of an effect, and $p$ values cannot quantify evidence for the absence of an effect[30]. For instance, we expect that removing money outcomes in considerate participants should not alter choices, while removing shocks should. We used traditional bounds of $BF_{10} > 3$ to infer the presence of an effect and $BF_{10} < 1/3$ to infer the absence of an effect[30,47]. Two-tailed tests are indicated by $BF_{10}$, i.e., $p(Data|H_1)/p(Data|H_0)$ while one-tailed tests are indicated by $BF_{+0}$. Where ANOVAs were used, we report $BF_{incl}$ which reports the probability of the data given a model including the factor divided by the average probability of the data given the models not including that factor. Normality was tested using Shapiro-Wilk's. If this test rejected the null-hypothesis of a normal (or bi-variate normal for correlations) distribution, we used non-parametric tests such as the Wilcoxon signed rank or Kendall's Tau test, while if the null-hypothesis was not rejected, we used parametric tests. We always used default priors for Bayesian statistics as used in JASP.

## Participant's choices categorization

We categorized participants' preferences by determining whether their choices deviated from chance according to the cumulative binomial distribution. Specifically, for the Online experiment, we used a binomial considering 120 trials (10 trials × 4 ConflictNoDropout blocks + the 10 first trials × 8 ConflictDropout blocks). If participants fell above the 97.5% tail of a binomial distribution (with $N = 120$ choices, and $p = 0.5$ probability of a considerate choice in the absence of a preference), i.e., if they chose the considerate option in more than 71/120 trials, we classified them as having demonstrated a 'Considerate preference'; if they fell below the 2.5% tail, i.e., chose the considerate option in less than 49/120 trials, as having demonstrated a 'Lucrative preference'. If they fell between these bounds, we considered them as failing to demonstrate a preference, and classified them as having 'Ambiguous preference'.

For the fMRI experiment, we applied the same logic on the choices in the 60 available trials (10 trials × 6 ConflictNoDropout blocks), and participants with fewer than 22 or more than 38 considerate choices were then classified as demonstrating Lucrative or Considerate preferences.

## Computational modeling of behavioral data

Our experiment represents a variation of a classical two armed-bandit task and was modeled using a reinforcement learning (RL) algorithm with a Rescorla-Wagner updating rule[31]. We compared 4 models

explained in Fig. 1e. Models were fitted in RStan (version 2.18.2, http://mc-stan.org/rstan/) using a hierarchical Bayesian approach, i.e., by estimating the actual posterior distribution through Bayes rule. Our models were adapted from the R package hBayesDM (for "hierarchical Bayesian modeling of Decision-Making tasks") described in detail in Ahn et al. (2017)[48].

Model comparison was also performed in a fully Bayesian way. We first fit the data on the first 10 trials of each Conflict block. NoConflict blocks were fitted separately for Supplementary Fig. 6a, and parameter estimates reported in this manuscript for $wf$, $LR$, and $\tau$, are therefore not influenced by the NoConflict choices. To assess the ability of a model to fit data of these initial 10 trials that were included in the parameter-fitting, we use the leave-one-out informative criterion (LOOIC[34]), which computes a pointwise log-likelihood of the posterior distribution to calculate the model evidence, rather than using only point estimates as with the other methods, e.g., of Akaike information criterion (AIC[49]) and the deviance information criterion (DIC[50]). The LOOIC is on an information criterion scale: lower values indicate better out-of-sample prediction accuracy. For the ConflictDropout blocks, on the other hand, we aimed to assess the ability of a model fitted on the first 10 trials of a ConflictDropout block to predict behavior on the devaluation trial (trial 11). The model was then fitted using the first 10 trials of all the ConflictDropout blocks, and no leave-one-out procedure was necessary to assess the predictive performance on the 11th trial because it was not included in the parameter fitting. We thus directly used the log-likelihoods of the 11th trial, which capture the probability of the choices on the 11th trial given the model fitted on the preceding 10 trials. Specifically, at the group level, we added the log-likelihood of all the 11th trials of all the participants for each of the 4000 draws from the posterior distribution, and then examined the distribution of these values (Fig. 4c). To compare the ability of different models to predict this 11th trial at the single subject level, we then calculated a likelihood ratio (similar to a Bayes factor) for each participant: we summed the log-likelihood of the eight 11th trials available for each participant, and used the mean across the 4000 draws as our point estimate. We then calculated the ratio of the exponents of the log-likelihoods for the competing models.

As priors on the hyperparameters $LR$ and $wf$, we use the recommended Stan method of using a hidden variable distributed along $N(0,1)$, that is then transformed using the cumulative normal distribution to map onto a space from 0 to 1. For $\tau$, we use the same method but then multiply the result by 5 to have the function map onto the interval [0,5].

For the Dropout blocks of the Online experiment, in which one of the two outcomes was removed after the 10th trials, the three models M1, M2Out, M2Dec were modified from the 11th trial to account for the fact that participants were told which quantity would be removed. For the M2 models, this was implemented by setting $EV = 0$ for the removed quantity before decision-making on the 11th trial. In addition, $wf$ is modified only to value the remaining EV (i.e., $wf = 1$ if shocks are removed and $wf = 0$ if money is removed). For M1, we cannot reset expectations for a specific quantity (shock or money) and EV has to remain unchanged. However, the $wf$ is adapted (i.e., $wf = 1$ if shocks are removed and $wf = 0$ if money is removed), which maximizes what can be learned from the following trials. In all cases, the outcomes for the missing quantity are always set to zero after Dropout.

## MRI data acquisition

MRI images were acquired with a 3-Tesla Philips Ingenia CX system using a 32-channel head coil. One T1-weighted structural image (matrix = 240 × 222; 170 slices; voxel size = 1 × 1 × 1 mm) was collected per participant together with an average of 775.83 EPI volumes ± 23.11 SD (matrix M × P: 80 × 78; 32 transversal slices acquired in ascending order; TR = 1.7 s; TE = 27.6 ms; flip angle: 72.90°; voxel size = 3 × 3 × 3mm, including a .349 mm slice gap).

## fMRI data preprocessing

MRI data were processed in SPM12[51]. EPI images were slice-time corrected to the acquisition time of the middle slice and realigned to the mean EPI. High-quality T1 images were coregistered to the mean EPI image and segmented. The normalization parameters computed during the segmentation were used to normalize the gray matter segment (1 mm × 1 mm × 1 mm) and the EPIs (2 mm × 2 mm × 2 mm) to the MNI templates. Finally, EPIs images were smoothed with a 6 mm kernel.

## fMRI data analysis

The design matrix to analyze the fMRI data of the learning task included 13 regressors: (1) A decision regressor starting with the appearance of the two symbols and ending with the button press of the participant. (2) The outcome regressor was aligned with the presentation of the video and had a fixed duration of 2 s, corresponding to the duration of the stimulus. (3) A button-press regressor with zero duration was aligned to the moment of button-pressing. (4–5) The decision regressor had 2 parametric modulators ($EV_M$ and $EV_S$ of the chosen option); and (6–7) the outcome regressor 2 parametric modulators ($PE_S$ and $PE_M$). The modulators were derived from the winning M2Out, then mean-subtracted and *wf*-normalized before being entered into the design matrix (see *wf*-normalization below and Supplementary Note §15). (8–13) Finally, six regressors of no interest were included to model head translations and rotations.

Which parametric modulator to include was based on the results of parameter correlations and recovery. Prediction errors and actual outcomes could not be used within the same GLM as they were too highly correlated ($r(PE_S, Out_S) = 0.741$, ranging from 0.750 to 0.866 across our 27 participants; $r(PE_M, Out_M = 0.749$, ranging from 0.781 to 0.862). We however examined if signals were associated with Out vs PE using the method suggested by Zhang and colleagues[41]. During the outcome phase, we therefore only included $PE_S$ and $PE_M$, which are only weakly correlated (average $r = -0.26$, ranging from $-0.49$ to $-0.03$ across the participants) and for which parameter recovery is robust (Supplementary Note §17).

**wf-normalization.** Because we are interested in whether $PE_S$ or $PE_M$ representations depend on *wf* or not, we divided $PE_S$ with (1-*wf*) and $PE_M$ with *wf* before entering them into the design matrix. As a result, the first $PE_S$ value in the parametric modulator would always be $PE_S = -1$ if it was a high-shock outcome or $PE_S = +1$ for a low-shock outcome, independently of the participant's *wf* value. If signals covary with $PE_S$ in a way that does not linearly depend on *wf*, the parameter estimate across participants ($\beta_{PES}$) would violate $H_0: \beta_{PES} = 0$, but not $H_0: r(\beta_{PES} = 0, wf) = 0$. If signals covary with $PE_S$ in a way that does linearly depend on *wf*, it would violate both of these $H_0$. Note that for outcomes, the coding was +1 for good outcomes (i.e., high money or low-shock) and −1 for bad outcomes (i.e., low money or high shock). EV and PE follow that polarity. The same was applied to $EV_S$ and $EV_M$, which were divided by (1-*wf*) and *wf*, respectively. This approach is illustrated with an example in Supplementary Note §15.

Results were then analyzed in two ways. First, to improve reverse inference, we used two multivariate signatures the affective vicarious pain signature (AVPS[24]) and the reward signature (RS[29]) To explore if signals in this network covary with $PE_S$ or $PE_M$ we then simply dot-multiplied the $PE_S$ or $PE_M$ parameter estimate volume for each participant separately with the AVPS and RS, after having brought the AVPS and RS into our fMRI analysis space using ImageCalc. The result of the dot-multiplication indicates how much the covariance with $PE_S$ or $PE_M$ loads on the AVPS or RS. We then brought these values into JASP, and compared them against zero and correlated them with *wf*. Because the loadings were normally distributed, we used parametric analyses. Second, we performed a similar analysis at the voxel level, by bringing the parameter estimate images for $PE_S$ and $PE_M$ into a second-level linear regression using a constant and *wf* as the two predictors. A *t* test on the constant then reveals regions in which signals covary with $PE_S$ or $PE_M$ after removing variance explained by *wf*. A *t*-test on the *wf* parameter estimate then reveals regions where the signals covary with $PE_S$ or $PE_M$ in ways that depend on *wf*. To test if signals covary with 1-*wf*, we simply used a negative contrast in the *t*-test. Results were family-wise corrected at the cluster level using the established two-step procedure in SPM: (i) for cluster-cutting we visualized results at $p_{unc} < 0.001$ $k = 10$, and identified the FWEc minimum cluster-size for family-wise error correction from the results table, (ii) we reloaded the results at $p_{unc} < .001$ $k = $ FWEc, so that all displayed results survive FWE at cluster-level. The same was done for $EV_S$ and $EV_M$, but only reported in Supplementary Fig. 18. For contrasts not revealing significant clusters at that threshold, we also mention results that were cluster-cut at $p_{unc} < 0.01$, and then applied the FWEc that SPM calculates at that cluster-cutting threshold. However, it should be noted that cluster-extent corrections following such permissive cluster-cutting thresholds are more subject to false positives and should be interpreted with care[52].

## Reporting summary

Further information on research design is available in the Nature Portfolio Reporting Summary linked to this article.

## Data availability

The behavioral choice data generated in this study have been deposited in OSF.io under https://doi.org/10.17605/OSF.IO/RK8W4. The raw fMRI data are protected and are not available due to data privacy laws. The processed fMRI data are available at OSF.io under https://doi.org/10.17605/OSF.IO/RK8W4. The data directly illustrated in the figures are provided in the Supplementary Information/Source Data file. Source data are provided with this paper.

## Code availability

The code for our learning models is available at OSF.io with the identifier (https://doi.org/10.17605/OSF.IO/RK8W4) in the folder Fig. 5 \StanCode, code to generate all other figures of the main manuscript can be found in the respective directory at the same OSF.io identifier.

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

## Acknowledgements

We thank C. Gavanozi, I. Gembutaite, and B. Hoekzema for helping with the data acquisition of pilot data. We thank A. Veggerby Lind for helping record the stimuli. We thank A. Gentile for input and help with the learning models, and P. Lockwood as her work inspired the development of the learning tasks. Alessandra D. Nostro and Nathan J. Evans contributed equally to the role of the second author. We thank J. Campdepadros for helping with data collection during the revision. The research was funded by a European Union's Horizon 2020 research and innovation program grant of the European Research Council ERC-StG 'HelpUS' 758703 to V.G., and the Dutch Research Council (NWO) VIDI grant

452-14-015 to V.G. and VICI grant 453-15-009 to C.K. M.S. received funding from the John Templeton Foundation Grant 21338.

## Author contributions
The experiments were conceived by V.G. and C.K. with input from all authors. Funding and project leadership by V.G. The fMRI data were collected by K.I. with the help of S.G.; the behavioral data by L.F. Pilot data for the preparation of the study were obtained by R.P., A.N., and S.G. FMRI data were analyzed by K.I. with guidance from V.G. and C.K.; the RLT models were developed and programmed by A.N., C.K., L.D., L.F., M.S., N.E., and V.G. SPHS performed analyses during the revision process and generated the reward signatures. L.F., K.I., A.N., N.E., M.S., C.K., and V.G. wrote the manuscript with edits and comments from all other authors.

## Competing interests
The authors declare no competing interests.
