## [Peer Review File · Nature Communications]

Neuro-computational mechanisms and individual biases in action-outcome learning under moral conflictReviewers' Comments:

Reviewer #1:

Remarks to the Author:

This study examines how people learn to choose between different options that differ between gaining monetary rewards for self and inflicting pain on others. The researchers conduct two experiments, one using an online behavioural task in 79 people and one using fMRI in 25 people. The researchers find that there are substantial individual differences in the preferences of different people in how they select stimuli on the task. They suggest that a bias towards expectations of self gain and other pain is related to the ventromedial prefrontal cortex and pain prediction errors are represented independently of individual preference.

The task the researchers use is novel, and they apply computational modelling techniques to analyse the data and distinguish between different behaviours which is impressive. However, there are major concerns regarding the framing of the study as able to distinguish between model-free and model-based learning. The final sample size for the imaging study is 25, which when coupled with such strong individual differences in how people approached the task, challenges the robustness of the imaging results. There is a lot of work in the manuscript and an impressive amount of analyses, but I do not feel the conclusions surrounding model-free and model-based learning or the associated neural signals are supported by the data and analyses.

Major:

1. Framing surrounding model-free and model-based learning. The researchers frame their experiment as distinguishing between model-free vs. model-based learning by the kinds of computational model that they compare. However, to distinguish model-free and model-based learning they would need a behavioural paradigm that favours different strategies to be adopted, such as the two-step task by Daw and colleagues or subsequent modifications suggested by Kool and colleagues. From the current task it is not possible to distinguish between these possibilities. Their weighting factor does not capture the relative reliance on model-free and model-based learning, which is important to make inferences about the use of model-free and model-based behaviour.
2. Low power for individual trial analysis. There is a concern that very strong inferences are made regarding trial 11 and whether there is enough power to compare performance on an individual trial across participants. Even if there are multiple repetitions of that trial could participants come to expect it over blocks? If the question is how people weigh up money and shocks differently when deciding for others it would seem better to use a decision-making task without any learning component.
3. Simply because studies have already investigated neural responses to reward in the self this does not appear a convincing justification to exclude inferences and analysis of these responses neurally, as the task manipulates both rewards and shocks.
4. Throughout the paper sometimes Bayesian statistics are used and sometimes p values. I understand the wish to use Bayesian stats to draw inference on the strength of the null, but it would be important to qualify the strength of evidence, and consistently add p values when there are bayes factors.
5. Sample. There are quite large gender and age differences between the online study and the fMRI study. The fMRI participants are significantly older and all female compared to the online study. I understand that the researchers suggest no difference overall in choices between the two groups but the bayes evidence is not very strong. There are several studies showing age differences in reinforcement learning tasks. More generally, an fMRI sample size of 25 when there are quite large individual differences is underpowered. There is a section on power analysis but this seemed to run parameter recovery on imaging signals rather than to show the computational model could reproduce

behaviour on the task.

6. Imaging analysis. One of the main imaging analyses focuses on the absence of correlation with the weighting factor and suggests that brain signals do not depend on personal preferences. However, in the fMRI study most participants were identified as considerate or ambiguous, so this is not surprising. Is there enough variance to assess this?

Minor:

1. Terms are used inconsistently in the manuscript. "Considerate" people choose "pain reducing" option but "lucrative" people choose "lucrative" option – either match both of these terms or make them different?

2. Lack of clarity in methods

a. Not clear if order of conflict / no conflict mixed and randomised

b. P3 line 87 – "Importantly, to partially de-correlate representations of shock and money, the probabilities of high and low monetary reward and pain and no-pain to others were drawn independently." – Not clear whether it was probabilities drawn independently or outcomes that were drawn independently from the set probabilities

3. Interpretation

a. Inference on p14 line 340 doesn't seem justified re. conflict vs. no conflict – no conflict just means removing one option, it doesn't change which option is best

b. The researchers suggest that they introduce a "neurocognitive model" (p 28 line 761) but this is not clear and hasn't been discussed before

4. Clarity / consistency throughout the manuscript:

a. NoConflict vs. No-Conflict

b. P56 line 1482 & fig 7b legend on plot the middle group is called "neutral", but figure 4c legend on plot "indifferent", other places "ambiguous"

c. Consider using different symbols for kendalls tau and tau softmax for clarity p 63, line 1594.

d. Supplementary table 3 - rename wf_10T_M2Dec to give a more descriptive name.

e. Was eye gaze data collected through gorilla? prolific is mentioned p 84 line 1868, as prolific does not contain a way to collect eye gaze data as far as I am aware?. Fig S7 mentioned but it is Fig S8

Reviewer #2:

Remarks to the Author:

The current study examined how people learn associations between actions and their moral and financial consequences, as implemented by means of personal financial gains and pain delivered to other people. The study examines such learning for new associations under conflict and no conflict between these two outcome features, as well as after a change in such learned contingencies. The authors conducted two experiments, one on-line behavioral experiment and one fMRI experiment. In behavioral data analyses, the authors categorized subjects into three groups (considerate vs.

ambiguous vs. lucrative), based on their proportion of pain-reducing choices, and showed that these different groups exhibit different dynamic changes in choice patterns over time (learning curves). Moreover, after contingency changes (when either self-gain or other-pain was removed), subjects changed their choice tendency more strongly when the feature was removed that they weighted more strongly before. These results suggest that people are able to learn the values of each feature, not just a compound value integrating both features. This was also confirmed by computational modelling analyses suggesting that the data are best fit by models that incorporate separate terms for the values of the two features. The weighting parameter estimated from the winning model was related to not only pain-reducing choices but also altruistic donation behavior in an independent task. Finally, fMRI results suggested that the prediction error of other-pain intensity was represented in the pain-observation network, and that activity of the ventromedial prefrontal cortex was associated with individuals' bias on a specific feature.

In general, I enjoyed reading this paper. The questions are quite detailed but interesting, the experiment is generally well designed, and the analyses are thorough. However, I still have several concerns regarding the data analyses and interpretation that the authors should address to render their study conclusive.

Major issues:

1. The authors claim that their design and analyses allow them to separate "model-free" from "model-based" learning, in close similarity to similar approaches in other studies in animals and humans. I am not sure this statement is in line with how most people in the field define and study model-based learning. First, the learning task in the present study is fundamentally different from the variants of 2-stage learning tasks commonly used to separate model-based and model-free learning, which actually make opposing choice predictions for critical trials. I do not think model-based learning is unambiguously separated from model-free learning by separate versus conjoint feature valuation, as the authors imply. Where does this definition come from? Second, the winning (M2Out) and alternative models (M1 and M2Dec) used here are all variants of feature-based reinforcement learning models, and thus do not seem to computationally differentiate model-based learning from model-free learning. Which feature of the computational models allow inferences on the use of a cognitive model of reward, pain, and feature contingencies? If the authors really want to separate "model-based" from "model-free" learning with their task and computational models, they should clearly define the rationale underlying this separation and clarify the similarity and differences between this rationale and that commonly used in the literature. However, the authors could also focus solely on the question whether financial versus moral features are learned separately and how they are integrated; this is an interesting issue that does not necessarily need to be related to the debate about model-based vs. model-free learning.

2. Although the authors showed that people in the ambiguous group do not have clear preferences for self-gain or other-pain, it is still unclear how and why these individuals differed from those with clear preferences. The authors claim that "the ambiguous preference group appears to contain a mix of individuals that do not show significant learning in our task even without conflict, and individuals that can learn without conflict but fail to display clear preferences in the presence of conflict". However, the data do not clearly support this conclusion. For example, Figure 4A-D shows no clear group difference in perceived association between ambiguous and considerate or lucrative people. Thus, ambiguous people seem to learn the symbol-outcome associations as well as the others, even in conflict conditions. Relatedly, while Figure 5A and supplementary Figure 5 show no significant changes from the 10th to the 11th trial in ambiguous people, these choices were still significantly different between different contingency contexts (i.e., shock removed vs. money removed) and showed adaptations from the 11th to the 20th trial. Thus, ambiguous people can in fact learn the symbol-outcome associations, implying that the flat "learning" curves may reflect a neutral preference for self-gain vs other-pain rather than deficient association learning. If the authors want to maintain that the learning process differs between the groups, they should really show statistical differences in estimated

learning-rate parameters across groups. In addition, since the learning curves are averaged across blocks, it is hard to tell if the flat curves are driven by neutral preferences across all the trials or by different strategies across different blocks (e.g., greater weight on other-pain in half blocks and greater weight on self-gain in the half blocks). This should be tested for as well.

3. Related to the last point, I wonder to what extent the current design and modelling approach can differentiate the learning rate and the weighting factor. Are these two parameters correlated?

4. In the imaging analyses, the authors selectively focused on how PES covaried with the AVPS. Why? One could equally have hypothesized such a relationship for the neural prediction of other-pain and the AVPS; or for the prediction or prediction errors for self-gain and reward networks. Moreover, the authors directly relate the outcome of a multivariate analyses with one selectively chosen results of a univariate analysis. This does not seem to be clearly motivated by a priori hypotheses. The authors should be more unbiased in their report of the neuroimaging results. In particular, they should provide a clear rationale for which neural hypotheses they want (or don't want to) to examine, and they should comprehensively report the corresponding analyses even if they don't yield significant results.

5. In the neuroimaging analyses, the authors used the wf-normalized PE to avoid "that wf influences results twice". They "divided PES with (1-wf) and PEM with wf", so that the wf-normalized PES equals $(OutS*(1-wf) - EVS)/(1-wf)$. I don't fully understand this rationale. First, wf is a constant for each subject, so the wf-normalized PE should be highly correlated with PE (and may even lead to identical neuroimaging results if this scaling factor is removed when the corresponding regressor is standardized during GLM construction). Second, since the wf is embedded in the first-level PE analysis, it is unclear how this transformation can avoid "that wf influences results twice", given that now wf is in fact included in both the first-level and second-level analyses. Please provide a clearer rationale and possibly simulation to illustrate the usefulness of (and necessity for) this transformation.

6. Figure 8D shows regions with positive and negative loading on the AVPS. What is the neurophysiological or psychological basis for these different signs? To get the measure of loading on AVPS shown in Figure 8B, the authors actually sum up across all voxels (that show opposite relations to the APVS). Is it possible that this removes potentially interesting results of specific brain regions for both PES and PEM? The authors should provide information on this.

7. The authors may have mainly focused on PES because it correlates with PEM, but it looks like regions associated with PEM and PES do not entirely overlap (as shown in the supplementary tables). Have the authors estimated GLMs in which both PES and PEM are included as parametric modulators? It would be good to see whether such GLMs reveal qualitatively similar results or in fact distinct results for PES and PEM.

Minor:

8. I suggest that the authors include GLM analyses of decision utility as control analyses. Decision utility may also be represented in vmPFC, so it is important to differentiate the roles of vmPFC in representing decision utility and PEM.

9. The authors claim "prediction" based on correlation analyses in some cases. This is misleading. Please rephrase all the relevant statements.

10. On Page 15, the authors hypothesized that the M2Out model should be better than the other models, based on their model-free results. This is unclear (in fact, it comes out of the blue). Please provide a detailed explanation for why the model-free results support this hypothesis.

11. In Figure 6, the authors did not include results of the ambiguous group. The authors should at least put these results in the supplementary materials.

12. In Figure 2B, there should be a comma, rather than "+", between two EVs. If it is a typo, please correct it. If it is not, then the authors need to explain this relation.

Reviewer #3:

Remarks to the Author:

Fornari, Ioumpa and colleagues used a combination of behavioural tasks, computational modelling, and neuroimaging to understand the mechanisms that underlie decisions in a morally conflicting situation. Here, conflict is defined as the trade-off between selfish choices (more money for the participant) vs prosocial choices (a smaller number of shocks for the other player). The main behavioural results show individual differences in how these two options – prosocial vs selfish – are traded with each other. This preference or bias does not only impact decisions but also the learning about the options' contingencies. A computational model that includes an individually specific preference parameter (wf) that scales the prediction error during the outcome phase is predictive of participants' behaviour, in particular during the 11th trial where participants behaviour can be distinguished according to model-based vs. model-free learning. By using a devaluation task and computational modelling, the authors show that participants are in fact representing the options contingencies in a model-based manner. Finally, they make use of fMRI to understand the scaling of the prediction error by the model-derived parameter wf and find primarily a correlate in vmPFC.

The authors explain their approach and results in a clear and transparent manner. The combination of methodological approaches is very interesting, in particular the verification of computational parameters in a second independent helping task. While the online study offers a rich data set that has been analyzed in an interesting manner, the fMRI data set lacks the same rigorous approach. In the comments below, I give suggestions of additional analyses that might be interesting to look at when analyzing neural but also behavioural data.

Major points:

[model comparison]

The authors show that participants can be divided according to their choice preference (Figure 3) and that this preference also impacts learning about contingencies (Figure 4). However, the computational model that is selected is a model that includes the preference weighted parameter, wf , in the outcome phase and not choice phase. Would it not be more accurate to assume that both choice and PE are impacted by wf ? How does such a model including wf for both choice and outcome phase perform during model comparison?

[fMRI analysis]

In the methods, the authors state that it is not possible to look at the decision phase (line 1020), because of the difficulties of time-locking and a possible confound of activity related to button press. The decision phase can be analysed as long as it is decorrelated from the outcome phase, which can either be done with a long enough ITI jitter or statistical decorrelation of variables across both phases. Here, the ITI jitter is long enough, which should allow to inspect neural correlates at time of the decision phase, where variables should be time-locked to the onset of the decision screen. Second, the authors should include a regressor time-locked to all button presses modelled as stick function to account for movement-related effects; as is usually done in fMRI analysis (eg see Wittmann et al., Neuron 2016 Supplementary material). This would also address their second point (line 1022). It would be interesting to see which brain regions correlate with a decision related variable that computes the expected value between the considerate and lucrative choice. Again, as has been done

before, if on a given trial participants choose the considerate option, the authors could include the difference between the chosen EV (shock) minus the unchosen EV (money). This variable can either be directly calculated, normalised and included into the GLM or alternatively, the chosen EV(s) and unchosen EV(m) can be included and their contrast can be calculated in SPM. Further, it would be interesting to extend this analysis by including a covariate into the second-level analysis that weights this value difference by wf.

Some other points regarding the fMRI analysis:

1. Did the authors normalize their regressors before including them into the GLM? I might have overlooked it, however this should be the case.
2. Weighting the prediction error in the outcome phase by wf is in itself interesting. However, it seems rather arbitrary to divide the PE by the wf parameter. Equally, one could multiply it, in line with the computational model. However, both approaches divide a variable that changes trial by trial (as can be seen in Figure 2), by one static preference that is fitted to all data. Alternatively, the authors can include wf as a covariate into the second-level analysis and inspect whether the PE of shock or money covariates with wf. This would also allow you to see whether there are individual areas that covariate with wf, instead of calculating a dot product between a whole network (AVPS).
3. It is surprising that there is no activation in ventral striatum for prediction errors, which is observed across many studies. More related to the current study, Lockwood et al. (2020, PNAS) shows prediction errors of pain avoidance for other (and self) in VS which is conceptually similar to the current aim. Is there activation in VS at subthreshold level?

[behavioural and neural regression analysis]

Most analyses rely on the initial categorisation of three preference groups. This categorisation is somewhat arbitrary and let's someone wonder what the ambiguous group entails. For example, in Supplementary Figure 3, it shows clearly that this group was also motivated to prevent harm to others compared to other motivations. Noteworthy, I still think it is an acceptable approach as the authors show analyses that rely on more continuous measures (Figure 4E). However, it might be informative not only to analyse a summary measure such as proportion of lucrative or considerate choices, but instead look at a trial-by-trial choice analysis. Such an analysis could potentially not only show a choice bias but also a learning bias and might also help to link more directly to the fMRI GLM. A logistic regression predicting trial-by-trial choices (eg, 1= lucrative, 0=considerate) could be predicted by model-derived variables such as expected value difference between the lucrative and considerate choice and the prediction error at the last trial/ last encounter of the choice. This would allow to inspect not only choice bias, that would be captured in the intercept, but also the learning bias, which would be the regression weight of the prediction error on the next choice. The same variables could then also be used to analyse the decision phase in the fMRI analysis.

[conceptual novelty]

I appreciate the authors' methodological efforts (devaluation task, online task, fMRI study, an array of self-reports and questionnaires), however, the study resembles quite closely to other studies that inspect the trade-off between choosing beneficial options for self vs. others. In particular, one previous study by Lockwood et al. (2020, PNAS) inspects the computational mechanisms of model-based and model-free learning in a two-step task where participants choose on behalf of another person or for themselves whether they want to administer shocks or reward. One difference between studies is the type of decision participants make here compared to the other study: trade between something good for oneself or another person. The authors should feel free to argument against this, but I think to increase the scientific contribution of the current study, it would be interesting to inspect the decision phase as mentioned before by looking at the decision correlates that bias choices instead of only focusing neural analyses on the outcome phase.

Minor points:

- The introduction and discussion are very long. Can you reduce it to the most important topics? For example, it is good to give an overview about the methods in the introduction, however this overview is very detailed and could instead be absorbed into the methods section.
- It is great that you use a second independent task to verify your computational parameter. In many other studies, computational parameters are correlated with psychological questionnaires to verify their external validity. However, going forward it might be worth considering the inclusion of an independent short task. For transparency and interest, can you please include the correlation between the wf and the psychological questionnaires; all included questionnaires are sensible and it would be interesting to see how they relate to wf.
- It is great that the authors use Bayesian statistics, however it is still not the most common method to use, therefore can the authors add a short explanation, similar to line 947 into the beginning of the results section.
- Typos: Line 216 't' too much
Line 937: ranging
- Figure B last panel: it would be good to explain why 64% is bold.
- How was the sample size for the online study selected?
- Title: all behavioural and neural results aim to show that there is a bias that impacts decision and learning when having to choose between prosocial vs selfish choices. It would be great to see something about this in the title.

Reviewer #1

This study examines how people learn to choose between different options that differ between gaining monetary rewards for self and inflicting pain on others. The researchers conduct two experiments, one using an online behavioural task in 79 people and one using fMRI in 25 people. The researchers find that there are substantial individual differences in the preferences of different people in how they select stimuli on the task. They suggest that a bias towards expectations of self gain and other pain is related to the ventromedial prefrontal cortex and pain prediction errors are represented independently of individual preference.

The task the researchers use is novel, and they apply computational modelling techniques to analyse the data and distinguish between different behaviours which is impressive. However, there are major concerns regarding the framing of the study as able to distinguish between model-free and model-based learning. The final sample size for the imaging study is 25, which when coupled with such strong individual differences in how people approached the task, challenges the robustness of the imaging results.

There is a lot of work in the manuscript and an impressive amount of analyses, but I do not feel the conclusions surrounding model-free and model-based learning or the associated neural signals are supported by the data and analyses.

We thank the reviewer for appreciating our work, and for these constructive comments. As mentioned below, we reframed the manuscript.

We additionally agree that the sample size of the imaging study could have been larger, but COVID impeded such extension initially, and now the person that played the confederate role finished her PhD, and we were reticent to combine data acquired with different confederates. The limits of our sample size were one of the reasons to focus on a small number of neuroimaging analyses at the core of our conceptual questions, and leveraged neural signature analyses, where possible, to perform some of the critical statistical inferences on univariate measures (the so called signature expression values), for which $n=25$ is adequately powered.

A power analysis shows that with 25 participants, we would be well-powered to find a robust association ($\rho=0.5$) for our signature analysis: using the software *g*power*, with $\rho=0.5$, $p=0.05$, $n=25$, $\text{power}=0.87$. Such a power analysis also shows that any effect size $\rho>0.3$ would be detected in at least 50% of cases. In Bayesian terms, the data we observed, also provides evidence of absence, in showing that the data has $\text{BF}_{10}<1/3$, with the observed pattern of AVPS loading 3.9 times more likely under the null-hypothesis of no association with *wf* than under the alternative hypothesis of an association. This shows that although our data cannot exclude the presence of a weak association ($\rho<0.3$), we provide evidence against the presence of a robust association, which we would be more likely to miss than to detect. Accordingly, we specify these considerations in our fMRI results by specifying in Figure 8 legend:

“It should be noted, that with our sample size ($n=25$), we are adequately powered to detect strong associations with *wf* ($\rho=0.5$, $\alpha=0.05$, $\text{power}=87\%$) but cannot exclude weaker associations for which we were underpowered ($\rho=0.3$, $\alpha=0.05$, $\text{power}=45\%$), as 80% power for such associations would require over 60 participants”

Exhaustive whole-brain, mass univariate analysis of all the possible epochs of our task where not our aim, and those indeed suffer from conservative corrections for multiple comparisons that lack sensitivity with our sample-size.

In Supplementary Materials §17, where we additionally verify the power to detect core effects in our sample size using simulations.

17. PE_M and PE_S separability

The average correlation between the time courses of the parametric modulators for PE_S and PE_M was -0.26, ranging from -0.49 to -0.03. Due to this correlation, we explored whether our experimental design and GLM approach can disentangle voxels that represent PE_S from those representing PE_M , and whether they can differentiate voxels linearly dependent on wf from those that are not. Our GLM included, during the outcome period, a boxcar for the duration of the movie with two parametric modulators, one for PE_S and one for PE_M . Both have been normalized by dividing them with $1-wf$ and wf respectively. This was done, as described in the Methods and Materials section of the main manuscript, to ensure that PE_S and PE_M predictors become independent of preference and wf per se. When used in the GLM, the parameter estimates for these normalized PE_S values can then be compared across participants to identify if the brains of participants with higher weight on shocks (i.e. larger value for $1-wf$) show larger signals for a given outcome than participants with lower weight on shocks. Using the original PE_S values would make that interpretation difficult, because they are already dependent on wf .

For this parameter recovery, we ran 1000 simulations. In each, we simulated 25 participants. For each participant, we used their own design matrices (the same used for the actual GLM first level analysis of the fMRI activity after convolution with the haemodynamic response function) to mix signals in each subject using three mixings (i) $-1 \cdot PE_S + 0 \cdot PE_M + \text{noise}$; (ii) $0 \cdot PE_S + 1 \cdot PE_M + \text{noise}$, and (iii) $-1 \cdot PE_S + 1 \cdot PE_M + \text{noise}$. Noise was a random gaussian set at 1std of the mixed signal. Next, we ran a GLM using the same design matrix, and saved the parameter estimates for PE_S (we will call βPE_S) and PE_M (we will call βPE_M) for each participant. We then perform a t-test for βPE_S and one for βPE_M to see if across the 25 parameter estimates (one per participant) there is evidence against the null hypothesis $H_0: \beta PE_S = 0$ or $H_0: \beta PE_M = 0$. Of course, if PE_S was mixed into the voxels activity (case i or iii), a significant t-test would be a hit, while a non-significant t-test would be a miss, and the same applies to PE_M for case ii. After repeating this procedure 1000 times, we count the proportion of the 1000 simulations where a t-test was significant against $H_0: \beta PE_S = 0$ or $\beta PE_M = 0$. Additionally, to see how often the analysis falsely detects a dependence on wf although wf was not included in the mixing, we also look at $r(wf, \beta PE_S) = 0$ and $r(wf, \beta PE_M) = 0$. Initially, we use $p < 0.05$ as a criterion, to look at the specificity and sensitivity for the case in which we explore responses in the AVPS, which is univariate. We also indicate proportions at $p < 0.001$, but this time for a one-tailed test, in parenthesis, to provide results relevant for an explorative whole brain analysis where the cluster-cutting threshold was set at 0.001. The proportion of significant results was as follows:

H_0	mixing	$-1 \cdot PE_S + 0 \cdot PE_M$	$0 \cdot PE_S + 1 \cdot PE_M$	$-1 \cdot PE_S + 1 \cdot PE_M$
$\beta PE_S = 0$		99.8(92.4)	4.4(0)	99.8(90.6)
$\beta PE_M = 0$		5.5(0.1)	99.8(90.8)	99.5(91.9)
$r(wf, \beta PE_S) = 0$		5.2(0.7)	5.3(0.1)	5.4(0.4)
$r(wf, \beta PE_M) = 0$		4.5(0.2)	5.1(0.1)	3.8(0.0)

Supplementary Table 17. Frequentist tests for simulations without wf dependence.

Percentage of two-tailed t-tests significant at $p < 0.05$ from the 1000 simulations using signals generated without multiplications with wf or $(1-wf)$, and in brackets, the percentage of one tailed $p_{1\text{-tailed}} < 0.001$. The top row specifies how the signals were generated before adding 1std of noise, the leftmost column, the null hypothesis that was tested in the hypothesis testing.

We then repeated the same analysis, but this time multiplying the signals with $(1-wf)$ and wf as indicated in Supplementary Table 13 to simulate cases of voxels where signal strength depends on preference.

H₀ mixing	-1·(1-wf)·PE_S+0·wf·PE_M	0·(1-wf)·PE_S+1·wf·PE_M	-1·(1-wf)·PE_S+1·wf·PE_M
βPE_S=0	99.3(81.1)	3.5(0.3)	99(80.5)
βPE_M=0	3.8(0.2)	98.5(64.8)	98.7(62.3)
r(wf, βPE_S)=0	75.9(26.4)	5.1(0.0)	74.6(26.5)
r(wf, βPE_M)=0	5(0.0)	98.3(77.3)	98.5(79.3)

Supplementary Table 18. Frequentist tests with or without wf dependence.

Percentage of two-tailed t-tests significant at $p < 0.05$ from the 1000 simulations using signals generated with multiplications with wf or $(1-wf)$, and in brackets, the percentage of $p_{1-tailed} < 0.001$. The top row specifies how the signals were generated before adding 1sd of noise, the leftmost column, the null hypothesis that was tested in the hypothesis testing.

The above tables explore evidence against the null hypothesis, but for univariate analysis we also ask whether we can actually provide evidence for voxels mixed without a certain factor that the GLM provides evidence for the null hypothesis using Bayesian statistics⁵⁷, using a bound of $BF_{10} < 1/3$. Using a Bayesian test, with $n=25$, we know that $|t| < 1$ provides evidence in favour of $H_0: \beta PE_S = 0$ over $H_1: \beta PE_S \neq 0$, and $|r| < 0.17$ for $H_0: r(wf, \beta PE_S) = 0$ over $H_1: r(wf, \beta PE_S) \neq 0$ ($BF_{10} < 1/3$, using default priors in JASP). We thus counted the proportion with evidence in favour of H_0 over H_1 in all cases using these bounds.

H₀ mixing	-1·PE_S+0·PE_M	0·PE_S+1·PE_M	-1·PE_S+1·PE_M
βPE_S=0	0	69.6	0
βPE_M=0	66	0	0
r(wf, βPE_S)=0	57.1	58.5	59.3
r(wf, βPE_M)=0	58.0	56.8	58.4

Supplementary Table 19. Bayesian tests for simulations without wf dependence

Percentage of $BF_{10} < 1/3$ from the 1000 simulations using signals generated without multiplications with wf or $(1-wf)$. The top row specifies how the signals were generated before adding 1sd of noise, the leftmost column, the null hypothesis that was tested in the hypothesis testing.

H₀ mixing	-1·(1-wf)·PE_S+0·wf·PE_M	0·(1-wf)·PE_S+1·wf·PE_M	-1·(1-wf)·PE_S+1·wf·PE_M
βPE_S=0	0	70.1	0
βPE_M=0	68.1	0.1	0
r(wf, βPE_S)=0	2.0	57.5	3.6
r(wf, βPE_M)=0	57.5	0.0	0.0

Supplementary Table 20. Bayesian tests with or without wf dependence.

Percentage of $BF_{10} < 1/3$ from the 1000 simulations using signals generated with multiplications with wf or $(1-wf)$. The top row specifies how the signals were generated before adding 1sd of noise, the leftmost column, the null hypothesis that was tested in the hypothesis testing.

Summary: In our simulations, with 1sd of noise, we can detect voxels with signals linearly dependent on PE_S and/or PE_M accurately: If we use $\alpha = 0.05$, as we would for the AVPS analysis, signals generated by including PE_S but not PE_M are detected as representing PE_S in ~99% of cases, and only in ~5% of cases as representing PE_M , and vice versa for voxels

generated to include PE_M but not PE_S . Using Bayesian statistics, we can even provide evidence in favor of the H_0 for the former ($\beta_{PE_S}=0$) in $\sim 70\%$ of cases, and the latter H_0 ($\beta_{PE_M}=0$) in 66% of cases. Within our sample size, we thus have power to arrive at conclusions that match the way we generated the signals in the majority of simulations. Even when $p < 0.001$ is used, as it would for our exploratory whole brain analysis, power remains decent.

With regard to linear dependence on wf , we find that for voxels generated with PE_S but not PE_M signals, if the signal was generated using $(1-wf)$ as a multiplier, a significant correlation is detected in 76% of cases, and when not used in the generation, a significant correlation is found in 5% of cases, while evidence for $H_0: r(wf, \beta_{PE_S})=0$ is found 57% of cases.

Major:

R1C1 [Framing surrounding model-free and model-based learning]

The researchers frame their experiment as distinguishing between model-free vs. model-based learning by the kinds of computational model that they compare. However, to distinguish model-free and model-based learning they would need a behavioural paradigm that favours different strategies to be adopted, such as the two-step task by Daw and colleagues or subsequent modifications suggested by Kool and colleagues. From the current task it is not possible to distinguish between these possibilities. Their weighting factor does not capture the relative reliance on model-free and model-based learning, which is important to make inferences about the use of model-free and model-based behaviour.

We thank the reviewer for this comment, which is also echoed by the first comment of reviewer 2 (see R2C1), who suggests *“the authors could also focus solely on the question whether financial versus moral features are learned separately and how they are integrated; this is an interesting issue that does not necessarily need to be related to the debate about model-based vs. model-free learning.”*

Model-based and model-free learning has been used by some with meanings that are highly overlapping with goal-directed and habitual, under the assumption that model-based learning leads to goal-directed behaviors, and model-free learning to habits, and that they can therefore be dissociated by devaluation. Gillan, Otto, Phelps and Daw for instance write *“It has been proposed, though so far largely on theoretical grounds, that these two frameworks coincide: Specifically, that model-free and model-based learning (respectively) give rise to habits and goal-directed actions, as operationalized by devaluation sensitivity”* and conclude that *“Across both experiments, we found that the extent to which actions were learned using model-based updating during instrumental training predicted their later sensitivity to devaluation”*. This had tempted us to connect to this debate with our data, but we agree that doing so is actually distracting and indirect at best, and our data is better framed in terms of asking whether participants' choices are better explained by a process that results in separable representations of the association between a given symbol and self-money and other-shock outcomes, and whether these representations are biased towards the outcome that choices reveal to have more weight in the decision. This was also the way we originally framed it in the grant application that financed this work. We therefore removed the use of model-based and model-free entirely from the introduction and the results, and instead reworded our framing in terms of separability and bias of the learned symbol-money and symbol-shock representations. We only link to the distinction model-based and model-free in one section of the discussion that reads:

“Some have found that learners that respond flexibly to devaluation, as most of ours did, which indicates goal-directed behavior instead of habit, are more likely to have used a model-based form of learning (Dolan and Dayan, 2013; Gillan et

al., 2015). Others found participants' choices to be dominated by model-free learning when preventing shocks to others in two-choice Markov decision tasks (Lockwood et al., 2020). It may thus be that participants represent the nature of outcomes (as revealed by our devaluation study) while avoiding the cognitively expensive decision-tree necessary to ascribe rewards following a rare transition in a two-choice Markov decision task to an alternative (nonselected) action."

We think this revised framing cuts more effectively to the core of what is central to conflictual moral learning, but had to our knowledge so far never been explored: how participants deal with the intrinsic duality of outcomes when learning under conflictual moral situations.

Our introduction now reads:

"We often have to learn that certain actions lead to favorable outcomes for us, but harm others, while alternative actions are less favorable for us but avoid or mitigate harms to others¹. Much is already known about the brain structures involved in making moral choices when the relevant action-outcome contingencies are known²⁻⁹, but how we *learn* these contingencies remains poorly understood, especially in situations pitting gains to self against losses for others.

Reinforcement learning theory (RLT) has successfully described how individuals learn to benefit themselves^{10,11} and most recently, how they learn to benefit others¹²⁻¹⁵. At the core of reinforcement learning is the notion that we update expected values (EV) of actions via prediction errors (PE) – the differences between actual outcomes and expected values. Ambiguity in morally relevant action-outcome associations raises specific questions with regard to RLT, especially if outcomes for self and others conflict. If actions benefit the self and harm others, are these conflicting outcomes combined into a common valuational representation; or do we track separate expectations for benefits to the self and harm to others¹⁶? Also, people differ in how they represent benefits and harms to self¹⁷, and in whether they prefer to maximize benefits for the self vs. minimize harms to others^{3,4,6}. How can such differences be computationally represented using RLT? Would people maximizing benefits for the self show reduced prediction errors and expected value signals for other-harm, as motivated accounts of empathy may suggest^{41,42}, or are expectations tracked independently of one's preferences, such that preferences only play out when decisions are made? These important questions can only be addressed by studying the dynamics and neural underpinnings of action-outcome associations while outcomes for self and others conflict.

To address these questions, we combined online behavioral and fMRI data from two independent studies. In the core task (Conflict condition, Figure 1a,b; Table 1), common to both experiments, participants had to learn that one of two symbols led to high monetary gains for the self 80% of the time, and to a painful but tolerable shock to the hand of a confederate with the same probability. We refer to this symbol as 'lucrative', since it was associated with higher monetary outcomes. The other symbol led to low monetary gains for the self 80% of the time, and to lower intensity, non-painful shocks to the confederate with the same probability. We refer to this symbol as 'considerate'. At the beginning of each block, participants did not know the associations between symbols and outcomes. Choosing which symbol best satisfies the moral values that participants act upon in the task thus requires learning to predict the outcomes associated with each symbol.

In the Online experiment, participants performed a number of additional tasks to explore whether they learned the symbol-outcome association probabilities for both the self-money and other-shocks, even if their preferences may prioritize one. First, we added blocks in which we removed the conflict. In this NoConflict condition (Figure 1c), the symbol that led to high money in 80% of cases also led to low shocks in 80% of the cases. Second, after 1/3 of the Conflict and NoConflict blocks, we explicitly asked participants to report the learned associations. Only if participants learn symbol-outcome association for the less preferred outcome, should they report different probabilities under the Conflict and NoConflict for this outcome. Third, we also leveraged the devaluation approach pioneered in animals¹⁶ by adding blocks (Dropout blocks, Figure 1b,c) in which after 10 trials to learn the symbol-outcome associations, we informed participants that the self-money (MoneyDropout) or the other-shocks (ShockDropout) would not be delivered on the following 10 trials. We then examined the choice on the 11th trial, before participants witnessed the modified outcome. If participants track separable representations for self-money and other-shocks, we expect them to show different choices depending on which outcome is removed in ConflictDropout blocks, with choices expected to change substantially, if the outcome they weigh more is removed. Such changes should not occur in the NoConflictDropout blocks. If participants track a single, combined value for each option, based on the history of past experienced values, we expect them to continue choosing their previously favored option in either Conflict and NoConflict blocks. We then used Hierarchical Bayesian Model comparisons to test which computational formulations of the RLT models better describe participant's choices, and gain insights into how people combine the two outcomes in their morally relevant learning experience.

When a child's selfish actions cause pain to a sibling, parents intuitively resort to drawing the child's attention to the distressed facial expressions of the victim. To optimize our study to capture how such sights become a learning signal we (i) made shocks visible to the participants through pre-recorded videos showing facial expressions from the confederate, instead of the symbolic feedback more often used in neuroeconomic paradigms (Supplementary Material §1); and (ii) collected and analyzed neuroimaging data with a focus on how the brain updates values when learning from the facial reaction of others. Given the extensive literature on empathy for pain, we expect networks involved in processing the painful facial expressions of others¹⁸⁻²¹, as captured by the affective vicarious brain signature (AVPS²²), to have BOLD signals that covary with learning-relevant signals such as the prediction errors for shocks (PE_S). Given that the ventromedial prefrontal cortex (vmPFC) is known to have BOLD signals that covary positively with the current value of multiple outcomes, in particular for chosen options^{23,24}, and that outcomes for others appear to also be encoded in this region^{25,26}, we also expect the vmPFC to have such learning-relevant signals regarding shocks to others. Finally, given the involvement of financial rewards in our task, we expect the reward circuitry, as captured by a neural reward signature²⁷, to have signals that covary with prediction errors at least for money (PE_M). Whether signals in either network or region would be stronger in participants with a stronger preference for reducing shocks to others remains unexplored, and will be a key question.

Using these approaches, here we show that participants vary substantially in whether they choose to maximize self-money or minimize other-shocks. Their choices are best described by a reinforcement learning model separately tracking the values of these self-money and other-shocks. Importantly, we find that individual differences are best captured by including an individual valuation parameter that biases expected values towards the outcome that bears more weight in the decision-making. Signals in the ventromedial prefrontal cortex reflect this bias, while the pain-observation network represents pain prediction errors independently of these individual preferences."

R1C2 [Low power for individual trial analysis]

There is a concern that very strong inferences are made regarding trial 11 and whether there is enough power to compare performance on an individual trial across participants. Even if there are multiple repetitions of that trial could participants come to expect it over blocks? If the question is how people weigh up money and shocks differently when deciding for others it would seem better to use a decision-making task without any learning component.

- **Power at the 11th trial**

We thank the reviewer for encouraging us to elaborate on the power we have to dissociate our three models based on the results from the 11th trial in Dropout blocks.

It should be noted that the utility of the 11th trial is not to determine how people weigh up money and shocks: that question is captured by the weighting factor wf , which is estimated exclusively using the first 10 trials in which conflict occurs. The utility of the 11th trial, in the tradition of devaluation studies, is to adjudicate between the three alternative learning models that perform equally well during the first 10 trials of conflict, and ask: do participants keep track of a single joint value (M1), or do they separately track the expected values for shocks and money (M2); and if the latter, do they track both expected values equally well, with preferences only playing out at the level of decisions (M2Dec), or do preferences already play out at the outcome stage, such that the less considered outcome is associated with smaller expected values (M2Out). These learning-specific questions cannot be arbitrated using a simple decision task which does not involve learning.

Instead, we arbitrate across these 3 models in two complementary ways: by examining explicit reports on the probability of each symbol to trigger a particular money and shock outcome, and by looking at choices following devaluation of one of the outcomes. The question of power that the reviewer encourages us to elaborate on is presented separately for these two approaches below.

Power of the explicit report: For the explicit reports, the critical questions are two-fold: (i) do participants have access to separate probabilities for money and shocks and (ii) does the accuracy of

their report depend on their preference. (i) is tested using the ability of the participants to provide estimates of probabilities separately for money and shock, and report that the Conflict and NoConflict conditions differ in the likelihood of the high-shock symbol to lead to high-money (Fig. 3a-d, also reported below). (ii) is tested most powerfully via the correlation between the bias and the preference (Fig. 3e). With regard to statistical power, (i) depends on a within-subject comparison of two mean probabilities, with $n=79$ when considering all participants or $n=24$ (the size of the smallest group), when considering each group separately. This yields a sensitivity (considering a one-tailed test, $\alpha=0.05$, $\text{power}=80\%$) to detect even a small difference in means (Cohen's $d=0.28$) for the whole group or $d=0.52$ within each sub-group. (ii) depends on a correlation between preference and bias in the whole group of $n=79$, and a power-analysis reveals that this provides the sensitivity to detect even a small association $\rho=0.27$. Hence, our group-size provides the required sensitivity to detect even small effects in our data.

Figure 3. Participants' report bias. **a** Participant reports of perceived high-outcome probability for each symbol in the Conflict condition. **b** Same for the NoConflict conditions. The x-axis specifies which symbol and the probed question: short/tall pile of money=reported high-money probability for the symbol actually associated with low/high-money; painless/painful face=reported high-shock probability for the symbol associated with low/high-shock. As a group, most participants correctly assigned higher probabilities to symbols that had higher probability, and reported a different pattern of probabilities after Conflict and NoConflict blocks. Thick black lines = average of reported probabilities; dotted black lines = programmed/expected probability. Square brackets below the graphs in A and B indicate the direction of the difference computed in c and d. **c** Difference between reported probabilities for the low-shock minus high-shock symbol, separately for the Considerate (green), Lucrative (blue) and Ambiguous (orange) preference group, and for the Conflict condition. The yellow inlet in the bottom right corner illustrates how the differences in panels c,d,e are calculated. **d** Same for the NoConflict condition. Violin plots represent the distribution, the box-plot within, the median and quartiles. The BF_{10} and p-values above a violin represents the result of a Wilcoxon signed rank test of the differences vs. zero. The BF and p-values on the dotted green and blue lines indicate the result of a Bayesian Wilcoxon signed rank test comparing the difference for money and shock. The BF and p-values below a pair of violins represent the result of a Wilcoxon signed rank test comparing the Conflict and NoConflict conditions. All statistical values are presented in Supplementary Table 4,5. Dotted black lines = actual probability difference ($80\%-20\%=60\%$) as programmed in the task. **e** Correlation between the participant's report bias and

the proportion of considerate choices during the Conflict task, with $\text{Bias} = (rpS(\text{high-shock}) - rpS(\text{low-shock})) - ((rpM(\text{high-money}) - rpM(\text{low-money})))$, where rpS/rpM stand for reported probability of high-Shock/high-Money outcome, and high-shock, low-shock, high-money, low-money refer to the symbols based on their actually most likely outcome.

Power of the choices at the 11th trial: For the choices on the 11th trial, traditional power-analysis software (e.g. G*power) does not cater for situations in which each participant provides a small number of binary choices. We therefore performed simulations to assess our power to differentiate the predictions of the three models (M1, M2Out, M2Dec) within our critical subgroup (e.g. the N=29 Considerate participants or the N=26 Lucrative participants). Specifically, our Bayesian estimation in Stan provides, for each participant, the predicted probability for a pain reducing choice according to the 3 models on the 11th trial of the 4 blocks of a particular dropout type (see Fig. 5e). We thus took these subject and block-specific predicted probabilities and generated the 4 choices based on these probabilities, and then performed the same Wilcoxon signed rank test we use in the actual analysis to check whether the predicted behavior, with the number of pain-reducing choices for each participant j ($x_j \in \{0,1,2,3,4\}$) for N participants, suffices to significantly ($p < 0.05$) discriminate M2Out from M1 or M2Out from M2Dec in terms of predicted behavior. We repeat this 1000 times, and the proportion of significant tests is then the equivalent of our power to discriminate the models based on only 1 trial (i.e. the 11th trial) x 4 blocks x N participants. Results indicate that we get at least 80% power to discriminate the predicted behavior for the critical cases: for participants with considerate preferences when the shocks are removed, M1 vs M2Out, $p < 0.05$ in 100% of simulations, M2Dec vs M2Out, $p < 0.05$ in 96% of simulations; and for participants with lucrative preferences when the money is removed M1 vs M2Out, $p < 0.05$ in 98% of simulation, and M2Dec vs M2Out, $p < 0.05$ in 81% of simulations. The script can be found on OSF (<https://doi.org/10.17605/OSF.IO/RK8W4>) in the directory 'ReplyToReviewer'. Hence, we have sufficient trials and participants to differentiate the predictions of our three models in >80% of cases.

The most compelling evidence for our power to adjudicate the competing models however comes from Fig. 5f (see below). While the above power analysis by simulation only considers a subgroup of participants at a time, and only some of the Dropout conditions at a time, Fig. 5f considers all the available data. Specifically, in this figure, we show the result of an analysis performed in Stan, in which the three models were fitted using the data from the first 10 trials only. With the fitted parameters, the model then generated 4000 predictions for the choices on the 11th trial, using a fully Bayesian model (i.e. using the posterior distribution of the parameters rather than maximum likelihood point estimates), by sampling from the posterior distributions, for each of the 3 models and for all 79 participants, including the ambiguous participants, for all 11th trials (i.e. 4 trials x 4 dropout conditions). The plot then shows the distribution of the log likelihood of the actual measured choices for each of these 4000 posterior draws (i.e. predictions of the behavior given the 3 models). That these distributions are so clearly different shows that we have the power to adjudicate across our models with the data we have: there is clear separation in how likely the measured choice data on the 11th trial is based on the winning model M2Out (indigo), compared to the competing M2Dec (turquoise) and M1 (gray). Had the number of trials been insufficient to clearly differentiate the predictions of these models, these distributions would have overlapped considerably. Instead, the likelihood of the data under these models shows little overlap, providing the most complete proof that we have the power to discriminate between the models.

In the manuscript we now added a sentence referring to Fig. 5f to tackle the question of power:

“Importantly, that the distribution of likelihood of the observed data given M2Out (indigo, Figure 5f) does not overlap with that given M2Dec (turquoise) or M1 (gray) shows that our approach focusing on the 11th trial in 4 blocks is sufficiently powered to adjudicate between our candidate models: a given choice pattern uniquely more likely under one model than the others.”

Figure 5. Model Comparison. **f** Distribution over 4000 posterior draws of the summed log likelihood of the 11th trial over all participants multiplied by -2 to place values on the information criterion scale as for LOOIC. M2Out outperformed all other models. We use LOOIC in d, because these first 10 trials were included in the fitting of the model, but log-likelihood in f because it is not included in the model fit.

Overall, we hope that these considerations assuage the reviewer’s concerns and show that we have sufficient power to adjudicate our models using those single trials in 4 blocks across our subjects. Because the manuscript is already quite rich, we felt inclined not to include the simulations in the

manuscript itself.

Finally, as we’ll detail below, when it comes to gender, we have since acquired additional online data. We thus now have a total of N=235 online participants, and M2Out outperforms the other models also in this larger sample (see reviewer figure below). Because these additional participants will serve another publication focusing specifically on gender, we only mention this here for the reviewer as evidence for the robustness of our finding.

Reviewer figure. Model comparison on 235 participants. Paired Wilcoxon Signed-Ranked test between the ln(likelihood) at trial 11 of the three models.

● **Participant’s expectations at the 11th trial**

Beyond the above considerations on power, the reviewer raises an issue regarding participant expectations that we fully agree with. To adjudicate the different forms of learning, we measure the behavior on multiple Dropout trials, and this measurement does create an expectation that Dropout trials can occur. *Which* dropout trial will occur next is however unpredictable: Given equal probability of Conflict and NoConflict blocks and of Money- and ShockDropout conditions, participants cannot predict which of the different forms of Dropout will occur in a given block, and therefore cannot anticipate the best way to react to the Dropout: a default strategy, e.g. “switch to the other symbol when my favorite outcome is removed”, cannot explain the observed behavior because in half the trials, Dropout occurs in a NoConflict condition, where participants need not to switch if their favorite outcome is removed, and Supplementary Fig. 6 (also reported below) shows that participants display different behavior based on this Conflict/NoConflict difference. To perform the task as they do, participants therefore must have separable representations of Money and Shock outcomes, and need to track both probabilities. Nevertheless, to

have sufficient power to differentiate the models, we had to present the Dropout conditions multiple times, and participants therefore could expect Dropouts to occur. This might have encouraged them to track probabilities of their less favored outcome more than they would have if Dropouts never occur. This is why in the limitations we noted:

“However, we must consider the possibility that these very trials also influenced participants to separate their representations for self-money and other-shocks to enable more optimal decisions during drop-out trials, and more accurate probability reports. Performing a conflict task with only a single devaluation trial on a large number of participants may be a way to exclude this possibility. However, if participants were to have adapted their strategy to optimally fit the requirements of the task, M2Dec would have been even more adaptive”.

a NoConflict condition

c NoConflict condition first 10 trials

b NoConflict condition

d Conflict condition

Supplementary Figure 6. Participant's choices in the Dropout blocks. **a** For the NoConflict condition, the average \pm sem considerate choices as a function of trial, separately for the three preference groups. Note that preference grouping is based on all NoConflict trials, i.e. including the NoDropout blocks, while the averages of proportion of considerate choices only include choices on the Dropout trials. Here we also show choices on trials 12-20 after Dropout. These choices are less relevant to the main paper, because they capture what is probably a different form of learning, when symbols are only associated with 1 outcome rather than 2 outcomes that can conflict. **b** Averaged NoConflict choices at the 10th and 11th trials, separately for condition and group. Numbers on top of box plots indicate the Wilcoxon signed-rank values from comparing choices at the 10th and 11th trials, BF_{10} and p-values. $BF_{10} > 3$, evidence for the presence of a difference, and $BF_{10} < \frac{1}{3}$ evidence against. **c** Proportion of favorable choices (i.e. choosing the symbol most likely to lead to high-money and low-shock) for the first 10 trials of the NoConflictDropout condition, separately for the three groups. Bayesian one way ANOVA, $F(2,76)=15.6$, $p < 0.001$; $BF_{incl\ group}=8284.52$. Red dotted line: learning threshold determined using binomial distribution (71/120 correct choices). 54% of the ambiguous preference subjects fell below this learning threshold, while just 7% of those with considerate and 4% of those with lucrative preference were below this threshold. **d** Same as in **a** and in Figure 4a, but for the Conflict condition, with the purpose to illustrate choices up to trial 20.

R1C3 [reward]

3. Simply because studies have already investigated neural responses to reward in the self this does not appear a convincing justification to exclude inferences and analysis of these responses neurally, as the task manipulates both rewards and shocks.

The reasons why we focused on shocks are driven by our lab's mission: the project originally developed around the specific question of whether regions involved in sharing the pain of others are necessary to teach people not to hurt others, in particular in cases in which hurting others is a means to gain something. To address this question we planned a series of experiments each addressing specific sub-questions and steps, such as identifying the model best describing participant's learning behavior, identify regions involved in the processing of the outcome and PE for shock, and establishing causality between these regions and behavior. While writing our first important steps toward our final aim, we remained focused toward our main overarching goal to identify how the pain of others influences our choices in conditions of conflict with our own goals. We recognize though that for a reader that only has access to these first steps, our focus on shock might be surprising.

Following the reviewer's comment, we now also present inferences and analyses on reward. In particular, we took advantage of datasets previously collected by the author S. Speer and additional open access datasets on economic paradigms meant at identifying the circuit of economical reward and losses, to build a reward signature for reward. This work was inspired by the vicarious pain signatures we use to test the involvement of regions processing pain. To avoid overloading the current manuscript with additional datasets, we decided to prepare a separate publication on the reward signature. A detailed description of our work leading to the reward signature can therefore be found on bioRxiv at the following link: <https://doi.org/10.1101/2022.06.16.496388>.

The new analyses on reward are now presented in the manuscript. Below the relevant manuscript extracts in which we now include analyses on reward and reward prediction error.

In the introduction we write:

“Finally, given the involvement of financial rewards in our task, we expect the reward circuitry, as captured by a neural reward signature (Speer et al., 2022), to have signals that covary with prediction errors at least for money (PEM). Whether signals in either network or region would be stronger in participants with a stronger preference for reducing shocks to others remains unexplored, and will be a key question.”

In the result session we present the following:

“With regard to the reward signature, we found PES and PEM to load positively on RS (Figure 8c,f; Supplementary Table 9) showing that receiving less intense than expected shocks or higher than expected monetary rewards both triggered a pattern of activity typical of receiving a reward. Examining whether the loading on RS depended on wf led to inconclusive evidence leaning in favor of absence (Figure 8d; Supplementary Table 9).”

Figure 8. Loading on AVPS and RS. **a** Loading of the normalized parameter estimate images (β_{PE}) for PE_S and PE_M on the AVPS²². Each dot is one participant, the horizontal line is the mean. p and BF values reflect a one-sample parametric (t -test against zero in JASP) t -test against zero in JASP. **b** Loading of the parameter estimate for PE_S (β_{PE_S}) and PE_M (β_{PE_M}) onto the AVPS as a function of wf . p and BF values represent the outcomes of a parametric correlation analysis using Pearson's r . **c-d** Same as in a-b, but for the RS. It should be noted, that with our sample size ($n=25$), we are adequately powered to detect strong associations with wf ($\rho=0.5, \alpha=0.05, \text{power}=87\%$) but cannot exclude weaker associations for which we were underpowered ($\rho=0.3, \alpha=0.05, \text{power}=45\%$), as 80% power for such associations would require over 60 participants. **e** Voxels contributing substantially to the loading of β_{PE_S} (top) and β_{PE_M} (bottom) onto AVPS. Because the overall loading is calculated by first

multiplying each voxel's β_{PE_S} value with the value in this voxel of AVPS, and then summing these multiplication over all voxels, here we simply calculated the first step (the voxelwise multiplication of β_{PE_S} and AVPS, and β_{PE_M} and AVPS), and averaged the result across our participants. To not overcrowd the image, we only show voxels with values above 50 or below -50 after multiplying the images by 10^6 . **f** Same as in e, but for the RS.

[...]

“The same voxel-wise analysis applied to PEM only generates significant results when the cluster-cutting threshold was reduced to $p_{unc} < 0.01$ (Figure 7c), and reveals striatal and ventral prefrontal clusters in line with those described in the literature for $PE_M > 0$ after removing the variance explained by wf (Bartra et al., 2013; Fouragnan et al., 2018; Rushworth and Behrens, 2008), and clusters in the right cerebellum, ventral temporal lobe and hippocampus with PE_M signals that depend on wf (Figure 7d). Comparing the networks associated with PE_S and PE_M reveals that these networks are largely non-overlapping (Figure 7c). However, inclusively masking the contrast $PE_M > 0$ with the contrast $PE_S > 0$ reveals five clusters of overlap, the largest of which was in the vmPFC

(Supplementary Figure 17), but these clusters remained too small to survive a whole brain FWE cluster-size correction.”

Figure 7. Key fMRI results. **a** Voxels where BOLD signals are increased during the outcome phase, independently of prediction errors, ($\text{Outcomes} > 0$), $p_{\text{unc}} < 0.001$, $k = \text{FWEc} = 903$ voxels (Supplementary Table 8). **b** Red: regression between PE_S and $1-wf$ ($p_{\text{unc}} < 0.001$, $k = \text{FWEc} = 181$ voxels, Supplementary Table 10). This identifies voxels with signals that increase more for shock outcomes that are less intense than expected (i.e. positive PE_S) in participants that place more weights on shocks (high $1-wf$ values). Yellow: results of the contrast $\text{constant} > 0$ in the same linear regression with $(1-wf)$ ($p_{\text{unc}} < 0.001$, $k = \text{FWEc} = 167$ voxels). This identifies voxels with signals that increase with increasing PE_S after the variance explained by $(1-wf)$ is removed (Supplementary Table 11). **c** Cyan: $\text{PE}_M > 0$ (after removing variance explained by wf ; Supplementary Table 12) at $p_{\text{unc}} < 0.01$, $k = \text{FWEc} = 542$ voxels. Purple: $\text{PE}_S > 0$ presented the same cluster-cutting threshold of $p_{\text{unc}} < 0.01$, $k = \text{FWEc} = 950$ voxels (Supplementary Table 11) in order to visualize that the two networks are largely distinct (but see Supplementary Figure 17 for overlap). **d** Correlation between PE_M and wf , $p_{\text{unc}} < 0.01$, $k = \text{FWEc} = 1642$ voxels (Supplementary Table 13). **e** Results of the contrast $\text{PE}_S * \text{LR}_s > 0$, $p_{\text{unc}} < 0.001$, $k = \text{FWEc} = 121$ voxels (Supplementary Table 14), which represent the BOLD signal associated with shock value updating. Note that across all panels, results are FWE cluster corrected at $\alpha < 0.05$ using (i) cluster

cutting at $p_{\text{unc}} < 0.001$ (panels a, b, e) or (ii) $p_{\text{unc}} < 0.01$ (panel c, d) then excluding clusters with an extent below the critical FWE cluster size FWEc . Cluster-extent threshold size always indicated in figure panels. Renders are created based on the cortex_20484 surface from spm12, slices are taken from the average T1 anatomical scan from our participants.

Supplementary Figure 17. Overlap between PEM and PES. Cluster of activity obtained by inclusively masking PEM (after the variance explained by wf is removed) inclusively masked with PES (after the variance explained by (1-wf) is removed) at punc<0.01 without FWEc correction.

R1C4 [consistency in statistical annotations]

Throughout the paper sometimes Bayesian statistics are used and sometimes p values. I understand the wish to use Bayesian stats to draw inference on the strength of the null, but it would be important to qualify the strength of evidence, and consistently add p values when there are bayes factors.

We now added frequentist p values where BF values are mentioned. In some analysis, BF values however remain unavailable (e.g. mixed linear models).

R1C5 [Sample]

There are quite large gender and age differences between the online study and the fMRI study. The fMRI participants are significantly older and all female compared to the online study. I understand that the researchers suggest no difference overall in choices between the two groups but the bayes evidence is not very strong. There are several studies showing age differences in reinforcement learning tasks. More generally, an fMRI sample size of 25 when there are quite large individual differences is underpowered. There is a section on power analysis but this seemed to run parameter recovery on imaging signals rather than to show the computational model could reproduce behaviour on the task.

We thank the reviewer for having invited us to ascertain our premise that M2Out is appropriate even considering gender and age differences across studies.

The aim of our study was more generally to shed light on the following question “Would people maximizing benefits for the self show reduced prediction errors and expected value signals for other-harm, as motivated accounts of empathy may suggest^{17,18}, or are expectations tracked independently of one’s preferences, such that preferences only play out when decisions are made?”

and for our fMRI study more specifically “Whether signals in either network or region would be stronger in participants with a stronger preference for reducing shocks to others remains unexplored, and will be a key question.”

It is therefore important to note, that the premise for our combination of more extensive behavioral testing in the online study with the fMRI study is not that the participants in the two studies have the same distribution of preferences between self-money and other-shocks, but rather, that the same computational model can describe their choices. This is because the fMRI study does not include the Dropout trials that allow us to determine which of the models best describe their learning strategy, and we must thus use a model chosen in the online study to extract the hidden variables used in the fMRI analysis. For our

approach, and to address the reviewer's concern, it is thus important to determine whether (i) M2Out remains the winning model even when only considering the female sub-sample of our Online study, and (ii) a sample of female participants with an age composition that does not differ significantly from that in the fMRI study would still yield evidence in favor of M2Out outperforming M2Dec and M1.

• Age differences

Given that our initial Online sample differed in age from the fMRI sample, to address the reviewer's concern, we collected an additional sample of women at the higher age-end of our fMRI age sample (N=22; lighter green dots in Supplementary Figure 9a, reported below), so that by combining the females from the Online study already in the manuscript with this newer sample, we now have a group of female participant that no longer differs in age from that in the fMRI sample. We then compared the log likelihood of the 11th trial for the three models in this combined online sample that is gender and age-matched, and found evidence that M2Out still outperforms M2Dec and M1 (Supplementary Figure 9b). We finally ran a correlation between the log likelihood of the 11th trial estimated with M2Out and age, and found evidence in favor of a lack of correlation (Supplementary Figure 9c).

Supplementary Figure 9. Gender and age differences across studies. a. Distribution of women participants' age in the different subgroups. Lightest green represents the newly acquired online older female participants (N=22); darkest green the online female group presented in the main text (N=39), and intermediate shade of green, the whole sample (N=61). Error bars are the 95% confidence intervals, and thicker lines in the box plots are the median. **b.** Log-Likelihood at trial 11 estimated for the three models of interests. **c.** Correlation between age (combining the full Online female and the fMRI samples) and Log-Likelihood at trial 11 for M2Out. Statistical significance is reported in each panel for the relevant non-parametric test (independent sample t-test in a, paired t-test and b and correlation in c), and for both the frequentist and Bayesian approach. $BF > 3$ = evidence in favor of a difference, $BF < 1/3$ = evidence in favor of a lack of difference.

• Gender differences

Again, the most important analysis here, in order to justify the use of M2Out in the fMRI sample, is to show that M2Out remains the winning model, even when only considering the female participants of the Online sample. This is true both when using the complete sample that includes older female online participants (N=61) as Supplementary Figure 9b, and when using the original female sample (N=39; M2Out vs M1: Wilcoxon $W=123$, $BF_{10}=269.565$, $p=9.093 \cdot 10^{-5}$; M2Out vs M2Dec: Wilcoxon $W=65$, $BF_{10}=976.468$, $p=6.247 \cdot 10^{-7}$).

From these additional control analyses we can therefore conclude that using M2Out to estimate the learning parameters in the fMRI study is justified.

These observations have now been included in Supplementary Material §11.

• Additional note for the reviewer

These gender and age specific analyses point to a number of interesting questions that we were already investigating and will

report in a separate manuscript: does participant's gender matter, and does confederate's gender matter, do choices and model parameters change with age? Further, would it matter whether the other person's pain is visible or only symbolically communicated? Does the amount of pain that participant's perceive the other to feel matter? Does the *wf* generalize to other domains than pain? Although we have collected data to address some of these questions since submitting the manuscript, we felt that including all these data and questions would make the current manuscript too dense, and would distract the reader from our initial questions. On the other hand, making use of the newly acquired data can help reassure some of the reviewer's concerns. We therefore ran two additional correlations, including newly acquired data not included in the manuscript (from female and male participants, and female and male confederates; total $N=232$), between age and log likelihood at the 11th trial for (i) M2Out, (ii) M2Out-M2Dec and (iii) M2Out-M1. For all, there was evidence against a correlation with age: (i) Kendall's Tau= -0.014, $BF_{10}=0.09$; $p=0.762$; (ii) Kendall's Tau=0.059, $BF_{10}=0.21$; $p=0.187$; (iii) Kendall's Tau= -0.04, $BF_{10}=0.129$; $p=0.376$. Importantly, M2Out remains the winning model independently of participant's or confederate's gender and age. This further supports that the use of M2Out for the fMRI study is justified. All these aspects will be the topic of a follow up publication.

- **Power analysis on model prediction.**

The reviewer is requesting additional information about power, which is indeed an important question.

With regard to our power to determine which model best fits our data (M1 vs M2Out vs M2Dec) which is based on the 11th trial of the online experiment, the group size is 79 participants. We provide an extensive reply to this issue above, in response to reviewer 1 item 2 (R1C2). Specifically, we show that the models make sufficiently distinct predictions in this group to give us the power to differentiate them with the available data. This is also visible in the new manuscript in Figure 5f, and we therefore write in the manuscript: "Importantly, that the distribution of likelihood of the observed data given M2Out (indigo, Figure 5f) does not overlap with that given M2Dec (turquoise) or M1 (gray) shows that our approach focusing on the 11th trial in 4 blocks is sufficiently powered to adjudicate between our candidate models: a given choice pattern uniquely more likely under one model than the others."

A number of other considerations are detailed in the reply to reviewer 1 comment 2 (R1C2).

With regard to the fMRI study, with $n=25$ participants, as mentioned above, we chose to focus on specific hypothesis driven questions and methods to maximize our power. The use of neural signatures reduces the 167791 voxels or 1185 resels into a single dependent variable (signature expression) that can then be compared against zero at the group level to test whether these networks have signals that covary with prediction errors, or correlated with *wf* to explore the degree to which their recruitment is biased or not by *wf*, i.e. participants' preferences. A power analysis using G*power indicates that for the t-test in the fMRI-signature analyses we have a sensitivity at $\alpha=0.05$, $\beta=0.8$ of cohen's $d=0.51$ -> medium effect size. For the correlation test, we have a sensitivity at $\alpha=0.05$, $\beta=0.8$ of $\rho=0.45$, reflecting a moderate to large correlation. In the Participants section of the Methods we now write:

"Sample size was based on a power calculation to have 80% power to detect a medium effect size for t-tests at $\alpha=0.05$ "

In addition to these hypothesis driven analysis, we also provide some more explorative whole brain analysis that require a cluster-cutting threshold of $p=0.001$ (or $p=0.01$ for certain analyses as specified) at the voxel level, with a correction at the cluster level using random field theory (FWEc in SPM, which is larger when cluster-cutting was a $p=0.01$ than at $p=0.001$) at familywise error $\alpha=0.05$. At 0.001 for any given voxel, a traditional G*power analysis indicates a minimum effect size of $d=0.87$ to have 80% power at 0.001, and we are thus powered to detect large effect sizes in a truly activated voxel. Applying such a power-metric to fMRI mass-univariate analysis has however been criticized as it ignores the smoothness

of the data and the fact that inference is typically performed at the peak level. Applying the methods developed by Tom Nichols in the software Neuropower, as proposed in the biorXiv manuscript doi: <https://doi.org/10.1101/049429> suggests that we have a power between 64% (using FWE at the voxel level using random field correction) and 95% (using FDR correction at the voxel level) to detect the kind of effects we observe during the outcome phase of our fMRI study. Unfortunately, this software does only provide estimates for the detection of true activations at the voxel level, and does not implement cluster-level corrections, which is why we report the sensitivity using both the voxelwise corrections, although in our experience, a combination of 0.001 clustercutting threshold and 0.05 FWEc is closer to voxelwise FDR corrections in its sensitivity).

Based on these estimates, we feel that our study is appropriately powered for analysis using our signatures, and sufficiently powered to detect strong effects in our more explorative mass-univariate studies in the majority of cases (>50% of cases).

Finally, the reviewer asks whether “the computational model could reproduce behavior on the task” in our fMRI study. To illustrate the ability of our model to replicate behavior, we now include panels c and d in Supplementary Figure 7 that compares the actual and predicted choices as a function of trials number for our fMRI participants. As can be seen in Supplementary Figure 7c,d (panels also reported below for the reviewer), the models

predict the observed separation in choices between the Considerate, Ambiguous and Lucrative fMRI participants and the overall shape of the learning curve, much as it did in the Online sample.

Supplementary Figure 7.

[...] c. Observed (in green, orange and dark blue) and predicted (in gray, light and dark violet) choices over the 10 trials of the ConflictNoDropout blocks of the fMRI subjects. d Mean LOOIC for all fMRI participants

R1C6 [Imaging analysis].

One of the main imaging analyses focuses on the absence of correlation with the weighting factor and suggests that brain signals do not depend on personal preferences. However, in the fMRI study most participants were identified as considerate or ambiguous, so this is not surprising. Is there enough variance to assess this?

We thank the reviewer for raising this important point. As can be seen from Fig. 8b,d, although participants with Lucrative preference were rare, there is substantial spread in the *wf* values across

participants, due to the large numbers of Considerate and Ambiguous participants, leading to a standard deviation of wf of 0.34. Given that the correlation is not focusing on differences across the subgroups, but looking for relationships across the entire range of wf , this variability in wf provides the basis for examining whether the strength of shock representations depends on wf . In terms of sample size, a power analysis shows that with 25 participants, we would be well-powered to find a robust association ($\rho=0.5$) for our signature analysis: using the software *g*power*, with $\rho=0.5$, $p=0.05$, $n=25$, $\text{power}=0.87$. Such a power analysis also shows that any effect size $\rho>0.3$ would be detected in at least 50% of cases. In Bayesian terms, the data we observed, also provides evidence of absence, in showing that the data has $\text{BF}_{10}<1/3$, with the observed pattern of AVPS loading 3.9 times more likely under the null-hypothesis of no association with wf than under the alternative hypothesis of an association. This shows that although our data cannot exclude the presence of a weak association ($\rho<0.3$), we provide evidence against the presence of a robust association, which we would be more likely to miss than to detect. Accordingly, we specify these considerations in our fMRI results by specifying in Figure 8 legend:

“It should be noted, that with our sample size ($n=25$), we are adequately powered to detect strong associations with wf ($\rho=0.5$, $\alpha=0.05$, $\text{power}=87\%$), and therefore our result speak against such a strong association. However, our data cannot exclude the existence of weaker associations for which we were underpowered ($\rho=0.3$, $\alpha=0.05$, $\text{power}=45\%$), as 80% power for such associations would require over 60 participants.”

It should also be noted, that in our Supplementary Materials §17, our simulations assessing the ability to detect an association with wf use the 25 wf values of our fMRI participants to simulate cases in which PE_S/PE_M do, or do not, depend on wf , and are thus tailored to the particular distribution of wf we have in our sample.

Minor:

1. Terms are used inconsistently in the manuscript. “Considerate” people choose “pain reducing” option but “lucrative” people choose “lucrative” option – either match both of these terms or make them different?

We now tried to more consistently use ‘Considerate’ for both participants and choices, and ‘Lucrative’ for participants and choices

2. Lack of clarity in methods

a. Not clear if order of conflict / no conflict mixed and randomised

In the method section we now write: “The position of each symbol on the screen within a pair was randomized across trials. Symbols composing a pair were kept the same across individuals (i.e. the same two symbols formed the same pair throughout the experiment and across participants), but pairs were randomly distributed across conditions. Symbol-outcome associations were determined based on the matrix in Figure 1d,c and were kept constant across participants. ConflictDropout, ConflictNoDropout, NoConflictDropout and NoConflictNoDropout blocks were presented in a randomized order within and across participants.”

Additionally, because of the complexity of the task we revised Figure 1 (also reported below), the method section, and added an overview of experimental procedures across studies in Supplementary Figure 1, in the hope that readers would have an easier time to understand all details.

Figure 1. Learning Task. **a** Trial structure. Red outline indicates an exemplary choice for a particular trial. Pictures showing the confederate's response are still frames captured from a video used in the Online experiment in which the confederate received a painful shock (stimuli details in Supplementary Materials §1). Inter-stimulus and inter-trial intervals were adapted to the fMRI and Online situations, and are indicated separately below each relevant instance of the trial. For illustrative purposes, median reaction times (RT), and 25% and 75% quartiles, are estimated from the fMRI and Online ConflictNoDropout blocks, and the first 10T of the Online ConflictDropout blocks, as they have the same structure of ConflictNoDropout blocks. When considering all Online trials, the median is 0.91 [0.67 1.38] **b-c** Different types of block used for the Conflict (**b**; filled in rectangles) and NoConflict (**c**; empty rectangles) conditions (see also Table 1). Each individual rectangle represents a single trial within a block. The fMRI experiment only included the ConflictNoDropout blocks. The Online experiment included six different types of blocks: ConflictNoDropout, ConflictMoneyDropout, ConflictShockDropout, NoConflictNoDropout, NoConflictMoneyDropout and NoConflictShockDropout. The Dropout blocks always included 10 trials of NoDropout, in which both money and shock were presented in the outcome phase, followed either by 10 trials of MoneyDropout, in which money is removed from the outcome, or by 10 trials of ShockDropout, in which shock is removed from the outcome. An informative screen indicated participants which outcome will be removed. The Explicit report task was presented after the NoDropout blocks only in the Online experiment. The money and face on top of each block indicates which outcome was most likely for that particular pair of symbols. x4 or x6= number of block repetitions for each

experiment. See Supplementary Materials §2 for an overview of the experimental procedure across studies. **d-e** Probability table associated with each symbol of a pair for the Conflict (in **b**) and NoConflict (in **c**) condition. The most likely outcome is specified in bold font.

Supplementary Figure 1. Experimental procedures. For both the Online (a) and fMRI (b) experiments the schema details the sequence of procedures, tasks (with included conditions) and questionnaires each participant went through.

b. P3 line 87 – “Importantly, to partially de-correlate representations of shock and money, the probabilities of high and low monetary reward and pain and no-pain to others were drawn independently.” – Not clear whether it was probabilities drawn independently or outcomes that were drawn independently from the set probabilities

The latter interpretation is correct. Outcomes were drawn independently from the set probabilities: the probabilities of an outcome to occur were decided a priori to be 80 vs 20 based on relevant literature, and kept unchanged over the experiment. In every trial though, the money and shock outcomes are drawn independently. In other words, that in a given trial symbol A is associated with high shock, does not automatically mean that symbol A for that trial is also associated with high money. Over the 10 trials, one of the symbols will mostly result in high shock and high money, but this is not necessarily true at every individual trial. The matrices in Figure 1d,e illustrate for each symbol and condition, the 4 possible

combinations of outcomes. This method was chosen to help partially decorrelate the representation of shock and money.

We revised the statement to read:

“Importantly, to partially de-correlate representations of shock and money, for each individual trial, the outcomes of high and low monetary reward and painful and no-painful shock to others were drawn independently, resulting in the four possible outcome combinations shown in Figure 1d,e.”

3. Interpretation

a. Inference on p14 line 340 doesn't seem justified re. conflict vs. no conflict – no conflict just means removing one option, it doesn't change which option is best

Unlike Dropout blocks that include the removal of one outcome from the 11th trial onward, the term NoConflict refers to a condition in which both money and shock are present but without conflict: while in the Conflict condition, the high-money option also generates high-shocks to others, in the NoConflict condition, the high-money option is associated with non-painful shocks.

We hope the reorganized Figure 1 and methods help graphically clarify the structure of each block.

b. The researchers suggest that they introduce a “neurocognitive model” (p 28 line 761) but this is not clear and hasn't been discussed before

We thank the reviewer for pointing this out. We rephrased the sentence to clarify that we mean our mathematical formalization and task:

“We foresee that the mathematical formulation of learning under conflict we introduce and this task will be particularly useful to understand how neurocomputational processes may differ in atypical populations in the spirit of computational psychiatry (Montague et al., 2012), particularly in antisocial populations”

4. Clarity / consistency throughout the manuscript:

a. NoConflict vs. No-Conflict

We checked the manuscript to remove inconsistencies.

b. P56 line 1482 & fig 7b legend on plot the middle group is called “neutral”, but figure 4c legend on plot “indifferent”, other places “ambiguous”

We now corrected the figures to be more consistent with our nomenclature.

c. Consider using different symbols for kendalls tau and tau softmax for clarity p 63, line 1594.

We now use Tau for the Kandell's Tau, and τ for the softmax.

d. Supplementary table 3 - rename wf_10T_M2Dec to give a more descriptive name.

We now renamed it and Supplementary table 3 became Supplementary Table 6.

e. Was eye gaze data collected through gorilla? prolific is mentioned p 84 line 1868, as prolific does not contain a way to collect eye gaze data as far as I am aware?. Fig S7 mentioned but it is Fig S8

We apologize for the mistake. Eye gaze data were indeed collected through Gorilla, and not prolific. We now corrected the text. We also corrected the reference to the correct figure.

Reviewer #2 (Remarks to the Author):

The current study examined how people learn associations between actions and their moral and financial consequences, as implemented by means of personal financial gains and pain delivered to other people. The study examines such learning for new associations under conflict and no conflict between these two outcome features, as well as after a change in such learned contingencies. The authors conducted two experiments, one on-line behavioral experiment and one fMRI experiment. In behavioral data analyses, the authors categorized subjects into three groups (considerate vs. ambiguous vs. lucrative), based on their proportion of pain-reducing choices, and showed that these different groups exhibit different dynamic changes in choice patterns over time (learning curves). Moreover, after contingency changes (when either self-gain or other-pain was removed), subjects changed their choice tendency more strongly when the feature was removed that they weighted more strongly before. These results suggest that people are able to learn the values of each feature, not just a compound value integrating both features. This was also confirmed by computational modelling analyses suggesting that the data are best fit by models that incorporate separate terms for the values of the two features. The weighting parameter estimated from the winning model was related to not only pain-reducing choices but also altruistic donation behavior in an independent task. Finally, fMRI results suggested that the prediction error of other-pain intensity was represented in the pain-observation network, and that activity of the ventromedial prefrontal cortex was associated with individuals' bias on a specific feature.

In general, I enjoyed reading this paper. The questions are quite detailed but interesting, the experiment is generally well designed, and the analyses are thorough. However, I still have several concerns regarding the data analyses and interpretation that the authors should address to render their study conclusive.

Major issues:

R2C1

The authors claim that their design and analyses allow them to separate “model-free” from “model-based” learning, in close similarity to similar approaches in other studies in animals and humans. I am not sure this statement is in line with how most people in the field define and study model-based learning. First, the learning task in the present study is fundamentally different from the variants of 2-stage learning tasks commonly used to separate model-based and model-free learning, which actually make opposing choice predictions for critical trials. I do not think model-based learning is unambiguously separated from model-free learning by separate versus conjoint feature valuation, as the authors imply. Where does this definition come from? Second, the winning (M2Out) and alternative models (M1 and M2Dec) used here are all variants of feature-based reinforcement learning models, and thus do not seem to computationally differentiate model-based learning from model-free learning. Which feature of the computational models allow inferences on the use of a cognitive model of reward, pain, and feature contingencies? If the authors really want to separate “model-based” from “model-free” learning with their task and computational models, they should clearly define the rationale underlying this separation and clarify the similarity and differences between this rationale and that commonly used in the literature. However, the authors could also focus solely on the question whether financial versus moral features are learned separately and how they are integrated; this is an interesting issue that does not necessarily need to be related to the debate about model-based vs. model-free learning.

We thank the reviewer for this comment and for encouraging us to focus the introduction solely on the question whether financial versus moral features are learned separately and how they are integrated. Accordingly, we removed the use of model-based and model-free entirely from the introduction and the results, and instead reworded our framing in terms of separability and bias of the learned symbol-money and symbol-shock representations. We only link to the distinction model-based and model-free in one section of the discussion that reads: “Some have found that learners that respond flexibly to devaluation, as most of ours did, which indicates goal-directed behavior instead of habit, are more likely to have used a model-based form of learning (Dolan and Dayan, 2013; Gillan et al., 2015), yet others found participants’ choices to be dominated by model-free learning when preventing shocks to others in two-choice Markov decision tasks (Lockwood et al., 2020). It may thus be that participants represent the nature of outcomes (as revealed by our devaluation study) while avoiding the cognitively expensive decision-tree necessary to ascribe rewards following a rare transition in a two-choice Markov decision task to an alternative (nonselected) action”.

We agree with the reviewer that this revised framing cuts more effectively on the interesting question at the actual core of our paper that had remained unexplored in the field: how participants deal with the duality of outcomes in conflictual moral situations (see also answer to R1C1).

R2C2 [ambiguous participants]

Although the authors showed that people in the ambiguous group do not have clear preferences for self-gain or other-pain, it is still unclear how and why these individuals differed from those with clear preferences. The authors claim that “the ambiguous preference group appears to contain a mix of individuals that do not show significant learning in our task even without conflict, and individuals that can learn without conflict but fail to display clear preferences in the presence of conflict”. However, the data do not clearly support this conclusion. For example, Figure 4A-D shows no clear group difference in perceived association between ambiguous and considerate or lucrative people. Thus, ambiguous people seem to learn the symbol-outcome associations as well as the others, even in conflict conditions. Relatedly, while Figure 5A and supplementary Figure 5 show no significant changes from the 10th to the 11th trial in ambiguous people, these choices were still significantly different between different contingency contexts (i.e., shock removed vs. money removed) and showed adaptations from the 11th to the 20th trial. Thus, ambiguous people can in fact learn the symbol-outcome associations, implying that the flat “learning” curves may reflect a neutral preference for self-gain vs other-pain rather than deficient association learning.

If the authors want to maintain that the learning process differs between the groups, they should really show statistical differences in estimated learning-rate parameters across groups. In addition, since the learning curves are averaged across blocks, it is hard to tell if the flat curves are driven by neutral preferences across all the trials or by different strategies across different blocks (e.g., greater weight on other-pain in half blocks and greater weight on self-gain in the half blocks). This should be tested for as well.

We agree that the ambiguous group merits further characterization and we therefore ran some explorative analyses to gain deeper insights into the behavior of the ambiguous participants, and report these analyses in Supplementary Materials §12, which read as follows:

“To gain deeper insights into the behavior of the ambiguous participants, we ran some explorative additional analyses to address the following questions.

1. Do Ambiguous individuals alternate their preference across blocks?

To understand whether individuals of the Ambiguous group choose to alternate their preference between blocks, we first plotted the average of the proportion of pain-reducing choices per block for each participant and all the blocks of the Ambiguous group. Only the first 10T are included, and we combined both the Dropout and NoDropout blocks (total number of blocks = 12: 4 ConflictNoDropout, 4 ConflictShockDropout, 4 ConflictMoneyDropout). From this graph it is hard to see whether participants choose totally randomly (therefore staying around 0.5 for each block), or whether they use a strategy in which within a block they have a preference, but voluntarily switch preference from block to block (going from a clearly considerate choice for one block to a clearly lucrative choice in another block, i.e. being above chance considerate, or lucrative, but only within blocks). The graph includes some participants that have relatively extreme scores in certain blocks, suggesting that some participants may have had per-block-preferences. The problem is that doing a binomial per block with only 10 trials to see if there was a significant preference per block lacks sensitivity - one needs 9/10 choices in one direction to have a significant binomial, and that can hardly be expected given that participants need to learn within a block.

To investigate whether ambiguous participants display significant preference within certain blocks, but may switch preference between blocks, we therefore reasoned as follows. If participants have a preference in a block, the probability to choose the considerate option should deviate from 0.5. If they switch their preference between blocks, sometimes the deviation might be upwards (when expressing a considerate preference) and sometimes downwards (when expressing a lucrative preference). Hence, evidence of - albeit alternating - preference would be captured by the sum of the squared deviations from 0.5 (SSD) across the 12 blocks. To assess how surprising a given summed square deviation would be, we can compare it against simulated choices of a random chooser (possibly as a consequence of not having learned the task). We therefore estimated this SSD across blocks for each participant, and ran a simulation to estimate how a distribution of random choices would look like. We then estimated the critical value (i.e. 95th or 99th percentile of the null distribution SSD) and compared our subjects to it. Three (blue lines in Supplementary Figure 10) out of 26 participants show SSD values in excess of the 99th percentile of the null distribution (Supplementary Figure 11). This suggests that there is evidence that in the Ambiguous group at least 3 out of 26 ambiguous participants do not choose randomly, but do show a preference in some of the blocks, albeit with different preferences across different blocks. Note that 3/26 at $\alpha=0.01$ is larger than what we would expect based on a 1% false positive rate alone (binomial, $p=8.8E-5$).

However, for the vast majority of the ambiguous group (23/26=88%) we have no evidence that their choices are non-random at $\alpha=0.01$, and for 20/26 (77%), that they are non-random at $\alpha=0.05$.

Supplementary Figure 10. Choices over blocks. Gray and blue lines (one for each participant of the Online dataset) indicate the average choice over the 10 trials of ConflictNoDropout and ConflictDropout for each of the 12 blocks. Blue highlighted participants are those that show evidence of having significant, albeit changing, preference over blocks, as evidenced by an SSD above the 99th percentile of the null distribution SSD as shown in Supplementary Figure 11. Red represents the overall average across participants.

2. Do Ambiguous individuals lack a clear preference (i.e. choose more randomly)?

We also examined whether the group of Ambiguous participants overall shows higher SSD than would be expected by chance, by summing the SSD of the 26 participants, and comparing it against a null distribution obtained via 10000 simulations of a group of 26 random choosers, and found the real group SSD to be in the 98.1th percentile of the null distribution of group SSD, showing that as a group, the ambiguous participants also showed more extreme choices than expected if they were not to have any preference at all.

Supplementary Figure 11. Real vs. Simulated Sum of Squared Deviation from 0.5. We simulated 100000 random choosers performing 10 trials per block for 12 blocks. For each block and simulated participant, we calculated the proportion of considerate choices, and then, for each participant a sum of the squared deviations (SSD) by summing the squared deviation from 0.5 across the 12 blocks. The blue histogram shows the distribution of the SSD across these 100000 simulated random choosers. The SSD of the 26 ambiguous participants is shown as open orange circles above the histogram. The solid and dotted vertical lines represent the 99th and 95th percentile of the simulated random choosers. Real participants with values above those lines thus are unlikely to have chosen entirely randomly, as their preference per block fluctuated more than random choices would predict.

3. Do Ambiguous individuals have poorer implicit learning?

During the Conflict condition, because of the presence of a conflictual decision, participants' flatter learning curves could be explained by a poorer learning, but also by other factors, such indecision or different strategies in performing the task. Looking at the NoConflict condition is therefore the most appropriate test to investigate whether the Ambiguous group learns similarly to the other groups, as it represents the win-win situation in which participants can learn to increase their chances to add a financial bonus without causing any pain to the confederate. Importantly, in the NoConflict condition, whether participants value reducing shocks to others and/or increasing their own financial gains, one option is clearly better than the other, and the percentage favorable choices can be interpreted as a measure of performance. Along that reasoning, when looking at Ambiguous participants' choices, we can clearly see that on average their choices remain worse than the other groups also in the NoConflict conditions (Supplementary Figure 5a,b and c). Even at the 10th trial - i.e. at the moment in which participants should have learned the symbol-outcome association for each block - the Ambiguous group remains the lowest in % favorable choices (main effect of group Kruskal-Wallis ANOVA=10.924, $p=0.004$; Mann-Whitney U independent sample t-tests: $W_{\text{Ambiguous vs Lucrative}}=197$, $p=0.023$, $BF_{10}=1.516$; $W_{\text{Ambiguous vs Considerate}}=192$, $p=0.001$, $BF_{10}=8.235$) confirming that overall the Ambiguous group learns more slowly than the other groups. Interestingly, this 10th trial is also the one after which participant's explicit reports are collected in the NoDropout blocks.

We then, as suggested by the reviewer, looked at individual parameters derived from our learning model, LR and Tau in particular. We ran two one-way non-parametric ANOVAs, one on the LR for money and one on the LR for shock, with group (Ambiguous, Considerate and Lucrative) as between-subjects factor. The Kruskal-Wallis test indicates a significant effect of group for both the LR for money ($H=18.842$, $p=8.099 \cdot 10^{-5}$) and shock ($H=6.049$, $p=0.049$) in the NoConflict condition. Post-hoc non-parametric frequentists and Bayesian t-tests show that the LR for shock differs from that of the Considerate group, and the LR for money differs from that of the Lucrative group. The figure below illustrates the result and indicates the statistical values of this analysis (Supplementary Figure 12a,b) and the same analysis performed on the Conflict condition (Supplementary Figure 12d,e). Overall, these results suggest that despite the LR of the Ambiguous group remaining within the range of the other two groups - therefore supporting that some learning occurs - their average is lower than that of the groups that values a particular outcome (as revealed by their choices under Conflict). That is to say, for Money, the Lucrative participants are known to value maximizing Money, and their LR for Money is significantly higher than that of the Ambiguous participants. For Shocks, a similar trend occurs: the considerate participants value avoiding shocks, and their LR for Shocks is higher than that of the Ambiguous participants. On the other hand, the LR of the Ambiguous group resembles that of the groups that seem to value that outcome little, and hence

the Ambiguous LR for shocks resembles that of the Lucrative participants that do not particularly value shocks, and their LR for money resembles that of the Considerate participants that do not particularly value money. A limitation of this analysis is the fact that LR is less accurately estimated for the non-preferred outcome. As Ambiguous participants show a less clear preference, their LR may also be less accurately estimated.

The non parametric ANOVA on the parameter Tau, also indicates a significant main effect of group (Kruskal-Wallis main effect of group, $H=17.105$, $p=1.93 \times 10^{-4}$). Non-parametric post-hoc t-tests indicate the parameter Tau is significantly the lowest for the Ambiguous group compared to the others (Supplementary Figure 12c), suggesting the Ambiguous group makes more random decisions than the other groups even when the model suggests that they have learned a similar difference in expected values across the alternative options. The exact same patterns of results was observed for the Conflict condition (Supplementary Figure 12f).

Supplementary Figure 12. Learning parameters across groups. a-c LR_M , LR_S and Tau values for the Ambiguous (orange), Considerate (green) and Lucrative (dark blue) online participants, estimated by M2Out for the first 10 trials of the NoConflict condition. Barplots indicate the median and 95% confidence interval. Note that these NoConflict trials were not included in the model fits presented elsewhere, which only include the Conflict trials. d-f same as in a-c but estimated for the Conflict condition. Statistical significance is reported in each panel for the non-parametric independent samples planned post-hoc testing, both for the frequentist and Bayesian approach. $BF > 3$ = evidence in favor of a difference, $BF < 1/3$ = evidence in favor of a lack of difference.

Finally, we looked at whether Ambiguous participants were overall slower in making their choices in terms of reaction times (Supplementary Figure 13). The Kruskal-Wallis ANOVA shows a main effect of group (8.166, $p=0.017$). Non-parametric independent samples t-tests on reaction time (RT) data from the NoConflict condition, indicate a trend for the Ambiguous participants to be slower than the Considerate group, but there is evidence for a lack of a difference in RT with the Lucrative group. The same ANOVA on the RT during the Conflict condition does not reveal a main effect of group (4.657, $p=0.097$). Overall, the evidence suggesting the Ambiguous group takes longer to choose is very small, and the RT analysis supports more the idea that the Ambiguous responds with a comparable RT than the other groups.

Supplementary Figure 13. Reaction time across groups. RT values for the Ambiguous (orange), Considerate (green) and Lucrative (dark blue) online participants, for the first 10 trials of the NoConflict condition. Barplots indicate the median and 95% confidence interval. Statistical significance is reported in each panel for the non-parametric independent samples planned post-hoc testing, both for the frequentist and Bayesian approach. $BF > 3$ = evidence in favor of a difference, $BF < 1/3$ = evidence in favor of a lack of difference.

4. Do Ambiguous individuals have poorer explicit learning?

When looking at the graphs of Figure 3c,d, even Ambiguous participants report the association probabilities with above-chance accuracy. However, it is also noticeable that the Ambiguous group's accuracy in recalling the difference between the two symbols for a particular outcome seems closer to the accuracy of the group that values this particular outcome less: they report this difference for money less accurately than the Lucrative group and only as well as the considerate group, and they report this difference for shocks less accurately than the Considerate participants and only as well as the Lucrative participants. In other words, while the Considerate groups tends to recall more accurately the difference between the two symbols when reporting the probabilities of shock, and the Lucrative recall better the difference when reporting the probabilities of money, the Ambiguous group accuracy is comparable to the probabilities the Considerate group reports for the money and the Lucrative for the shock.

Supplementary Figure 14 illustrates this effect more clearly and reports the statistical values of the non-parametric independent t-tests we ran. While in the NoConflict condition the accuracy in recalling the probabilities of the two symbols associated with money seems not to clearly differ across group (Kruskal-Wallis ANOVA main effect of group=1.146, $p=0.284$), a difference in accuracy becomes visible when recalling the probabilities associated with shock (Kruskal-Wallis ANOVA main effect of group=10.371, $p=0.001$). In particular, the Ambiguous group recalls the probabilities of the symbol-shock association worse than the Considerate group. When repeating the same analyses on the Conflict condition, this effect accentuates and the accuracy of the Ambiguous group worsens also for the symbol-money associations, for which the Ambiguous becomes worse than the Lucrative group.

These analyses, again support the idea that although the Ambiguous group explicitly learns *something* about the difference between the two symbols, their accuracy tends to be lower than the accuracy of the Considerate and Lucrative groups for their outcome of value. If the choices of Ambiguous participants were to have been the result of excellent learning combined with a strategy to balance the two outcomes, one may have expected them to recall both outcomes as well as the group prioritizing that outcome, rather than as poorly as the group not prioritizing that outcome in their decisions.

Supplementary Figure 14. Explicit recall comparison across groups. **a-b** Difference between Money a and Shock b reported probabilities for the low-shock minus high-shock symbol (Symbol1-2), separately for the Considerate (green), Lucrative (blue) and Ambiguous (orange) preference group, for the NoConflict condition (same as in Figure 3d). **c-d** same as in a-b (and Figure 3c) but for the Conflict Condition. Statistical significance is reported in each panel for the non-parametric independent samples planned post-hoc testing, both for the frequentist and Bayesian approach. $BF > 3$ = evidence in favor of a difference, $BF < \frac{1}{3}$ = evidence in favor of a lack of difference.

To summarize, evidence that the ambiguous group does learn comes from:

- The fact that in the explicit report the ambiguous group reports differences between the probabilities associated with the two symbols that are significantly different from zero, both for the Conflict and NoConflict condition and for both the money and shock probabilities (Figure 3a,c)
- The fact that choices in the NoConflict conditions are above chance level, and in the same directions of the other groups
- The fact that at the 11th trial of the Dropout blocks participants significantly discriminate between MoneyDropout and ShockDropout blocks (i.e. Figure 4a, 11th trial)
- The fact that on average their choices are more extreme than expected by chance (Supplementary Figure 11)
- Reaction time mostly remains within the range of the other two groups (Supplementary Materials §13).

Evidence that the ambiguous learn more slowly than the other groups comes from:

- The proportion of favorable choices is on average lower than in the other groups in the NoConflict condition (Supplementary Figure 6b). Even at the 10th trial - i.e. at the moment in which participants should have learned the symbol-outcome association for each block - the Ambiguous group remains the lowest in % favorable choices, suggesting that overall the Ambiguous group learns more slowly than the other groups.

- For ambiguous participants the LR for money is lower than the Lucrative group, and the LR for shock lower than the Considerate group (Supplementary Figure 12a,b).
- Tau indicates that the Ambiguous group performs more random decisions compared to the other groups (Supplementary Figure 12c).
- Accuracy in explicit recall remains lower for the Ambiguous group compared to the preferred outcomes of the other groups (Supplementary Figure 14a,b).

We refer to these analyses in different parts of the Result section of the main manuscript:

“The curve of the group with Ambiguous preference, consistently remains around 50% showing no clear learning curve (Supplementary Materials §12).”

“the lack of clear preferences amongst participants with Ambiguous preferences was not due to an utter lack of explicit symbol-outcome learning (but see Supplementary Materials §12)”

“Although ambiguous participants explicitly learn the symbol-outcome associations, their accuracy tends to be lower than the accuracy of the Considerate and Lucrative groups for their outcome of value (Supplementary Materials §12).”

“In summary, choices reveal substantial individual variability in preference, but most participants show evidence of some form of learning (See Supplementary Materials §12 for learning in participants with Ambiguous preference).”

R2C3

Related to the last point, I wonder to what extent the current design and modelling approach can differentiate the learning rate and the weighting factor. Are these two parameters correlated?

As seen in Supplementary Figure 16 reported below, the two learning rates are not significantly correlated with wf across our 79 participants from the online study. The top row shows the relationship across the actual values, the bottom row across the ranks of the values to more directly reflect the fact that Kendall's Tau is calculated on ranks. Overall, we can therefore conclude that there is sufficient unique variance for the model to sufficiently differentiate between LR and wf .

What we do observe in our data and in simulations, is that, somewhat unsurprisingly, the LR parameter estimates in our Bayesian framework are more influenced by the data for the quantity that is more valued: participants with very low wf , that mainly chooses to reduce shocks, have estimates for LR for shock that are more constrained by the data while participants with very high wf , that mainly choose to maximize gains, have estimates for LR for money that are more constrained by the data.

We now included these observations in Supplementary Material §13 and Supplementary Figure 16.

“To investigate whether the current design and modeling approach can differentiate the learning rate and the weighting factor we run non-parametric correlations between the two parameters, separately for money and shock. Overall, results indicate a lack of significant correlations between LR and wf . BF additionally indicates clear evidence of a lack of correlation between LR_S and wf , and a similar trend for LR_M and wf (Supplementary Figure xx). We can therefore conclude that there is sufficient unique variance for the model to differentiate between LR and wf .

What we do observe in our data and in simulations, is that, somewhat unsurprisingly, the LR parameter estimates in our Bayesian framework are more influenced by the data for the quantity that is more valued: participants with very low wf , that mainly chooses to reduce shocks, have estimates for LR for shock that are more constrained by the data while participants with very high wf , that mainly choose to maximize gains, have estimates for LR for money that are more constrained by the data.”

Supplementary Figure 16: Correlations between LR and wf . **a.** Online participants' ($N=79$) learning rate values for money (LR money) plotted against participant's wf . **b.** Same as in (a) but for the learning rate for shock (LR shock). **c.** Correlation between learning rate for money (LR money) and wf . **c.** As LR and wf were not normally distributed (Saphiro-Wilk $p<0.01$), the graph plots the rank LR money against wf to directly reflect the fact that Kendall's Tau is calculated on ranks, rather than actual values as shown in (a). **d.** Same as in (c) but for the LR shock against wf .

R2C4

In the imaging analyses, the authors selectively focused on how PES covaried with the AVPS. Why? One could equally have hypothesized such a relationship for the neural prediction of other-pain and the AVPS; or for the prediction or prediction errors for self-gain and reward networks. Moreover, the authors directly relate the outcome of a multivariate analyses with one selectively chosen results of a univariate analysis. This does not seem to be clearly motivated by a priori hypotheses. The authors should be more unbiased in their report of the neuroimaging results. In particular, they should provide a clear rationale for which neural hypotheses they want (or don't want to) to examine, and they should comprehensively report the corresponding analyses even if they don't yield significant results.

We thank the reviewers for this insightful suggestion and we regret not having pre-registered the study. As also mentioned above in response to a previous comment, as a natural follow up of our past work on shared circuits for actions, sensations and emotions, one of the questions we had at the time of writing the grant that allowed for this work, was to explore whether sharing and simulating the pain of others helps to attribute a value to the pain of others.

Extract from the public abstract of the ERC-StG grant HelpUS to VG, submitted in 2017 (https://erc.europa.eu/projects-figures/erc-funded-projects/results?search_api_views_fulltext=&f%5B0%5D=funding_scheme%3AStarting%20Grant%20%28StG%29&page=204): *"The success of humans depends on their ability to cooperate. Cooperation requires learning to avoid actions that harm others and select those that balance benefits for self and other. Reinforcement learning captures how individuals learn to optimize benefits for themselves, by associating actions and outcomes for the self. The social context requires to incorporate outcomes for others into that equation by transforming them into the currency used to value our own outcomes. Research on empathy, by suggesting that we transform the emotions of others*

into neural representation of how we would feel in their stead, provides testable mechanistic hypotheses of how we do that. The painful facial expression of our friend after we kick him would be transformed into the pain we would feel when kicked, associating kicking with negative value, thereby motivating us to stop kicking.”

Over the years the difficulty of testing this question without excessive reverse inference became increasingly clear. The development of multivariate signatures, although still imperfect, can constrain this inference. Although the reviewer is right, in that many hypotheses *could* have been made, we decided it was most appropriate to concentrate on our core questions, of how empathy related brain regions process the pain of others during learning, at first, and in the main result section. This decision was made in particular because the fMRI dataset we present here was limited in sample size by COVID. In such smaller sample sizes, we are keen to focus on our hypothesis driven analysis to avoid the curse of trying too many analyses. We therefore now tried to formulate more clearly our original hypothesis in the introduction, and focus on the most direct tests of these hypotheses in the main text.

However, it is only natural that readers may have questions that differ from our own, and may wish to see results for other analyses they may have liked to perform. In particular, many may be interested in PE_M , and reward-processing in our task as a direct pendant to the shock-processing. As mentioned in response to one of reviewer’s #1 comments (RIC3 in particular), we therefore now complement our inferences and analyses on PES and the pain-empathy network (as captured by the AVPS), with inferences and analyses on PEM, and the reward circuitry. In particular, we took advantage of datasets previously collected by the author S. Speer and additional open access datasets on economic paradigms meant at identifying the circuit of economical reward and losses, to build a reward signature for reward. This work was inspired by the vicarious pain signatures we use to test the involvement of regions processing pain. To avoid overloading the current manuscript with additional datasets, we decided to prepare a separate publication on the reward signature. A detailed description of our work leading to the reward signature can therefore be found on bioRxiv at the following link: <https://doi.org/10.1101/2022.06.16.496388>. The new analyses on reward are now presented in the manuscript side by side with the analysis on shocks in Figure 7 (for whole brain analyses) and 8 (for the signatures) and in the result text.

Figure 7. Key fMRI results. **a** Voxels where BOLD signals are increased during the outcome phase, independently of prediction errors, ($Outcomes > 0$), $p_{unc} < 0.001$, $k = FWEc = 903$ voxels (Supplementary Table 8). **b** Red: regression between PE_S and $1-wf$ ($p_{unc} < 0.001$, $k = FWEc = 181$ voxels, Supplementary Table 10). This identifies voxels with signals that increase more for shock outcomes that are less intense than expected (i.e. positive PE_S) in participants that place more weights on shocks (high $1-wf$ values). Yellow: results of the contrast constant > 0 in the same linear regression with $(1-wf)$ ($p_{unc} < 0.001$, $k = FWEc = 167$ voxels). This identifies voxels with signals that increase with increasing PE_S after the variance explained by $(1-wf)$ is removed (Supplementary Table 11). **c** Cyan: $PE_M > 0$ (after removing variance explained by wf ; Supplementary Table 12) at $p_{unc} < 0.01$, $k = FWEc = 542$ voxels. Purple: $PE_S > 0$ presented the same cluster-cutting threshold of $p_{unc} < 0.01$, $k = FWEc = 950$ voxels (Supplementary Table 11) in order to visualize that the two networks are largely distinct (but see Supplementary Figure 17 for overlap). **d** Correlation between PE_M and wf , $p_{unc} < 0.01$, $k = FWEc = 1642$ voxels (Supplementary Table 13). **e** Results of the contrast $PE_S * LR_s > 0$, $p_{unc} < 0.001$, $k = FWEc = 121$ voxels (Supplementary Table 14), which represent the BOLD signal associated with shock value updating. Note that across all panels, results are FWE cluster corrected at $\alpha < 0.05$ using (i) cluster cutting at $p_{unc} < 0.001$ (panels a, b, e) or (ii) $p_{unc} < 0.01$ (panel c, d) then excluding clusters with an extent below the critical FWE cluster size FWEc. Cluster-extent threshold size always indicated in figure panels. Renders are created based on the cortex_20484 surface from spm12, slices are taken from the average T1 anatomical scan from our participants.

Figure 8. Loading on AVPS and RS. **a** Loading of the normalized parameter estimate images (β_{PE}) for PE_S and PE_M on the AVPS²². Each dot is one participant, the horizontal line is the mean. p and BF values reflect a one-sample parametric (because normality was not violated) t -test against zero in JASP. **b** Loading of the parameter estimate for PE_S (β_{PE_S}) and PE_M (β_{PE_M}) onto the AVPS as a function of wf . p and BF values represent the outcomes of a parametric correlation analysis using Pearson's r . **c-d** Same as in a-b, but for the RS. It should be noted, that with our sample size ($n=25$), we are adequately powered to detect strong associations with wf ($\rho=0.5$, $\alpha=0.05$, power=87%) but cannot exclude weaker associations for which we were underpowered ($\rho=0.3$, $\alpha=0.05$, power=45%), as 80% power for such associations would require over 60 participants. **e** Voxels contributing substantially to the loading of β_{PE_S} (top) and β_{PE_M} (bottom) onto AVPS. Because the overall loading is calculated by first multiplying each voxel's β_{PE_S} value with the value in this voxel of AVPS, and then summing these multiplication over all voxels, here we simply calculated the first step (the voxelwise multiplication of β_{PE_S} and AVPS, and β_{PE_M} and AVPS), and averaged the result across our participants. To not overcrowd the image, we only show voxels with values above 50 or below -50 after multiplying the images by 10^6 . **f** Same as in e, but for the RS.

In Supplementary Figure 18 (also reported below) and Supplementary Tables 15-16 we present a number of additional analyses regarding the decision phase.

Supplementary Figure 18. Decision phase and EV. **a** Results of the second level t-test $\text{Decision} > 0$, indicating voxels where BOLD signals during the decision phase (independently of prediction errors) are increased, $p_{\text{unc}} < 0.001$, $k = \text{FWEc} = 235$ voxels. **b** BOLD signal negatively correlating with the expected value for shock, $p_{\text{unc}} < 0.01$, $k = \text{FWEc} = 420$ voxels (Supplementary Table 16). The white circle indicates the cluster also surviving at $p_{\text{unc}} < 0.001$, $t = 3.48$, $k = \text{FWEc} = 137$ voxels, with the peak at the indicated coordinates. **c** The renders and slices visually compare activity correlating with $-EV_S$ (purple) and PE_S (yellow), both shown at $p_{\text{unc}} < 0.01$ with each relative FWEc. The images clearly show that $-EV_S$ and PE_S can be dissociated, with several clusters independently correlating with $-EV$ and PE_S , and only four clusters (not surviving an FWEc correction at $p_{\text{unc}} < 0.0$) overlapping between PE_S and $-EV_S$. Note that $-EV_S$ is modeled as a parametric modulator of the decision phase, while PE_S as a modulator of the outcome phase. Expected values computed during PE_S cannot be isolated. **d** Overlapping clusters between $-EV_S$ and PE_S at $p_{\text{unc}} < 0.01$ without any cluster correction. Description based on the Anatomy toolbox for SPM, and the statistical whole brain tables from SPM12. As in Figure 7, all results are FWE cluster corrected at $p_{\text{unc}} < 0.05$, following cluster cutting at $p_{\text{unc}} < 0.001$ or $p_{\text{unc}} < 0.01$, specified using the critical FWE cluster size FWEc as indicated in figure panels. Renders are created based on the cortex_20484 surface from spm12, slices are taken from the average T1 anatomical scan from our participants.

Finally, using the procedure proposed by Zhang and colleagues (2020), we also explore associations with Out_S , Out_M and EV_S and EV_M in Supplementary Materials §18 and refer to them in the text “Using the method developed by Zhang and colleagues (2020) suggests that the AVPS, the RS and the ventral prefrontal network correlating with PE_S during the outcome phase do indeed represent PE_S rather than only encoding the raw witnessed outcomes (Supplementary Material §18).”, and Supplementary Material §18 reads as follows:

“18. Two-steps procedure

In learning paradigms like the one we use, that $PE = \text{Out} - EV$ causes outcomes (Out) and predictor errors (PE) to be highly correlated, as a positive Out trial (high-money or low-shock) will always be associated with positive PE, and negative Out trial with negative PE (see the formulae connecting Out and PE in Figure 5a). In our fMRI study, PE_S and Out_S had an average correlation of 0.741 (ranging 0.750 to 0.866 across our 27 participants), and the PE_M and Out_M had an average correlation of 0.749 (ranging 0.781 to 0.862 across our 27 participants). To disentangle whether a region or network with signals correlating with PE really encodes PE or simply encodes Out, Zhang and colleagues (2020) proposed a simple analysis: if the signal correlates with Out but not EV, they propose to consider it to encode Out not PE; if the signal correlates with Out and, in the opposite direction, with EV, they propose to consider it to encode PE^{51} . For the signals in our two signatures and in the frontal clusters associated with PE_S we applied this logic. We extracted raw blood-oxygen-level dependent (BOLD) time series from the two signatures (i.e. by multiplying each volume with the signatures to create a scalar time-series for each signature) and the average signal from the two prefrontal ROIs that had signals correlating with PE_S (yellow in Figure 7b). These time series of each participant were then time-locked to 2s before onset of the video clip to 12s after the end of the video. Time series were up-sampled to a resolution of 200 ms using 2D cubic-spline interpolation, resulting in a data matrix of size m-by-n, where m is the number of trials, and n is the number of the up-sampled time points (i.e., $14s/200ms = 70$ time points). A multiple regression model containing Out_S , Out_M , EV_S and EV_M was then estimated at each time point (across trials) for each participant. It should be noted that, although the linear regression here took a similar formulation as the first-level general linear model (GLM), it did not model any specific onset; instead, this regression was fitted at each time point in the entire trial across all the trials.

To test group-level significance of the above time series analysis, we employed a permutation procedure. We defined a time window of 4–8 s after the corresponding event onset, during which the BOLD response was expected to peak. In this time window, we randomly flipped the signs of the time courses of effect sizes for 5,000 repetitions to generate a null distribution, and tested whether the mean of the generated data from the permutation procedure was smaller or larger than 97.5% of the mean of the empirical data.

For the AVPS, we found an expected negative correlation with Out_S (i.e. higher AVPS signals for high-shock than low-shock trials, which is what the signature was designed to do²⁴) and a positive correlation with EV_S . To our knowledge, this may be the first evidence that the facial pain-witnessing network that had only been developed to discriminate the sight of high- vs low-pain facial expressions actually encodes predictor errors in a learning context - at least in our paradigm²⁴.

For the RS, we found the expected positive correlation with positive outcomes (i.e. with Out_S and Out_M) and a negative correlation with EV_S and EV_M . In agreement with a substantial literature associating this striato-prefrontal network with predictor error coding^{11,25}, we therefore may conclude that it encodes PE in our paradigm, both for Shocks and Money.

For the two prefrontal regions emerging from our PE_S analysis after removing variance explained by wf , we find both to show the pattern that would be expected if they encoded PE_S : correlating positively with Out_S and negatively with EV_S .

Somewhat less surprisingly, both also seem to show some positive correlation with EV_M , in line with the notion that the vmPFC could generate a common currency combining the values in terms of Shocks and Money to later enable a decision that combines these two measures^{25,26}.

Supplementary Figure 19: Dissociating Outcome vs Prediction Error coding. **a** Results of a multiple regression analysis in the two ventral prefrontal cortex clusters (left and right vPFC) that showed significant PE_S correlation after removing variance explained by wf in Figure 7b. Beta-weights that are significantly different from zero based on permutation statistics in the time interval indicated by the gray box are marked by stars (**: $p < 0.01$, ***: $p < 0.001$). Shading indicates the s.e.m. across participants. Time=0 indicates the onset of the movie showing the outcome of a trial. The location of the clusters is shown in yellow in the renders. **b** Same for the AVPS and RS neural signatures.”

R2C5

In the neuroimaging analyses, the authors used the wf -normalized PE to avoid “that wf influences results twice”. They “divided PES with $(1-wf)$ and PEM with wf ”, so that the wf -normalized PES equals $(OutS \cdot (1-wf) - EVS) / (1-wf)$. I don’t fully understand this rationale. First, wf is a constant for each subject, so the wf -normalized PE should be highly correlated with PE (and may even lead to identical neuroimaging results if this scaling factor is removed when the corresponding regressor is standardized during GLM construction). Second, since the wf is embedded in the first-level PE analysis, it is unclear how this transformation can avoid “that wf influences results twice”, given that now wf is in fact included in both the first-level and second-level analyses. Please provide a clearer rationale and possibly simulation to illustrate the usefulness of (and necessity for) this transformation.

We thank the reviewer for flagging that we insufficiently explained the utility of this transformation. We now added an entire section in the supplementary materials with example calculations to explain this procedure (please see Supplementary Materials §15 for details). In brief: the reviewer is right, that wf is a

scalar value, and using it to divide PE and EV values within a first level model does not alter the significance of location of regions associated with PE or EV. However, because we deliberately do not standardize PES, PEM, EVS and EVM predictors before entering them into the GLM, but only mean subtract them, division by wf or $1-wf$ does directly influence the magnitude of the parameter estimate maps that are generated at the first level. In the supplementary materials we go through why doing so is desirable in some detail, but what is important, is that two people with different wf values, witnessing the same first outcome of a block (e.g. a high-shock, high-money outcome) and having the same magnitude of the BOLD signal, have the same parameter estimates for the PE parametric modulator after we divided the PE values, but would have different parameter estimate values if we hadn't. If their BOLD signal, on the other hand, depends on wf ; after the appropriate division, this will be apparent from the parameter estimates. Standardizing the PE or EV predictors could achieve a somewhat similar goal, but our approach maintains the original units (+1=desired outcome, -1=undesired outcome), while standardization would have generated units of variance.

15. How to generate fMRI parameter estimates that can be easily interpreted.

In our winning model, M2Out, the magnitude of prediction errors and expected values depends on a participant's wf value. Let us consider two participants, A and B, with $wf_A=0.1$ and $wf_B=0.9$. Now let us focus on a hypothetical first trial in which both try a symbol and witness a high-money high-shock outcome. Because it is their first trial, their expected values were still set at 0, $EV_S=EVM=0$, and because it is a high-shock, high-money outcome, $OutS=-1$ (high-shock) and $OutM=+1$ (high-money). Their prediction errors, according to M2Out will be different, despite starting from the same EV and witnessing the same outcome, due to their difference in wf . Let us focus on the shocks, where M2Out specifies that $PES=OutS*(1-wf)-EVS$. For participant A: $PES_A=-1*(1-0.1)-0=-0.9$; for B, $PES_B=-1*(1-0.9)-0=-0.1$. Now let us also consider two hypothetical BOLD responses. Response pattern 1 (BOLD1), assumes that in this voxel witnessing the same shock intensity triggers a similar BOLD response across all participants, independently of wf , so that $BOLD1_A=BOLD1_B=1$. In contrast, response pattern 2 (BOLD2), assumes that in this different voxel, participants that care more about shocks (like participant A) have a stronger response to witnessing the high-shock than participants (like participant B) that care less about shocks, with a magnitude that linearly depends on wf , e.g. $BOLD2_A=0.9$, $BOLD2_B=0.1$. The core question for our fMRI analysis is now to build a design matrix at the first level of the fMRI analysis that yield parameter estimates for PES that can be easily interpreted across participants to identify voxels in which response magnitude does or does not depend on personal preferences. If we directly enter the PES values from M2Out in our fMRI model (after mean-subtraction but without dividing by standard deviation), the parameter estimate b for the PES predictor for each participant in our one-trial example would simply be $BOLD/PES$. In the case of BOLD1, where the BOLD response is the same across participants, $b1_A=1/-0.9=-1.11$, and $b1_B=1/-0.1=-10$. Hence, despite the same BOLD response across participants, the parameter estimates are very different across participants. To use the parameter estimates as a measure of individual differences, this is not desirable. How can we avoid that effect? If we divide PES by $(1-wf)$, to reverse the effect of $(1-wf)$ in the formula to calculate PES, and we use $PES/(1-wf)$ as the predictor, this issue is remedied — $PES_A/(1-wf)=-0.9/0.9=-1$, $PES_B/(1-wf)=-0.1/0.1=-1$ — and the parameter estimates now reflect the equality of BOLD response magnitude as $b1A=1/-1=-1$, and $b1B=1/-1=-1$. How do the two methods compare in case BOLD2, where the response *did* depend on personal preference? When using the actual PES values from M2Out as predictors, $b2_A=0.9/-0.9=1$ and $b2_B=0.1/-0.1=1$. So here, BOLD response magnitude actually differed across the participants, but the parameter estimates do not ($b2_A=b2_B$). Again, this is not desirable. If using $PES/(1-wf)$, $b2_A=0.9/-1=-0.9$, and $b2_B=0.1/-1=-0.1$. Here, by looking at the parameter estimates, we can directly observe that the responses of A were stronger than those of B for the same outcome. Hence, dividing the PES by $(1-wf)$, ensures that participants with similar response magnitudes in the brain get similar parameter estimates, and participants with different response magnitudes in the brain have different parameter estimates.

In reality, our experiments involved more than one trial, and PES becomes a vector of values across trials. Dividing this vector with $(1-wf)$ will not alter what voxels have significant parameter estimates (i.e. $b \neq 0$), i.e. significant associations with PES, but the magnitude of the parameter estimate now becomes more easily interpretable across participants.

The same logic of course applies to PEM. Because $PEM = OutM * wf - EVM$, here we need to divide PEM by wf to make parameter estimates interpretable across individuals. The same logic also applies to EVS and EVM, that need to be scaled by $(1-wf)$ and wf , respectively, to make their parameter estimates suited for an analysis regarding the dependence on wf .

Supplementary Table 7 shows a numerical example across our 25 participants considering the first shock trial for PES. In this example we took the actual wf estimate of our 25 participants, again only considering the first trial of a block with $OutS = -1$. BOLD1 is, as above, a hypothetical BOLD response in a voxel where it is constant across participants witnessing the same difference between outcome and expected value, and BOLD2 is, as above, a response in a voxel where the response linearly depends on wf , with $BOLD2 = (1-wf)$. We then added noise to the BOLD response (uniform random noise between 0 and 0.2), and calculated the parameter estimates $b1$ (for BOLD1) and $b2$ (for BOLD2), either given PES or $PES/(1-wf)$. Finally, we calculated the correlation between wf and the parameter estimates, as we will in the fMRI analysis at the second level, to infer whether the BOLD response in a network or voxel does, or does not, depend linearly on the participants' preferences as captured by wf . As can be seen, using $PES/(1-wf)$, the correlation is close to zero for BOLD1 ($r = 0.06$) and very high for BOLD2 ($r = 0.98$), and a Bayesian test provides evidence of absence for $b1$ ($BF_{10} < 1/3$) and evidence of presence of an association for $b2$ ($BF_{10} = 1.6E14$). When using PES directly, however, we find significant, but intermediate associations for $b1$ and $b2$, despite the very different BOLD situations. This illustrates that using the PES values from M2Out directly does not provide parameter estimate values in fMRI that lend themselves to be easily interpreted with respect to the relationship between BOLD activity magnitude and wf .

An alternative approach may have been to standardize the PES vector prior to entering it into the model, as that would also tend to bring the PES predictors to be similar in scale across participants. However, in our research we favored the division by $(1-wf)$ and wf for PES/EVS and PEM/EVM , respectively, as this ensures that the transformed predictors uses the same units as we used for outcomes (with a PES value of -1 on the first high-shock) rather than depending on the overall variance of the prediction errors.

R2C6

Figure 8D shows regions with positive and negative loading on the AVPS. What is the neurophysiological or psychological basis for these different signs? To get the measure of loading on AVPS shown in Figure 8B, the authors actually sum up across all voxels (that show opposite relations to the APVS). Is it possible that this removes potentially interesting results of specific brain regions for both PES and PEM? The authors should provide information on this.

Mathematically, the multiplication performed to estimate the loading corresponds to a whole brain spatial correlation, with the main difference being that correlations are within the 0 to 1 range, while the loading is not standardized, and can therefore cover different ranges. This means that for the signature to load on a particular brain map, the pattern of activity across voxels (with some voxels being positive and others negative) has to be spatially similar between the two maps (signature and map of interest) in order to have a significant positive or negative loading. This final loading will always be the results of positive and negative values across voxels, but what counts is the similarity of the spatial distribution of these values between the two maps of interest. The hot colors in the maps in Figure 8e therefore represent the voxels in which the sign of the voxels of the two maps corresponds (i.e. both positive or negative), while the cold colors indicates voxels in which the sign does not corresponds (i.e. one negative and one positive). This means that for our map of interest to load to the signature significantly, a significantly higher amount of hot colors compared to cold colors represents spatially higher similarity across the pattern of activity of the two maps (i.e. significant positive loading), while a significant higher amount of cold colors represents the fact that the activity of the two maps is differently spatially distributed (i.e. significant negative loading, or significant lack of loading).

The use of neural signatures to interrogate networks associated with mental functions has become increasingly popular because it has more specificity than using individual ROIs and more sensitivity than mass-univariate analysis. It is more specific, because combining positive weights across multiple regions, typically showing BOLD signal increases while a specific mental function is recruited, with negative weights in regions typically showing BOLD signal reductions while that mental function is recruited, provides readouts that are better at revealing the specific involvement of that mental function than more traditional ROI analysis that only sum activity in a particular region. This approach is more sensitive than mass-univariate analysis because it can be performed without requiring the penalty of corrections for multiple comparisons necessary in mass-univariate analyses to avoid excessive false positive rates (see Kragel et al., 2018 for a recent review of the benefits of signatures). In our study, the sensitivity and specificity of the two signatures we use (the AVPS of the original manuscript, and the reward signature we now add to also examine PEM signals following reviewer comments) are particularly important, because they allow us to leverage the interpretative power of Bayesian analysis to provide evidence for the Null as well as for the H1.

This sensitivity however comes at the price of spatial interpretability: what parts of the brain contribute to the AVPS association with PES or the RS association with PEM remains unaddressed by the signature approach. Indeed, the signature approach assumes that brain functions are localized in distributed networks rather than specific voxels, and therefore does not aim to pinpoint a particular location. We therefore supplement these hypothesis driven analysis of specific networks with a more explorative approach using mass-univariate analysis to provide locations of regions that have signals at the voxel-level that are associated with prediction errors.

As so often in neuroscience, people disagree about what approach is best - neural signatures vs. mass-univariate approaches, and we hope that by presenting both we strike a compromise that is meaningful to those preferring either approaches.

R2C7

The authors may have mainly focused on PES because it correlates with PEM, but it looks like regions associated with PEM and PES do not entirely overlap (as shown in the supplementary tables). Have the authors estimated GLMs in which both PES and PEM are included as parametric modulators? It would be good to see whether such GLMs reveal qualitatively similar results or in fact distinct results for PES and PEM.

As previously mentioned in responses R2C4 and R1C3, we focused on PE_S for our interest in how the brain processes the pain of other individuals and how others' pain serves as a learning signal and decision making. As described in Supplementary Materials §17, the correlation between PE_S and PE_M is not very high (“The average correlation between the time courses of the parametric modulators for PE_S and PE_M was -0.26, ranging from -0.49 to -0.03.”) and we still have the power to dissociate PE_S and PE_M signals. The presence of a correlation was therefore not the reason for focusing on PE_S .

The individual subjects GLM specification does include both PE_S and PE_M as predictors for the outcome phase. What we could not include, due to the high correlation (or anticorrelation) was the combination of 4 predictors (OutS, OutM, PE_S , PE_M) for the outcome phase. We now use EV_S and EV_M as predictors of the decision phase instead

In the Methods section “fMRI data analyses” we now write:

The design matrix to analyze the fMRI data of the learning task included 13 regressors: (1) A decision regressor starting with the appearance of the two symbols and ending with the button press of the participant. (2) The outcome regressor was aligned with the presentation of the video and had a fixed duration of 2 seconds, corresponding to the duration of the stimulus. (3) A button-press regressor with zero duration was aligned to the moment of button-pressing. (4-5) The decision regressor had 2 parametric modulators (EV_M and EV_S of the chosen option); and (6-7) the outcome regressor 2 parametric modulators (PE_S and PE_M). The modulators were derived from the winning M2Out, then mean-subtracted and wf -normalized before being entered into the design matrix (see wf -normalization below and Supplementary Materials §15). (8-13) Finally, 6 regressors of no interest were included to model head translations and rotations.

Which parametric modulator to include was based on the results of parameter correlations and recovery. Prediction errors and actual outcomes could not be used within the same GLM as they were too highly correlated ($r(PE_S, Out_S)=0.741$, ranging from 0.750 to 0.866 across our 27 participants; $r(PE_M, Out_M)=0.749$, ranging from 0.781 to 0.862). We however examined if signals were associated with Out vs PE using the method suggested by Zhang and colleagues³⁹. During the outcome phase, we therefore only included PE_S and PE_M , which are only weakly correlated (average $r=-0.26$, ranging from -0.49 to -0.03 across the participants) and for which parameter recovery is robust (Supplementary Materials §17).

Finally, Figure 7c (also reported above in R2C4) now directly compares the largely distinct networks for PE_S and PE_M when looking at the results with a lower but still corrected threshold ($p_{unc}<0.01$ with $k=FWEc$ to ensure family wise error correction at the cluster-level) we indeed find brain signals to correlate with PE_M , which include regions normally associated with monetary reward. These regions only marginally overlap with regions correlating with PE_M , again showing that PE_M and PE_S are dissociable. Supplementary Figure 17 (below) specifies the regions of overlap.

Supplementary Figure 17. Overlap between PE_M and PE_S .

Cluster of activity obtained by inclusively masking PE_M (after the variance explained by wf is removed) inclusively masked with PE_S (after the variance explained by $(1-wf)$ is removed) at $p_{unc}<0.01$ without FWEc correction.

In the main text we now refer to these results as:

“Next, we performed a more explorative voxel-wise linear regression that predicts the parameter estimate of the PE_S modulator using a constant and wf . As expected for a region involved in valuation, we found the vmPFC to have signals covarying positively with PE_S (i.e. higher signals when shocks were lower than expected) with a more ventral cluster associated with PE_S in a way that depended on wf (Figure 7b, red), while a more dorsal vmPFC cluster showed an association with PE_S after removing variance explained by wf (Figure 7b, yellow). In addition, we find that the left somato-motor cortex, including BA4 and 3, also harbored signals covarying positively with PE_S in ways that depended on wf . The same voxel-wise analysis applied to PE_M only generates significant results when the cluster-cutting threshold was reduced to $p_{unc}<0.01$ (Figure 7c), and reveals striatal and ventral prefrontal clusters in line with those described in the literature for $PE_M>0$ after removing the variance explained by wf ^{24,36–38}, and clusters in the right cerebellum, ventral temporal lobe and hippocampus with PE_M signals that depend on wf (Figure 7d). Comparing the networks associated with PE_S and PE_M reveals that these networks are largely non-overlapping (Figure 7c). However, inclusively masking the contrast $PE_M>0$ with the contrast $PE_S>0$ reveals five clusters of overlap, the largest of which was in the vmPFC (Supplementary Figure 17), but these clusters remained too small to survive a whole brain FWE cluster-size correction.”

Figure 8 (presented above in R2C4) also illustrates the loading of PE_M on both the AVPS and RS.

Minor:

R2C8

I suggest that the authors include GLM analyses of decision utility as control analyses. Decision utility may also be represented in vmPFC, so it is important to differentiate the roles of vmPFC in representing decision utility and PEM.

With regard to vmPFC specifically, using the method of Zhang et al., 2022, we explored the association between BOLD activity and Out and EV within the region associated with EV_S . We also do this for the AVPS and RS. This is now reported in Supplementary Materials §18 (reported in R2C4).

More generally, with regard to Expected Values during the decision phase, we now report these analyses in Supplementary Figure 18 (also reported in R2C4)

R2C9

The authors claim “prediction” based on correlation analyses in some cases. This is misleading. Please rephrase all the relevant statements.

We carefully tracked all uses of the term ‘predict’ in the manuscript, and find them in two contexts: either in relation to regression analyses (e.g. “Next, we performed a more explorative voxel-wise linear regression that predicts the parameter estimate of the PES modulator using a constant and wf.” or “wf had external validity: it predicted how much money the participant gave to reduce shocks to the same confederate in a different task not requiring learning(Gallo et al., 2018) and did no better than trait measures of empathy and money attitude”), or in relation to our models (e.g. “To compare the ability of different models to predict this 11th trial at the single subject level, we then calculated a likelihood ratio (similar to a Bayes factor) for each participant”). As far as we can tell, both of them fall within the remit of the use of the term ‘prediction’ in statistics according to the OECD dictionary of statistical terms, which writes: “In general, prediction is the process of determining the magnitude of statistical variates at some future point of time. In statistical contexts the word may also occur in slightly different meanings; e.g. in a regression equation expressing a dependent variate y in terms of dependent x’s, the value given for y by specified values of x’s is called the “predicted” value even when no temporal element is involved.” (see <https://stats.oecd.org/glossary/detail.asp?ID=3792>). This is distinguished from ‘forecasting’ which has a definitely temporal meaning. Specifically, even our statement “wf had external validity: it predicted how much money the participant gave to reduce shocks to the same confederate in a different task not requiring learning(Gallo et al., 2018) and did no better than trait measures of empathy and money attitude” results from a multiple regression analysis, a context in which ‘predict’ is appropriate according to the OECD . We were therefore not certain what instances the reviewer refers to as being inappropriate or misleading instances of the use ‘predict’, but would be thankful for any specific pointers.

R2C10

On Page 15, the authors hypothesized that the M2Out model should be better than the other models, based on their model-free results. This is unclear (in fact, it comes out of the blue). Please provide a detailed explanation for why the model-free results support this hypothesis.

To reduce the overall word-count, we have now removed the section where we hypothesize that M2Out would be likely to be the best fit for the model, and we go straight to comparing the models using hierarchical Bayesian modeling. However, our reasoning to expect M2Out to fit best was as follows:

Explicit reports and their difference across Conflict and NoConflict blocks reveal that participants have explicit and separable representations of outcome-probability-differences across symbols in terms of money and shock, although these representations are more differentiated for the outcome that their choices indicate to bear more weight. Finally, devaluation performed by informing participants that their preferred outcome will no longer be delivered leads to an immediate switch away from their previously preferred option during Conflict. Together this supports the notion that choices in our conflictual conditions are dominated by a form of learning in which participants represent separable outcomes for self-money and other-shocks, with the individual preferences influencing the magnitude of their respective symbol-outcome association. In terms of our candidate neurocomputational models (Figure 5a,b), that participants have separable representations of symbol-outcome associations for self-money and other-shocks suggests that M2 models (M2Out or M2Dec), with their separate expected values for money and shocks, should outperform our M1 model that only tracks a conjoint expected value. That individual preferences influence the symbol-outcome association suggests that M2Out, in which outcomes, and therefore expected values, are influenced by individual preferences (wf) should outperform M2Dec in out Conflict trials.

R2C11

In Figure 6, the authors did not include results of the ambiguous group. The authors should at least put these results in the supplementary materials.

We now integrated the ambiguous participants in Figure 5 for the online experiment (also see below for the relevant section of that Figure) and in Supplementary Figure 7 for the fMRI data.

Figure 5. Model Comparison. [...] **c** Choices in the first 10 trials of the ConflictDropout blocks (Online experiment) as a function of preference together with the predictions by M1, M2Dec and M2Out. M0 always predicts 0.5 and is not shown. Choices and predictions averaged over all blocks, error bars: s.e.m. across participants. [...] **e** Change of participant choices and model predictions for the 10th to the 11th trial. Dashed lines connect 10th to 11th trials when money is removed (gray background), dotted lines, when shocks are removed (yellow background). 11th trial not included in model fitting. M2Out (black arrowheads) makes the best predictions for the 11th trial.

Supplementary Figure 7. Model Comparison.

[...] **c** Observed (in green, orange and dark blue) and predicted (in gray, light and dark violet) choices over the 10 trials of the Conflict condition for the fMRI data. **d** Mean LOOIC for all fMRI participants. [...]

R2C12

In Figure 2B, there should be a comma, rather than "+", between two EVs. If it is a typo, please correct it. If it is not, then the authors need to explain this

relation.

The reviewer is correct, and we now corrected the typo in what is now Figure 5a.

Reviewer #3 (Remarks to the Author):

Fornari, loupma and colleagues used a combination of behavioural tasks, computational modelling, and neuroimaging to understand the mechanisms that underlie decisions in a morally conflicting situation. Here, conflict is defined as the trade-off between selfish choices (more money for the participant) vs prosocial choices (a smaller number of shocks for the other player). The main behavioural results show individual differences in how these two options – prosocial vs selfish – are traded with each other. This preference or bias does not only impact decisions but also the learning about the options' contingencies. A computational model that includes an individually specific preference parameter (wf) that scales the prediction error during the outcome phase is predictive of participants' behaviour, in particular during the 11th trial where participants behaviour can be distinguished according to model-based vs. model-free learning. By using a devaluation task and computational modelling, the authors show that participants are in fact representing the options contingencies in a model-based manner. Finally, they make use of fMRI to understand the scaling of the prediction error by the model-derived parameter wf and find primarily a correlate in vmPFC.

The authors explain their approach and results in a clear and transparent manner. The combination of methodological approaches is very interesting, in particular the verification of computational parameters in a second independent helping task. While the online study offers a rich data set that has been analyzed in an interesting manner, the fMRI data set lacks the same rigorous approach. In the comments below, I give suggestions of additional analyses that might be interesting to look at when analyzing neural but also behavioural data.

Major points:

R3C1[model comparison]

The authors show that participants can be divided according to their choice preference (Figure 3) and that this preference also impacts learning about contingencies (Figure 4). However, the computational model that is selected is a model that includes the preference weighted parameter, wf , in the outcome phase and not choice phase. Would it not be more accurate to assume that both choice and PE are impacted by wf ? How does such a model including wf for both choice and outcome phase perform during model comparison?

We thank the reviewer for the suggestion. We now ran a model in which wf could influence both the decision (D) and outcome (O) phase. We refer to this model as M2DO.

In the results section we write:

“Finally, we also considered a model in which the wf is influencing both the decision and outcome phase (M2DO; Supplementary Figure 8). This model outperformed M2Dec, but not M2Out, with M2Out still predicting choices in critical conditions (MoneyDropout for Lucrative and ShockDropout for Considerate participants) closest to the actual choices of our participants. For parsimony, we therefore use M2Out for further analyses.”

And in the discussion:

“We also briefly explored a more general model (M2DO), that uses a parameter α to distribute the effect of preference across the outcome and decision phase. Fixing α to 1 or 0 transforms M2DO into M2Out or M2Dec, respectively, and fixing it to 0.5 - half-way between M2Out or M2Dec - outperform M2Dec but not M2Out, and we therefore opted to use

the simpler M2Out for the rest of our analyses. In the future, fitting α to each participant's specific devaluation behavior may enable the use α to quantify how much a given participant's preference pervades their expected value representations under conflict."

Supplementary

Figure 8. M2 with weighting at both Decision and Outcome (M2DO). **a** Model formalization for a model that distributes the effect of preference across the decision and outcome phase. Same as in Figure 5, except that an exponent α (shown in green) distributes how much of the weighting occurs during outcome and decision phase. The models M2Out and M2Dec represent special cases of this model with $\alpha=1$ and $\alpha=0$, respectively. Using $\alpha=0.5$ distributes weighting equally across the outcome and decision phase. τ =inverse temperature, LR =learning rate, EV =Expected Value, PE =Prediction Error, Out =Outcome. Subscript M =money and S =shocks. Outcomes are coded by value: high-shock $Out_S=-1$, low-shock $Out_S=+1$, high-Money $Out_M=+1$, low-money $Out_M=-1$. **b** Distribution over 4000 posterior draws of the summed log likelihood of the 11th trial over all participants multiplied by -2 to place values on the information scale as for LOOIC, with lower values indicating better predictions, and zero, perfect predictions. M2DO with $\alpha=0.5$ and M2Out performed similarly well and outperformed all other models. Note, that in principle, α could be fit to each participants data. However, we know that M2Out and M2Dec make identical behavioral predictions for the first 10 trials, and only start to make distinguishable predictions on the 11th trial, when dropout occurs. Accordingly, the first 10 trials cannot constrain the estimate of α , as $\alpha=1$ and $\alpha=0$ make identical prediction, and fitting would thus only depend on 8 trials per participants in our design. We therefore did not attempt such fitting, and simply compared the likelihood of 3 possible values ($\alpha=1$, i.e. M2Out, $\alpha=0.5$, i.e. M2DO, and $\alpha=0$, i.e. M2Dec). **c** Thicker green, orange and dark-blue lines indicated participant's choices at trial 11. Thinner gray, light magenta, light and dark violet lines indicate model predictions at trial 11. Gray and yellow backgrounds highlight whether money (gray) or shock (yellow) were removed at trial 11. Trial 11 is not included in model fitting. Black arrowheads indicate our M2Out is still closer to actual choices in the most interesting conditions, when a shift in decision is expected: moving toward the considerate option when the

money is removed for the Lucrative group, and moving toward the lucrative option when shock is removed for the Considerate group.
d Same as Supplementary Figure 7e, but for M2DO with $\alpha=0.5$.

R3C2 [fMRI analysis]

In the methods, the authors state that it is not possible to look at the decision phase (line 1020), because of the difficulties of time-locking and a possible confound of activity related to button press. The decision phase can be analysed as long as it is decorrelated from the outcome phase, which can either be done with a long enough ITI jitter or statistical decorrelation of variables across both phases. Here, the ITI jitter is long enough, which should allow to inspect neural correlates at time of the decision phase, where variables should be time-locked to the onset of the decision screen. Second, the authors should include a regressor time-locked to all button presses modelled as stick function to account for movement-related effects; as is usually done in fMRI analysis (eg see Wittmann et al., Neuron 2016 Supplementary Materials). This would also address their second point (line 1022). It would be interesting to see which brain regions correlate with a decision related

variable that computes the expected value between the considerate and lucrative choice. Again, as has been done before, if on a given trial participants choose the considerate option, the authors could include the difference between the chosen EV (shock) minus the unchosen EV (money). This variable can either be directly calculated, normalised and included into the GLM or alternatively, the chosen EV(s) and unchosen EV(m) can be included and their contrast can be calculated in SPM. Further, it would be interesting to extend this analysis by including a covariate into the second-level analysis that weights this value difference by *wf*.

Our concern with analyzing the decision phase in our paradigm is not the inability to analyze signals time-locked to the response screen, but that, because the symbols/options remain constant across a block, participants have no reason to wait for the response screen to decide which symbol to choose. Instead, as soon as the outcomes of the previous trials are revealed, participants can update their EVs, and identify the choice with the highest expected utility at any time in the interval between outcome and button press. They only need to await the response screen to know whether the right or left button will be the way to select the option they may have already decided to select. This is because on each trial of a block, participants know the option they will be offered, but what button will be associated with a particular option is randomly selected on each trial to differ motor planning to the response screen. In contrast, the precise timing of the actual decision of which option to choose is not constrained to the decision-screen, and accordingly, using this paradigm to look at the decision-phase is inefficient and such analysis are thus likely to generate false negative results - e.g. just because a given voxel does not have BOLD activity correlating with an EV-derived signal in an analysis aligned on the decision-screen does not indicate that it does not do so when the actual decision is taken. To avoid such misinterpretations, we would rather not present analyses of the decision-screen epoch in the main text.

It is important to mention that our aim in this study is to investigate how participants update the value of action alternatives by witnessing the facial expressions of others, that reveal the consequences of our action on them, and how this is influenced by preferences (as captured using *wf*) - *not* how they take decisions once such associations have been learned. If one wants to study the latter - doubtlessly interesting - question, it is preferable to use designs without a need for learning but in which the alternatives are only disclosed when the decision-screen is presented, as is for instance the case in the

elegant paradigm of Molly Crockett [Crockett et al. (2014) Harm to Others Outweighs Harm to Self in Moral Decision Making, PNAS 111:17320–25. <https://doi.org/10.1073/pnas.1408988111>; Crockett et al. (2017) Moral Transgressions Corrupt Neural Representations of Value. Nature Neuroscience 20:879–85. <https://doi.org/10.1038/nn.4557>]. This ensures that comparing the alternatives and deciding amongst them, occurs at a specific time, which can be captured using a regressor aligned to the decision-screen.

Despite these caveats, we have adjusted our first level models in the light of the reviewer’s suggestions, and now write in the methods:

“The design matrix to analyze the fMRI data of the learning task included 13 regressors: (1) A decision regressor starting with the appearance of the two symbols and ending with the button press of the participant. (2) The outcome regressor was aligned with the presentation of the video and had a fixed duration of 2 seconds, corresponding to the duration of the stimulus. (3) A button-press regressor with zero duration was aligned to the moment of button-pressing. (4-5) The decision regressor had 2 parametric modulators (EV_M and EV_S of the chosen option); and (6-7) the outcome regressor 2 parametric modulators (PE_S and PE_M). The modulators were derived from the winning M2Out, then mean- subtracted and wf-normalized before being entered into the design matrix (see wf-normalization below and Supplementary Materials §15). (8-13) Finally, 6 regressors of no interest were included to model head translations and rotations.

Which parametric modulator to include was based on the results of parameter correlations and recovery. Prediction errors and actual outcomes could not be used within the same GLM as they were too highly correlated ($r(PE_S, Out_S)=0.741$, ranging from 0.750 to 0.866 across our 27 participants; $r(PE_M, Out_M)=0.749$, ranging from 0.781 to 0.862). We however examined if signals were associated with Out vs PE using the method suggested by Zhang and colleagues (Zhang et al., 2020). During the outcome phase, we therefore only included PE_S and PE_M , which are only weakly correlated (average $r=-0.26$, ranging from -0.49 to -0.03 across the participants) and for which parameter recovery is robust (Supplementary Materials §17)”.

In the result section we now write:

“Our learning paradigm is optimized to capture the neural processes involved in learning symbol-outcome associations, i.e. processing the outcomes and computing prediction errors, and these processes can be assumed to occur when the outcomes are presented. In contrast, as the options amongst which to choose remain constant across the trials of a block, participants need not wait for the decision-screen to choose which symbol to choose on the next trial (Figure 1a). Attempts to isolate brain activity related to this uncertainty timed decision-process are therefore inefficient in our paradigm and are only included in the Supplementary Materials (Supplementary Figure 18). Choice paradigms in which options vary from trial to trial and are only revealed once a decision can be made are a more powerful means to isolate choice-related brain activity (see for example Crockett et al., 2017, 2014; Gallo et al., 2018).”

And Supplementary Figure 18 looks as follows:

Supplementary Figure 18. Decision phase and EV. **a** Results of the second level t-test $\text{Decision} > 0$, indicating voxels where BOLD signals during the decision phase (independently of prediction errors) are increased, $p_{\text{unc}} < 0.001$, $k = \text{FWEc} = 235$ voxels. **b** BOLD signal negatively correlating with the expected value for shock, $p_{\text{unc}} < 0.01$, $k = \text{FWEc} = 420$ voxels (Supplementary Table 16). The white circle indicates the cluster also surviving at $p_{\text{unc}} < 0.001$, $t = 3.48$, $k = \text{FWEc} = 137$ voxels, with the peak at the indicated coordinates. **c** The renders and slices visually compare activity correlating with $-\text{EV}_S$ (purple) and PE_S (yellow), both shown at $p_{\text{unc}} < 0.01$ with each relative FWEc. The images clearly show that $-\text{EV}_S$ and PE_S can be dissociated, with several clusters independently correlating with $-\text{EV}$ and PE_S , and only four clusters (not surviving an FWEc correction at $p_{\text{unc}} < 0.0$) overlapping between PE_S and $-\text{EV}_S$. Note that $-\text{EV}_S$ is modeled as a parametric modulator of the decision phase, while PE_S as a modulator of the outcome phase. Expected values computed during PE_S cannot be isolated. **d** Overlapping clusters between $-\text{EV}_S$ and PE_S at $p_{\text{unc}} < 0.01$ without any cluster correction. Description based on the Anatomy toolbox for SPM, and the statistical whole brain tables from SPM12. As in Figure 7, all results are FWE cluster corrected at $p_{\text{unc}} < 0.05$, following cluster cutting at $p_{\text{unc}} < 0.001$ or $p_{\text{unc}} < 0.01$, specified using the critical FWE cluster size FWEc as indicated in figure panels. Renders are created based on the cortex_20484 surface from spm12, slices are taken from the average T1 anatomical scan from our participants.

With regard to our signatures, we find that the AVPS signature is associated with EV_S in a way that is negatively correlated with wf (Supplementary Table 9), i.e. participants that value shocks more (i.e. low wf) recruit the pain observation network more when choosing options with high EV_S (i.e. expected to reduce shocks).

		normality		versus zero						correlation with wf			
test		Shapiro	p	t	w	df	p(t)	p(w)	BF ₁₀ (t)	BF ₁₀ (w)	Tau	p _(Tau)	BF ₁₀ (Tau)
AVPS	PESxAVPS	0.967	0.567	-5.461	19	24	0.00001	0.00002	1702.9 4	416.25	0.01	0.944	0.257
	PEMxAVPS	0.964	0.505	1.041	197	24	0.308	0.367	0.34	0.37	-0.03	0.833	0.262
	EVSxAVPS	0.961	0.441	0.899	199	24	0.377	0.339	0.3	0.31	-0.29	0.042	1.855
	EVMxAVPS	0.904	0.022	2.806	248	24	0.01	0.02	4.86	6.21	-0.043	0.761	0.268
RS	PESxRS	0.984	0.955	3.278	267	24	0.003	0.004	12.71	51.29	-0.19	0.183	0.601
	PEMxRS	0.987	0.981	2.816	258	24	0.01	0.009	4.96	13.38	0.15	0.293	0.436
	EVSxRS	0.938	0.134	-2.052	100	24	0.051	0.096	1.26	1.12	-0.037	0.797	0.265
	EVMxRS	0.897	0.016	-0.581	130	24	0.567	0.396	0.25	0.32	0.157	0.272	0.458

Supplementary Table 9: Supplementary Signature Analyses.

For both the AVPS (top) and RS (bottom) signature, the table details for each loading the result of a shapiro normality test, including the test value and associated p value, followed by a two-tailed test of the loading against zero, where t indicates a student t -test and W a wilcoxon test with the latter particularly relevant for cases where normality is violated. A (t) following p or BF_{10} specifies that these values come from the student t -test, a (w) , from the Wilcoxon test. The final 3 columns reflect a Kendall's τ test of the correlation between wf and the loading. Kendall's τ was used because of the non-normal distribution of wf . Green numbers highlight significant results (p values below 0.05 or BF_{10} values above 3), red values indicate evidence of absence ($BF_{10} < \frac{1}{3}$).

R3C2.1 [fMRI analysis]

Did the authors normalize their regressors before including them into the GLM? I might have overlooked it, however this should be the case.

We thank the reviewer for pointing out this important aspect of the analysis. Because the dependence on wf of the brain activity in relation to PE_S and EV_S are of particular interest in our analysis, it was important for us to generate parameter estimates that have interpretable units of magnitude, that we can compare across participants, to see if participants that weigh shocks more strongly in their choices have larger parameter estimates. In particular, because outcomes have clear-cut magnitudes (1 vs -1), we reasoned that it would be helpful to maintain these units unchanged. Using standardization would potentially complicate such comparison, because temperature and learning rate may influence the variance of the prediction errors and expected values, and thereby lead to different standardization factors for different participants. Instead of dividing the PE_S and EV_S time-series of each participant with their standard deviation, we therefore divided them by $(1-wf)$, and instead of dividing PE_M and EV_M with their standard deviation, we therefore divided them by wf . We now added Supplementary Materials §15 to illustrate the effect of this procedure through a simple-to-follow example.

15. How to generate fMRI parameter estimates that can be easily interpreted.

In our winning model, M2Out, the magnitude of prediction errors and expected values depends on a participant's wf value. Let us consider two participants, A and B, with $wf_A=0.1$ and $wf_B=0.9$. Now let us focus on a hypothetical first trial in which both try a symbol and witness a high-money high-shock outcome. Because it is their first trial, their expected values were still set at 0, $EV_S=EVM=0$, and because it is a high-shock, high-money outcome, $OutS=-1$ (high-shock) and $OutM=+1$ (high-money). Their prediction errors, according to M2Out will be different, despite starting from the same EV and witnessing the same outcome, due to their difference in wf . Let us focus on the shocks, where M2Out specifies that $PES=OutS*(1-wf)-EVS$. For participant A: $PES_A=-1*(1-0.1)-0=-0.9$; for B, $PES_B=-1*(1-0.9)-0=-0.1$. Now let us also consider two hypothetical BOLD responses. Response pattern 1 (BOLD1), assumes that in this voxel witnessing the same

shock intensity triggers a similar BOLD response across all participants, independently of wf , so that $BOLD1_A=BOLD1_B=1$. In contrast, response pattern 2 (BOLD2), assumes that in this different voxel, participants that care more about shocks (like participant A) have a stronger response to witnessing the high-shock than participants (like participant B) that care less about shocks, with a magnitude that linearly depends on wf , e.g. $BOLD2_A=0.9$, $BOLD2_B=0.1$. The core question for our fMRI analysis is now to build a design matrix at the first level of the fMRI analysis that yield parameter estimates for PES that can be easily interpreted across participants to identify voxels in which response magnitude does or does not depend on personal preferences. If we directly enter the PES values from M2Out in our fMRI model (after mean-subtraction but without dividing by standard deviation), the parameter estimate b for the PES predictor for each participant in our one-trial example would simply be $BOLD/PES$. In the case of BOLD1, where the BOLD response is the same across participants, $b1_A=1/0.9=-1.11$, and $b1_B=1/0.1=-10$. Hence, despite the same BOLD response across participants, the parameter estimates are very different across participants. To use the parameter estimates as a measure of individual differences, this is not desirable. How can we avoid that effect? If we divide PES by $(1-wf)$, to reverse the effect of $(1-wf)$ in the formula to calculate PES, and we use $PES/(1-wf)$ as the predictor, this issue is remedied — $PES_A/(1-wf)=-0.9/0.9=-1$, $PES_B/(1-wf)=-0.1/0.1=-1$ — and the parameter estimates now reflect the equality of BOLD response magnitude as $b1A=1/-1=-1$, and $b1B=1/-1=-1$. How do the two methods compare in case BOLD2, where the response *did* depend on personal preference? When using the actual PES values from M2Out as predictors, $b2_A=0.9/0.9=1$ and $b2_B=0.1/0.1=1$. So here, BOLD response magnitude actually differed across the participants, but the parameter estimates do not ($b2_A=b2_B$). Again, this is not desirable. If using $PES/(1-wf)$, $b2_A=0.9/-1=-0.9$, and $b2_B=0.1/-1=-0.1$. Here, by looking at the parameter estimates, we can directly observe that the responses of A were stronger than those of B for the same outcome. Hence, dividing the PES by $(1-wf)$, ensures that participants with similar response magnitudes in the brain get similar parameter estimates, and participants with different response magnitudes in the brain have different parameter estimates.

In reality, our experiments involved more than one trial, and PES becomes a vector of values across trials. Dividing this vector with $(1-wf)$ will not alter what voxels have significant parameter estimates (i.e. $b \neq 0$), i.e. significant associations with PES, but the magnitude of the parameter estimate now becomes more easily interpretable across participants.

The same logic of course applies to PEM. Because $PEM=OutM*wf-EVM$, here we need to divide PEM by wf to make parameter estimates interpretable across individuals. The same logic also applies to EVS and EVM, that need to be scaled by $(1-wf)$ and wf , respectively, to make their parameter estimates suited for an analysis regarding the dependence on wf .

Supplementary Table 7 shows a numerical example across our 25 participants considering the first shock trial for PES. In this example we took the actual wf estimate of our 25 participants, again only considering the first trial of a block with $OutS=-1$. BOLD1 is, as above, a hypothetical BOLD response in a voxel where it is constant across participants witnessing the same difference between outcome and expected value, and BOLD2 is, as above, a response in a voxel where the response linearly depends on wf , with $BOLD2=(1-wf)$. We then added noise to the BOLD response (uniform random noise between 0 and 0.2), and calculated the parameter estimates $b1$ (for BOLD1) and $b2$ (for BOLD2), either given PES or $PES/(1-wf)$. Finally, we calculated the correlation between wf and the parameter estimates, as we will in the fMRI analysis at the second level, to infer whether the BOLD response in a network or voxel does, or does not, depend linearly on the participants' preferences as captured by wf . As can be seen, using $PES/(1-wf)$, the correlation is close to zero for BOLD1 ($r=0.06$) and very high for BOLD2 ($r=0.98$), and a Bayesian test provides evidence of absence for $b1$ ($BF_{10}<1/3$) and evidence of presence of an association for $b2$ ($BF_{10}=1.6E14$). When using PES directly, however, we find significant, but intermediate associations for $b1$ and $b2$, despite the very different BOLD situations. This illustrates that using the PES values from M2Out directly does not provide parameter estimate values in fMRI that lend themselves to be easily interpreted with respect to the relationship between BOLD activity magnitude and wf .

An alternative approach may have been to standardize the PES vector prior to entering it into the model, as that would also tend to bring the PES predictors to be similar in scale across participants. However, in our research we favored the division by $(1-wf)$ and wf for PES/EVS and PEM/EVM, respectively, as this ensures that the transformed predictors uses the same units as we used for outcomes (with a PES value of -1 on the first high-shock) rather than depending on the overall variance of the prediction errors.

In the Methods section “fMRI Data Analysis” we then write and refer to this Supplementary material as follows:

“*wf-normalization*: Because we are interested in whether PE_S or PE_M representations depend on wf or not, we divided PE_S with $(1-wf)$ and PE_M with wf before entering them into the design matrix. As a result, the first PE_S value in the parametric modulator would always be $PE_S=-1$ if it was a high-shock outcome or $PE_S=+1$ for a low-shock outcome, independently of the participant's wf value. If signals covary with PE_S in a way that does not linearly depend on wf , the parameter estimate across participants (β_{PES}) would violate $H_0:\beta_{PES}=0$, but not $H_0:r(\beta_{PES}=0,wf)=0$. If signals covary with PE_S in a way that does linearly depend on wf , it would violate both of these H_0 . Note that for outcomes, the coding was +1 for good outcomes (i.e. high money or low shock) and -1 for bad outcomes (i.e. low money or high shock). EV and PE follow that polarity. The same was applied to EV_S and EV_M , which were divided by $(1-wf)$ and wf , respectively. This approach is illustrated with an example in Supplementary Materials §15.”

R3C2.2 [wf]

Weighting the prediction error in the outcome phase by wf is in itself interesting. However, it seems rather arbitrary to divide the PE by the wf parameter. Equally, one could multiply it, in line with the computational model. However, both approaches divide a variable that changes trial by trial (as can be seen in Figure 2), by one static preference that is fitted to all data. Alternatively, the authors can include wf as a covariate into the second-level analysis and inspect whether the PE of shock or money covariates with wf . This would also allow you to see whether there are individual areas that covariate with wf , instead of calculating a dot product between a whole network (AVPS).

As mentioned in the above reply, we now provide a detailed analysis of why and how we normalize the PE values (Supplementary Materials §15). The reason to divide the PE_S by $(1-wf)$ or the PE_M by wf is to undo the effect that the weighting factor has on PE_S and PE_M in M2Out, so that participants with different wf now have similar PE_S or PE_M predictors in the fMRI GLM when witnessing the same outcome, so as to be able to then examine whether the parameter estimates associated with the so-normalized PE_S or PE_M vary with wf . Multiplying with wf or $1-wf$ would not undo but square the effect. Once this is done, we can then indeed use wf as a second level voxel-wise regressor, and do so in Figure 7.

Figure 7. Key fMRI results. **a** Voxels where BOLD signals are increased during the outcome phase, independently of prediction errors, (Outcomes >0), $p_{unc}<0.001$, $k=FWEc=903$ voxels (Supplementary Table 8). **b** Red: regression between PE_S and $1-wf$ ($p_{unc}<0.001$, $k=FWEc=181$ voxels, Supplementary Table 10). This identifies voxels with signals that increase more for shock outcomes that are less intense than expected (i.e. positive PE_S) in participants that place more weights on shocks (high $1-wf$ values). Yellow: results of the contrast constant >0 in the same linear regression with $(1-wf)$ ($p_{unc}<0.001$, $k=FWEc=167$ voxels). This identifies voxels with signals that increase with increasing PE_S after the variance explained by $(1-wf)$ is removed (Supplementary Table 11). **c** Cyan: $PE_M > 0$ (after removing variance explained by wf ; Supplementary Table 12) at $p_{unc}<0.01$, $k=FWEc=542$ voxels. Purple: $PE_S > 0$ presented the same cluster-cutting threshold of $p_{unc}<0.01$, $k=FWEc=950$ voxels (Supplementary Table 11) in order to visualize that the two networks are largely distinct (but see Supplementary Figure 17 for overlap). **d** Correlation between PE_M and wf , $p_{unc}<0.01$, $k=FWEc=1642$ voxels (Supplementary Table 13). **e** Results of the contrast $PE_S * LR > 0$, $p_{unc}<0.001$, $k=FWEc=121$ voxels (Supplementary Table 14), which represent the BOLD signal associated with shock value updating. Note that across all panels, results are FWE cluster corrected at $\alpha<0.05$ using (i) cluster cutting at $p_{unc}<0.001$ (panels a, b, e) or (ii) $p_{unc}<0.01$ (panel c, d) then excluding clusters with an extent below the critical FWE cluster size FWEc. Cluster-extent threshold size always indicated in figure panels. Renders are created based on the cortex_20484 surface from

spm12, slices are taken from the average T1 anatomical scan from our participants.

R3C2.3 [activity in the ventral striatum]

It is surprising that there is no activation in ventral striatum for prediction errors, which is observed across many studies. More related to the current study, Lockwood et al. (2020, PNAS) shows prediction errors of pain avoidance for other (and self) in VS which is conceptually similar to the current aim. Is there activation in VS at subthreshold level?

Indeed. In the original manuscript we corrected results for multiple comparisons at FWE=0.05 using an extent threshold ($k=FWEc$) following a cluster-cutting threshold of $p<0.001$. Using this cluster-cutting threshold, the results for PE_M do not survive $k=FWEc$. An alternative approach to achieve FWE=0.05 is to use a more permissive cluster-cutting threshold and a larger extent threshold. In SPM, a user can choose whichever p_{unc} threshold they desire, and SPM will then use random field theory to calculate a FWEc minimum cluster-size that ensures that clusters survive this threshold under the null hypothesis no more

often than $\alpha=0.05$ of cases, thereby ensuring $FWE=0.05$ at the cluster level. Doing so following a cluster-cutting threshold of $p_{unc}<0.01$ yields $FWE_c=542$ for PE_M , and we therefore do find significant clusters with signals that covary with PE_M , including ventral striatal activity (right putamen and accumbens in particular). We now show this network in Figure 7 (also reported above in response to R3C2.2). Although this network is largely non-overlapping with the PE_S network at the same cluster-cutting threshold (Figure 7c), the two overlap in the vmPFC and the orbitofrontal cortex, which could serve the brain to calculate a common-currency for decision-making.

We now report all these findings in the results:

“The same voxel-wise analysis applied to PE_M only generates significant results when the cluster-cutting threshold was reduced to $p_{unc}<0.01$ (Figure 7c), and reveals striatal and ventral prefrontal clusters in line with those described in the literature for $PE_M>0$ after removing the variance explained by wf (Bartra et al., 2013; Fouragnan et al., 2018; Rushworth and Behrens, 2008), and clusters in the right cerebellum, ventral temporal lobe and hippocampus with PE_M signals that depend on wf (Figure 7d). Comparing the networks associated with PE_S and PE_M reveals that these networks are largely non-overlapping (Figure 7c). However, inclusively masking the contrast $PE_M>0$ with the contrast $PE_S>0$ reveals five clusters of overlap, the largest of which was in the vmPFC (Supplementary Figure 17), but these clusters remained too small to survive a whole brain FWE cluster-size correction.”

PE_M also positively loads on our newly developed reward signature (Figure 8c).

Figure 7. Key fMRI results. **a** Results of the contrast $Outcomes > 0$ indicating voxels where BOLD signals during the outcome phase (independently of prediction errors) are increased, $p_{unc} < 0.001$, $k = FWEc = 903$ voxels (Supplementary Table 8). **b** Red: regression between PE_S and $1-wf$ ($p_{unc} < 0.001$, $k = FWEc = 181$ voxels, Supplementary Table 11). This identifies voxels with signals that increase more for shock outcomes that are less intense than expected (i.e. positive PE_S) in participants that place more weights on shocks (high $1-wf$ values). Yellow: results of the contrast $constant > 0$ in the same linear regression with $(1-wf)$ ($p_{unc} < 0.001$, $k = FWEc = 167$ voxels). This identifies voxels with signals that increase with increasing PE_S after the variance explained by $(1-wf)$ is removed. Supplementary Table 10 **c** Results of the contrast $PE_M > 0$ (after removing variance explained by wf ; Supplementary Table 12) in cyan only survived a corrected threshold of $p_{unc} < 0.01$, $k = FWEc = 542$ voxels. $PE_S > 0$ results in purple are therefore presented here using the same cluster-cutting threshold, $p_{unc} < 0.01$, $k = FWEc = 950$ voxels, in order to visualize that the two networks are largely distinct (but see Supplementary Figure 17 for overlap). **d** Correlation between PE_M and wf , $p_{unc} < 0.01$, $k = FWEc = 1642$ voxels (Supplementary Table 13). **e** Results of the contrast $PE_S * LR > 0$, $p_{unc} < 0.001$, $k = FWEc = 121$ voxels (Supplementary Table 14), which represent the BOLD signal associated with shock value updating. Note that across all panels, results are FWE cluster corrected at $\alpha < 0.05$ using (i) cluster cutting at $p_{unc} < 0.001$ (panels a, b, e) or (ii) $p_{unc} < 0.01$ (panel c, d) then excluding clusters with an extent below the critical FWE cluster size FWEc. Cluster-extent threshold size always indicated in figure panels. Renders are created based on the cortex_20484 surface from spm12, slices are taken from the average T1 anatomical scan from our participants.

R3C3 [behavioural and neural regression analysis]

Most analyses rely on the initial categorisation of three preference groups. This categorisation is somewhat arbitrary and let's someone wonder what the ambiguous group entails. For example, in Supplementary Figure 3, it shows clearly that this group was also motivated to prevent harm to others compared to other motivations. Noteworthy, I still think it is an acceptable approach as the authors show analyses that rely on more continuous measures (Figure 4E). However, it might be informative not only to analyze a summary measure such as proportion of lucrative or considerate choices, but instead look at a trial-by-trial choice analysis. Such an analysis could potentially not only show a choice bias but also a learning bias and might also help to link more directly to the fMRI GLM. A logistic regression predicting trial-by-trial choices (eg, 1= lucrative, 0=considerate) could be predicted by model-derived variables such as expected value difference between the lucrative and considerate choice and the prediction error at the last trial/ last encounter of the choice.

We thank the reviewer for this consideration. Perhaps we do not quite understand the approach proposed, because we would like to clarify that the Stan models we use (see our Figure 5a) are in fact trial-by-trial choice analysis using a logistic regressions of choice based on the difference in EVs, EVs which are updated using the PE from the last encounter with that choice. Stan performs these logistic regressions in a fully Bayesian way, rather than a maximum likelihood estimation, but does provide the same inference: it predicts trial-by-trial choices (y-pred in the stan codes provides in osf), and examines how well these logistic regression predictions match the data. Indeed, in Figure 5, the panels c-f all show metrics of how these trial-by-trial predictions match the actual trial-by-trial choices, separately for the 3 logistic regressions we used (M1, M2Dec and M2Out). Specifically, panel c shows these trial-by-trial predictions averaged over participants and blocks, panel d uses the LOOIC measure to approximate how much each left-out trial prediction matches the observed choice, panel e, how much the predictions for the 11th trial match the data averaged over blocks, and panel e how well the 11th trial predictions match the observed trial-by-trial choices using log-likelihood. All of these are trial-by-trial predictions based on logistic regressions performed in a fully Bayesian fashion using RStan. For visualization, we use metrics that summarize the trial-by-trial predictions, but the predictions and analyses are performed at the trial level. That is to say, stan does not predict average choices, but binary trial-by-trial choices. Visualizing predictions for individual trials in our paradigm without averaging over blocks and/or subjects is however not very diagnostic, because each trial is just a binary choice. We hope this clarifies that performing another logistic regression would end up duplicating what has already been done at the model level.

Because Ambiguous participants are classified as such because their decisions fail to deviate from randomness in a binomial test, investigating whether learning occurs during the Conflict trials would be a form of double dipping. However, we can use the Bayesian logistic regression to test whether EV estimates at the end of the 10th trial contribute significantly to the prediction of behavior after dropout on the 11th trial. This can be done in the Bayesian framework, by asking whether the behavior at the 11th trial is more likely given our winning model including EV values (M2Out) than given a random process (M0). We therefore compared the log_likelihood of M2Out and M0 on the 11th trial for Ambiguous participants, and found that log_likelihood was numerically but not significantly higher for M2Out (Wilcoxon test, $W=182$, $n=24$ ambiguous participants, one-tailed $p=0.187$, $BF_{+0}=0.402$). However, there is evidence that some Ambiguous participants actually did demonstrate significant learning, but in the form of preferences that alternate between blocks (see Supplementary Materials §12, also reported in response to reviewer #2, R2C2). Such alternating preferences then cancel each other out overall, creating a flat overall learning

profile, and are difficult to capture using logistic regressions with constant wf . We also performed several analyses associating wf with questionnaires (Supplementary Materials §5, 20, 21) and choices in the helping task.

R3C3.1

This would allow to inspect not only choice bias, that would be captured in the intercept, but also the learning bias, which would be the regression weight of the prediction error on the next choice. The same variables could then also be used to analyse the decision phase in the fMRI analysis.

If we understand this comments correctly, our M2Out model actually instantiates the proposed logistic regression, albeit in a Bayesian framework, and the learning bias suggested is instantiated in our model as the learning rates (LR_S and LR_M) which determine how strongly a given PE influences the next decision by altering the EV_S (See Figure 5a, right-most column). However, the learning rate and PE values are processes that we would expect to influence brain activity when outcomes are revealed and the value updating occur, rather than when the decision-screen is shown, at which time, the EV values should drive decisions.

Specifically to address the reviewer's comment of whether LR influences brain activity, we performed a regression model at the second level that examines whether we have voxels with a parameter estimate for PE_S that varies with LR_S and a parameter estimate for PE_M that varies with LR_M . This captures brain networks involved in the actual value-updating (which depends on LR in reinforcement learning theory) rather than in prediction-error calculation (which does not depend on LR in reinforcement learning theory). Results are now presented in Figure 7 (reported above in response to R3C2.3), and reveals a sizable network of prefrontal and in particular, medial prefrontal regions.

R3C4 [conceptual novelty]

I appreciate the authors' methodological efforts (devaluation task, online task, fMRI study, an array of self-reports and questionnaires), however, the study resembles quite closely to other studies that inspect the trade-off between choosing beneficial options for self vs. others. In particular, one previous study by Lockwood et al. (2020, PNAS) inspects the computational mechanisms of model-based and model-free learning in a two-step task where participants choose on behalf of another person or for themselves whether they want to administer shocks or reward. One difference between studies is the type of decision participants make here compared to the other study: trade between something good for oneself or another person. The authors should feel free to argue against this, but I think to increase the scientific contribution of the current study, it would be interesting to inspect the decision phase as mentioned before by looking at the decision correlates that bias choices instead of only focusing neural analyses on the outcome phase.

We hope that our reframing in line with the advice of Reviewer2 (see also R2C1 and R1C1) now focuses the attention of the reader to the core innovation of our study, which is to investigate the format through which participants *learn* the morally relevant action-outcome associations when two outcomes are pitted against each other: do participants track these action-outcome associations separately, or do

they combine them into a single, combined expected value, and do they so in ways that do, or do not, depend on preference.

Our results show that for the majority of participants, we have evidence that they do track separable representations of the associations with pain to others and money for self. We further show that individual variance in which of these conflictual outcomes dominates decision do significantly, but only partially, bias the representation of the action-outcome associations. That is to say, people's choices after devaluation and their explicit reports show that they have above-chance understanding of the most favorable outcome even for the outcome that their decisions under conflict seem to ignore (i.e. a person with lucrative preference is aware of what symbol is causing shocks to others, and a person with considerate preference is aware of what symbol could earn them more money), however, this knowledge is not as differentiated as for the outcome that they prioritize in their decisions. This insight into the learning process has important implications for moral responsibility and individual variance could provide a powerful handle on a deeper understanding of antisocial behavior in a computational psychiatry framework: would antisocial individuals show a stronger bias?

Our aim was thus to shed light on the learning process through which action-outcome associations are *learnt*, not how decisions are taken based on learned associations. The latter has already been done elegantly in paradigms in which participants need to choose between two combinations of self-money and other-pain by the group of Molly Crockett. Importantly, although our computational model does include an explicit formulation of the decision algorithm (see Fig. 5a), the fMRI paradigm is optimized to capture brain activity in the learning phase (i.e. when outcomes are revealed and EVs are updated), not the decision phase. This is because to capture the decision-process, participants must be forced to confine their decision process to a specific moment in time for an fMRI GLM to capture the brain activity relating to that process. To do so, participants must be unaware of the decision options they will have to choose from before that epoch. This is not the case in the kind of learning paradigm we use, in which participants know perfectly well that for 10 trials they must choose between the same two options. However, choice paradigms without learning, in which decision-screens include trial-unique choices, e.g. "2€ for a shock intensity 8 vs. 0.5€ for a shock intensity 2" on trial 1, and "4€ for shock intensity 9 vs. 0.3€ for shock intensity 1" on trial 2 etc., are means to effectively probe this process that we do not aim to study this paper.

With those caveats in mind, we do respect the reviewer's interest in the brain activity during our decision-screen, and, as mentioned above, now include such an analysis in the Supplementary Figure 18 (also reported in R3C2).

Minor points

- The introduction and discussion are very long. Can you reduce it to the most important topics? For example, it is good to give an overview about the methods in the introduction, however this overview is very detailed and could instead be absorbed into the methods section.

We thank the reviewer for this comment. We have shortened the introduction and discussion substantially, in particular, by moving the sections describing our Bayesian models to the relevant section of the results, and moving a number of other methodological details to the methods and result sections. We agree that this improves the readability of the manuscript in contrast to 'front loading' so much of the information

that only becomes relevant later. We have also shortened the discussion and hope that this reshaped manuscript will be faster paced and more rewarding to read.

• It is great that you use a second independent task to verify your computational parameter. In many other studies, computational parameters are correlated with psychological questionnaires to verify their external validity. However, going forward it might be worth considering the inclusion of an independent short task. For transparency and interest, can you please include the correlation between the wf and the psychological questionnaires; all included questionnaires are sensible and it would be interesting to see how they relate to wf.

We thank the reviewer for their encouraging comments on our Helping task. Regarding the correlation between *wf* and the psychological questionnaires, the questionnaires differed across Online and fMRI tasks (see new Supplementary Figure 1 for a complete task overview).

Supplementary Figure 1. Experimental procedures. For both the Online (a) and fMRI (b) experiments the schema details the sequence of procedures, tasks (with included conditions) and questionnaires each participant went through.

For the fMRI task:

We had included the MAS and the IRI, and now report their correlations with *wf* in Supplementary Material §14. None were significant.

	Coefficient	Bayesian					Frequentist		Kendall's Tau with wf
		P(incl)	P(incl data)	BF _{incl}	Mean	SD	t	p	
	Intercept	1	1	1	2.643	0.197	2.069	0.058	
	wf	0.5	0.922	11.74	-1.669	0.732	-3.009	0.009	
IRI	Fantasizing	0.5	0.277	0.383	-0.006	0.031	-0.433	0.671	-0.070
	Perspective Taking	0.5	0.261	0.354	-0.003	0.027	-0.575	0.574	-0.249
	Empathic Concern	0.5	0.284	0.396	0.008	0.029	0.899	0.384	-0.153
	Personal Distress	0.5	0.262	0.356	-0.002	0.024	-0.190	0.852	0.053
MAS	Power-Prestige	0.5	0.269	0.367	-0.001	0.025	0.170	0.868	0.019
	Retention-time	0.5	0.282	0.393	-0.007	0.029	-1.042	0.315	0.120
	Distrust	0.5	0.405	0.682	0.024	0.039	1.820	0.090	0.109
	Anxiety	0.5	0.311	0.451	-0.014	0.036	-1.137	0.274	0.243

Supplementary Table 6. Bayesian linear regression posterior summary of coefficients.

The table summarizes the Bayesian linear regression model comparison for models explaining the donation in the Helping task using the *wf* of the Learning task (estimated by fitting M2Out on the first 10 trials of the Conflict conditions), the subscales of the IRI⁵⁶ (FS, PT, EC and PD) and MAS⁴⁶ (Power-Prestige, Retention-time, Distrust and Anxiety). The first column indicates the variable under consideration, followed by the prior probability of inclusion (P(incl)), the posterior probability of inclusion (P(incl|data)), the BF_{incl} indicating how much more likely models including a variable are compared to the average of those not including this variable. BF_{incl}>3 is considered moderate, and BF_{incl}>10 strong, evidence that a variable explains donation. BF_{incl}<1/3 indicates moderate evidence against a variable explaining donation. While *wf* shows a BF_{incl}>11, all other variables have a BF_{incl}<0.7. The most likely model given the data is therefore the one including the intercept and the *wf* alone²⁸. Mean and SD represent the estimates of the weight of the parameter in the regression. The negative weight for *wf* indicates that people with higher *wf* (i.e. with more lucrative preferences) donate less to help others. A frequentist analysis (with its relevant *t* and *p* value) reaches the same conclusions, with *p*=0.009 for *wf* but all other *p*>0.05. Significant *p* values and BF>3 are highlighted in bold and green. The final column indicates the correlation between the psychological tests and the *wf*. None of these correlations were significant (all *p*>0.05).

For the Online experiment:

We removed these two questionnaires for two reasons. First, because the results of the fMRI suggested limited association. Second, because we needed to add several conditions to the online experiment to disentangle the different models (Dropout, NoConflict etc), and including these questionnaires would have risked reducing the number of participants that complete all tasks or reducing attention throughout the experimental session. However, we included a number of other questionnaires. We now report these results and associations with *wf* in Supplementary Materials §20 and 21:

Stress Tolerance Short Questionnaire (STSQ). As mentioned in the methods and in Supplementary Materials §20, this questionnaire was administered at the beginning of the online session to assess participants' vulnerability to stress and the STSQ score was used to advise for or against participation in

the study. Correlating the STSQ score with the wf remains inconclusive (Kendall's Tau=-0.124; $BF_{10}=0.529$; $p=0.113$). This result is now mentioned in the Supplementary Materials together with the STSQ at the end of Supplementary Materials §20.

Participant's rating of painfulness. In Supplementary Materials §21, we now write: "Soon after the learning task, participants were shown two randomly chosen videos from the pool used during the learning task ("Short pain rating" in Supplementary Figure 1). One was an example from the no-shock outcome and one was from the shock outcome. Participants were asked to rate how much pain they thought the person in the video felt ("How much pain do you think the person felt?"), using a scale from 0 to 10 in which 0=no pain: no sensation on the hand and 10=excruciating painful sensation. We then correlated the difference between the rating for the painful shock and non-painful shock videos with wf, and found evidence against the existence of such a correlation: Kendall's tau=0.010, $BF_{10}=0.149$, $p=0.906$. The same is true when correlating wf with the difference in reaction time between rating the painful shock and non-painful shock videos: Kendall's tau=0.018, $BF_{10}=0.152$, $p=0.812$ ".

Feedback questionnaires (pre- and post- cover story disclosure): In Supplementary Materials §21, we now write: "At the end of the experiment we collected two feedback questionnaires, one before disclosing the cover story ("Participant feedback questionnaire" in Supplementary Figure 1) and one after ("Post disclosure feedback questionnaire" in Supplementary Figure 1). As the two feedback questionnaires do not produce a single summary score, we ran two linear regression with wf as dependent variable and all the answers to the questionnaires as covariates (forward method, note that QQ plot confirms that the requirements for a multiple regression are met). For the feedback questionnaire prior to disclosing the cover story, the linear regression revealed that only Q4 and Q8 contribute significantly to explaining wf (Supplementary Figure 20, ANOVA containing intercept, Q4 and Q8, $F(2,76)=10.948$, $p=6.636 \times 10^{-5}$, $BF_{10}=360$ relative to intercept only). For the feedback questionnaire after disclosure, the first 3 questions relative to whether participants had believed the cover story are analyzed and reported in Supplementary Materials §5. The remaining 8 questions (Q4-Q11) were entered into a multiple regression that was not outperforming an intercept only model ($F(8,70)=0.693$, $p=0.697$. All $BF_{incl}<0.150$)."

Supplementary Figure 20: Multiple regression explaining wf using Q4 and Q8 of the pre-disclosure feedback questionnaire. a Regression explaining the residuals of wf given the residuals of Question 4 of the Feedback questionnaire. **b** Regression explaining the residuals of wf given the residuals of Question 8 of the Feedback questionnaire. Residuals are plotted because these are the results of a multiple regression including Q4 and Q8, and the figure thus shows the unique variance explained by each variable after removing what can be explained by the others. T and p refer to the parameter estimate for that particular variable, BF_{incl} , how much more likely the wf data is given a model including that variable compared to one excluding this variable.

- It is great that the authors use Bayesian statistics, however it is still not the most common method to use, therefore can the authors add a short explanation, similar to line 947 into the beginning of the results section.

We now added, next to the first Bayes factor result in the main text, the note: “We provide Bayes factors (BF), next to p values, to quantify how much more likely the observed data is if an effect were present than if it were absent to infer, using traditional bounds, whether the data provides evidence for an effect ($BF_{10} > 3$) or for the absence of an effect ($BF_{10} < 1/3$), or remains inconclusive ($1/3 < BF_{10} < 3$, see (Christian Keyzers et al., 2020) for a tutorial on how to interpret these tests). ”.

- Typos: Line 216 ‘t’ too much: we corrected this typo.

Line 937: ranging: corrected.

- Figure B last panel: it would be good to explain why 64% is bold. We now specify “The most likely outcome is specified in bold font”.

- How was the sample size for the online study selected? For the fMRI we now state: “Sample size was based on a power calculation to have 80% power to detect a medium effect size for t-tests at $\alpha=0.05$ ”, for the online “Sample size was based on a power calculation to have >80% power to detect a medium associations ($|\rho|=0.3$) between variables across participants at $\alpha=0.05$.”

- Title: all behavioural and neural results aim to show that there is a bias that impacts decision and learning when having to choose between prosocial vs selfish choices. It would be great to see something about this in the title: we now changed the title to: “Neuro-computational mechanisms and individual biases in action-outcome learning under moral conflict”

Bibliography

- Bartra, O., McGuire, J.T., Kable, J.W., 2013. The valuation system: a coordinate-based meta-analysis of BOLD fMRI experiments examining neural correlates of subjective value. *NeuroImage* 76, 412–427. <https://doi.org/10.1016/j.neuroimage.2013.02.063>
- Fouragnan, E., Retzler, C., Philiastides, M.G., 2018. Separate neural representations of prediction error valence and surprise: Evidence from an fMRI meta-analysis. *Hum. Brain Mapp.* 39, 2887–2906. <https://doi.org/10.1002/hbm.24047>
- Gallo, S., Paracampo, R., Müller-Pinzler, L., Severo, M.C., Blömer, L., Fernandes-Henriques, C., Henschel, A., Lammes, B.K., Maskaljunas, T., Suttrup, J., Avenanti, A., Keyzers, C., Gazzola, V., 2018. The causal role of the somatosensory cortex in prosocial behaviour. *eLife* 7. <https://doi.org/10.7554/eLife.32740>
- Kragel, P.A., Koban, L., Barrett, L.F., Wager, T.D., 2018. Representation, Pattern Information, and Brain Signatures: From Neurons to Neuroimaging. *Neuron* 99, 257–273. <https://doi.org/10.1016/j.neuron.2018.06.009>
- Montague, P.R., Dolan, R.J., Friston, K.J., Dayan, P., 2012. Computational psychiatry. *Trends Cogn. Sci.* 16, 72–80. <https://doi.org/10.1016/j.tics.2011.11.018>
- Rushworth, M.F.S., Behrens, T.E.J., 2008. Choice, Uncertainty and Value in Prefrontal and Cingulate Cortex. *Nat. Neurosci.* 11, 389–397.

- Schultz, W., 2013. Updating dopamine reward signals. *Curr. Opin. Neurobiol., Macrocircuits* 23, 229–238. <https://doi.org/10.1016/j.conb.2012.11.012>
- Zhang, L., Lengersdorff, L., Mikus, N., Gläscher, J., Lamm, C., 2020. Using reinforcement learning models in social neuroscience: frameworks, pitfalls and suggestions of best practices. *Soc. Cogn. Affect. Neurosci.* 15, 695–707. <https://doi.org/10.1093/scan/nsaa089>
- Zhou, F., Li, J., Zhao, W., Xu, L., Zheng, X., Fu, M., Yao, S., Kendrick, K.M., Wager, T.D., Becker, B., 2020. Empathic pain evoked by sensory and emotional-communicative cues share common and process-specific neural representations. *eLife* 9, e56929. <https://doi.org/10.7554/eLife.56929>

Reviewers' Comments:

Reviewer #1:

Remarks to the Author:

The authors have addressed my queries. I believe the manuscript will make an important contribution to the field.

Reviewer #2:

Remarks to the Author:

The authors have been very responsive to all comments, have conducted an impressive amount of control analyses, and have changed the text in line with my suggestions. In general, the issues I had with the first version of this manuscript have been addressed and I think the paper is almost ready to be published. The authors should only address the two outstanding issues I describe below.

If the authors wanted to improve their manuscript even further: I have the impression that the manuscript is very detailed and the amount of data, analyses, and text may be a bit much for the average reader. As a consequence, the "take home" message is not very clear. But this may be a matter of personal writing style - it is not up to reviewers to prescribe a certain style of reporting and interpreting results.

On the specific issues I had:

R2C1: I am glad the authors followed my advice and changed the theoretical framing of their study

R2C2: The authors provide many new analyses to scrutinize learning in ambiguous participants. This enables them to provide a much more nuanced picture, and the characterization of the data is fair and unbiased.

R2C3: It is good to see that the authors now report that there is no correlation between the weighting factor and the learning rate, and that the two can be dissociated.

R2C4: It is good for the reader that the authors clarified their hypotheses and that they provide a more comprehensive overview of the results relating to EV and PEs.

R2C5: The new section in the supplement removes the possibility of misunderstandings concerning the wf-normalization of the PEs.

R2C6: While I agree that multivariate analyses can provide interesting information concerning brain network activation that nicely complement univariate analyses of regional activation, I fundamentally disagree with the judgement that these analyses are "new" and that they "outperform" univariate analyses in discovering "specific" results. Corresponding analyses approaches have been around since the first days of neuroimaging analysis and it has always been clear that these approaches can characterize distributed activity patterns without any spatial specificity, whereas univariate analyses identify specific spatial locations where activity relates to task characteristics. Multivariate analysis approaches are therefore neither more "specific" nor do they "outperform" univariate analyses – these two approaches simply have different complementary aims and are both better suited than the other at reaching those specific aims. The authors should remove the misleading statements on page 19 and provide an objective rationale for what questions about information in neural network patterns they were trying to answer with their multivariate signature analysis (which cannot be answered by univariate analyses).

R2C7: It is good that the authors make it explicit that PEM and PES are partially dissociated, included in the same GLM, and correlated with distinct activity patterns.

R2C8: The authors have added the analyses I inquired about

R2C9: The authors claim they carefully tracked all uses of the term "predict" throughout the manuscript and could not find any instance where this term was used to characterize a correlation coefficient. They asked for specific pointers. Here they are:

Page 8: "..., the preference (...) was predictive of their bias (Figure 3e, Pearson's $r = 0.51$)..."

Page 16: "... could significantly predict how much money participants on average donated...(Kendall Tau=-0.47,...)"

It should also be noted that even if one can derive a "predicted value" from a fitted regression equation and an input predictor (as defined in the stats glossary the authors cite), it still appears somewhat misleading to interpret a significant regression weight in a non-cross-validated regression as indicating that the predictor variable indeed "predicts" the observed data used to fit that model. In most fields, it is good scientific practice to only interpret a relationship as predictive if this has been tested with unseen data, because a single regression model can obviously be overfitted to the specific dataset used to derive the weights. So I encourage the authors to follow this convention and use prediction for their out-of-sample predictions that are not prone to overfitting, and more cautious terminology for the other analyses.

R2C10-R2C12 have all been addressed.

Reviewer #3:

Remarks to the Author:

I have no further remarks, thank you for addressing the comments thoroughly.

Reply to Reviewers

Reviewer #1 (Remarks to the Author):

The authors have addressed my queries. I believe the manuscript will make an important contribution to the field.

We thank the reviewer for their kind comment.

Reviewer #2 (Remarks to the Author):

The authors have been very responsive to all comments, have conducted an impressive amount of control analyses, and have changed the text in line with my suggestions. In general, the issues I had with the first version of this manuscript have been addressed and I think the paper is almost ready to be published. The authors should only address the two outstanding issues I describe below.

If the authors wanted to improve their manuscript even further: I have the impression that the manuscript is very detailed and the amount of data, analyses, and text may be a bit much for the average reader. As a consequence, the “take home” message is not very clear. But this may be a matter of personal writing style - it is not up to reviewers to prescribe a certain style of reporting and interpreting results.

On the specific issues I had:

R2C1: I am glad the authors followed my advice and changed the theoretical framing of their study

R2C2: The authors provide many new analyses to scrutinize learning in ambiguous participants. This enables them to provide a much more nuanced picture, and the characterization of the data is fair and unbiased.

R2C3: It is good to see that the authors now report that there is no correlation between the weighting factor and the learning rate, and that the two can be dissociated.

R2C4: It is good for the reader that the authors clarified their hypotheses and that they provide a more comprehensive overview of the results relating to EV and PEs.

R2C5: The new section in the supplement removes the possibility of misunderstandings concerning the wf-normalization of the PEs.

R2C6: While I agree that multivariate analyses can provide interesting information concerning brain network activation that nicely complement univariate analyses of regional activation, I fundamentally disagree with the judgement that these analyses are “new” and that they “outperform” univariate analyses in discovering “specific” results. Corresponding analyses approaches have been around since the first days of neuroimaging analysis and it has always been clear that these approaches can characterize distributed activity patterns without any spatial specificity, whereas univariate analyses identify specific spatial locations where activity relates to task characteristics. Multivariate analysis approaches are therefore neither more “specific” nor do they “outperform” univariate analyses – these two approaches simply have different complementary aims and are both better suited than the other at reaching those specific aims. The authors should remove the misleading statements on page 19 and provide an objective rationale for what questions about information in neural network patterns they were trying to answer with their multivariate signature analysis (which cannot be answered by univariate analyses).

We replaced the passage on page 19 with more neutral language:

“We used neural signatures because they provide a principled method to reduce the involvement of a distributed neurocognitive systems to a univariate measure that can then be analyzed using Bayes factor hypothesis testing to provide evidence both for or against the involvement of this system^{30,35–37}.”

R2C7: It is good that the authors make it explicit that PEM and PES are partially dissociated, included in the same GLM, and correlated with distinct activity patterns.

R2C8: The authors have added the analyses I inquired about

R2C9: The authors claim they carefully tracked all uses of the term “predict” throughout the manuscript and could not find any instance where this term was used to characterize a correlation coefficient. They asked for specific pointers. Here they are:

Page 8: “..., the preference (...) was predictive of their bias (Figure 3e, Pearson’s $r = 0.51$)...”

We thank the reviewer for pointing out this specific passage. We replaced predictive of the bias with “was correlated with their bias”.

Page 16: "... could significantly predict how much money participants on average donated...(Kendall Tau=-0.47,...)"

We thank the reviewer for pointing out this specific passage. We replaced the "*could significantly predict*" with "*was significantly correlated with*".

It should also be noted that even if one can derive a "predicted value" from a fitted regression equation and an input predictor (as defined in the stats glossary the authors cite), it still appears somewhat misleading to interpret a significant regression weight in a non-cross-validated regression as indicating that the predictor variable indeed "predicts" the observed data used to fit that model. In most fields, it is good scientific practice to only interpret a relationship as predictive if this has been tested with unseen data, because a single regression model can obviously be overfitted to the specific dataset used to derive the weights. So I encourage the authors to follow this convention and use prediction for their out-of-sample predictions that are not prone to overfitting, and more cautious terminology for the other analyses.

We thank the reviewer for this well taken point.

R2C10-R2C12 have all been addressed.

Reviewer #3 (Remarks to the Author):

I have no further remarks, thank you for addressing the comments thoroughly.